# Provably Efficient RL for Linear MDPs under Instantaneous Safety Constraints in Non-Convex Feature Spaces

**Amirhossein Roknilamouki** [1]  **Arnob Ghosh** [2]  **Ming Shi** [3]  **Fatemeh Nourzad** [1]  **Eylem Ekici** [1]  **Ness B. Shroff** [1 4]

## Abstract

In Reinforcement Learning (RL), tasks with instantaneous hard constraints present significant challenges, particularly when the decision space is non-convex or non-star-convex. This issue is especially relevant in domains like autonomous vehicles and robotics, where constraints such as collision avoidance often take a non-convex form, and the state-space may be large. In this paper, we establish a regret bound of $\tilde{\mathcal{O}}\big((1 + \frac{1}{\tau})\sqrt{\log(\frac{1}{\tau})d^3 H^4 K}\big)$, applicable to both star-convex and non-star-convex cases, where $d$ is the feature dimension, $H$ the episode length, $K$ the number of episodes, and $\tau$ the safety threshold for a linear MDP setting. Moreover, the violation of safety constraints is *zero* with a high probability throughout the learning process. A key technical challenge in these settings is bounding the covering number of the value-function class, which is essential for achieving value-aware uniform concentration in model-free function approximation. For the star-convex setting, we develop a novel technique called *Objective–Constraint Decomposition* (OCD) to properly bound the covering number, and resolves an error in a previous work on the constrained RL. In non-star-convex scenarios, where the covering number can become infinitely large, we propose a two-phase algorithm, Non-Convex Safe Least Squares Value Iteration (NCS-

LSVI), which first reduces uncertainty about the safe set by playing a known safe policy. After that, it carefully balances exploration and exploitation to achieve the regret bound. Finally, numerical simulations on an autonomous driving scenario demonstrate the effectiveness of NCS-LSVI.

## 1. Introduction

Safe Reinforcement Learning (RL) has emerged as a powerful framework for safe online learning, enabling agents to learn autonomously from their environments while respecting safety constraints (Tessler et al., 2018; Ghosh et al., 2022b). In many safety-critical applications—such as autonomous driving, healthcare, and financial planning—it is imperative that the agent not only maximizes cumulative rewards but also maintains safety at every step of the decision-making process (Shi et al., 2023; Amani et al., 2021). These applications often feature instantaneous hard constraints, which must be satisfied at each time step to prevent catastrophic outcomes (e.g., avoiding collisions in autonomous driving). Unlike constraints that allow violations in expectation or across an entire trajectory, instantaneous hard constraints demand adherence strictly at every decision point, highlighting the need for RL methods that can explore and exploit without compromising immediate safety.

Amani et al. (2021) tackled this challenge by introducing a strategy for Safe RL with instantaneous hard constraints in star-convex decision spaces within linear MDPs. Amani et al. (2021) assume that at every state, there exists an initial safe action that is known to the RL agent. The agent begins interacting with the environment using this initial safe action, subsequently constructing and refining an estimated safe set of actions that remain within the actual safe set of actions. However, as described below, *their analysis to prove sublinear regret is not correct* (with further details provided in Section 5.1). The key issue arises in bounding the covering number for the value function, a crucial step for achieving value-aware uniform concentration in model-free function approximation. Their Theorem 2 relies on results from unconstrained RL (Lemma D.6 in Jin et al. (2020)), where the value function is obtained by maximizing the

[1]Department of Electrical and Computer Engineering, The Ohio State University, Columbus, OH, USA, [2]Department of Electrical and Computer Engineering, New Jersey Institute of Technology, Newark, NJ, USA [3]Department of Electrical Engineering University at Buffalo, Buffalo, NY, USA [4]Department of Computer Science and Engineering, The Ohio State University, Columbus, OH, USA,. Correspondence to: Amirhossein Roknilamouki <roknilamouki.1@osu.edu>, Arnob Ghosh <arnob.ghosh@njit.edu>, Ming Shi <mshi24@buffalo.edu>, Fatemeh Nourzad <nourzad.1@osu.edu>, Eylem Ekici <ekici.2@osu.edu>, Ness B. Shroff <shroff.11@osu.edu>.

*Proceedings of the $42^{nd}$ International Conference on Machine Learning*, Vancouver, Canada. PMLR 267, 2025. Copyright 2025 by the author(s).

Q-function over a fixed action set. In such cases, a covering of the Q-function automatically induces a covering of the value function, thanks to the contraction property of the max operator. However, in the constrained RL framework of Amani et al. (2021), the value function is computed by maximizing the Q-function over an estimated safe set that depends on the random historical data. Consequently, even when the Q-function remains the same, differences in the corresponding safe sets can lead to drastically different value functions. This invalidates the direct application of unconstrained RL's covering-number bounds, and the analysis in Amani et al. (2021) must be significantly revised to account for the dynamic nature of the safe set.

As our **first main contribution**, we show how to *correctly bound the covering number* when the feasible action set is updated over time in the star-convex case. Specifically, we identify that star-convexity provides a form of smooth geometry—small variations in the safety parameters do not cause *drastic* changes in the estimated safe set. Consequently, if two Q-functions are close, the corresponding value functions—obtained by maximizing each Q-function over these slightly different feasible sets—also remain close. Leveraging this property, we introduce the novel OCD (Objective–Constraint Decomposition) technique, which replaces the unconstrained covering-number argument with a refined bound that includes an additional term of the form $\mathcal{O}\left(\sqrt{\log\left(\frac{1}{\tau}\right)}\right)$, where $\tau$ is the safety threshold. This factor reflects how tighter safety requirements (*i.e.*, smaller $\tau$) make bounding the covering number more challenging, offering a fresh perspective on the cost of satisfying *instantaneous* constraints (See Remark 6.1).

While star-convex geometry can be gentle on covering-number bounds, real-world problems such as autonomous driving and robotics often induce non-star-convex or highly irregular safe decision spaces—e.g., disjoint regions due to obstacles or kinematic constraints. In these settings, as detailed in Section 5.2, the covering number can become arbitrarily large, rendering existing star-convex-based methods insufficient and highlighting the *necessity of developing new methods for non-star-convex environments*.

Motivated by these observations, we propose a new two-phase algorithm for non-star-convex scenarios that satisfy our Local Point Assumption (see Section 3), which we refer to as Non-Convex Safe Least Squares Value Iteration (NCS-LSVI). Drawing from our insight in star-convex analysis, we observe that controlling drastic changes in the estimated safe set under small variations in constraint parameters is a key strategy for bounding the covering number. Based on this, NCS-LSVI includes a pure-safe exploration phase, where the agent samples randomly from safe actions in a small neighborhood of the initial safe policy (whose existence is guaranteed by the Local Point Assumption). By the end of

this phase, the estimated safe set remains stable with high probability, enabling tighter covering-number bounds.

Having established this stable safe set, the agent then proceeds to an exploration–exploitation phase, refining its policy under a bounded covering-number framework. As our second contribution, we show that NCS-LSVI achieves a regret bound of $\tilde{\mathcal{O}}\big((1+\frac{1}{\tau})\sqrt{\log(\frac{1}{\tau})d^3H^4K} + \frac{1}{\epsilon^2\iota^2}\big)$ with high probability, nearly matching the regret in convex and star-convex cases while also respecting instantaneous hard constraints. Here, $d$ represents the feature space dimension, $H$ the episode length, $K$ the number of episodes, and $\tau$ a safety related parameter. The bounded constant parameters $\epsilon$ and $\iota$ are related to our Local Point Assumption (3.2). *To the best of our knowledge, this is the first result for non-star-convex settings.* Additionally, we conduct a numerical experiment on a merging scenario in autonomous driving, where the safe set is non-star-convex due to collision-avoidance constraints. The results demonstrate sublinear regret, consistent with the theoretical upper bound, and highlight the practical potential of our two-phase framework.

Pure exploration has also been studied in the context of safe linear bandits by Amani et al. (2019), where it was specifically designed to ensure that the optimal point is included in the estimated safe set at the end of the exploration phase. In comparison, linear bandits are simpler than RL, as they do not involve estimating a value function or managing the covering number. In our work, the safe pure exploration phase serves a different purpose: it is specifically designed to control the covering number in the second phase of our algorithm, enabling near-optimal performance even in non-star-convex spaces.

Altogether, our work highlights the pivotal role of the decision space's geometry in shaping the complexity of safe RL. While this study focuses on linear function approximation and local connectivity, the broader takeaway is that *additional mechanisms* beyond those used in unconstrained RL are crucial for maintaining tight covering numbers under instantaneous hard constraints. This insight opens up promising directions for future research, particularly in *deep* RL, where nonlinear representations often create irregular feature spaces. These complexities further underscore the importance of understanding how the geometry of the feature space impacts overall performance.

**Other related works.** RL problems with cumulative constraints are studied (Wu et al., 2016; Achiam et al., 2017; Tessler et al., 2018; Yang et al., 2019; Efroni et al., 2020; Qiu et al., 2020; Ding et al., 2021; Bai et al., 2022; Wei et al., 2022; Paternain et al., 2022; Ghosh et al., 2022b; Vaswani et al., 2022; Ghosh et al., 2022a; Ding & Lavaei, 2023; Ghosh & Zhou, 2023; Huang et al., 2023; Ghosh et al., 2024). This line of work focuses on ensuring the expected

cumulative cost remains below a threshold, unlike instantaneous constraints that must be satisfied with high probability at each time step. Another relevant direction is explored in Berkenkamp et al. (2017), which explicitly considers stability-based safety guarantees using Lyapunov methods within model-based reinforcement learning to certify policy safety during exploration.

**Bandits with instantaneous hard constraints:** Bandits with instantaneous constraints have been studied in Amani et al. (2019); Khezeli & Bitar (2020); Moradipari et al. (2020a;b; 2021); Pacchiano et al. (2021); Zhou & Ji (2022); Deng et al. (2022); Pacchiano et al. (2024); Hutchinson et al. (2024); Afsharrad et al. (2024). Unlike RL, Bandits do not require the estimation of a value function, and therefore the problem of covering number does not arise in this setting. This distinction significantly simplifies the analysis and algorithm design in the context of Bandits compared to RL with instantaneous hard constraints.

**RL with instantaneous hard constraints:** Problems with unsafe states in a star-convex setting have been studied in Shi et al. (2023). However, this setting focuses on the linear mixture model, which is fundamentally different from our setting, as explained in Section 2. Lastly, work in Wei et al. (2024) relaxed the assumption that a prior safe action is given to the algorithm, instead allowing sublinear constraint violation. Thus, none of the above works have studied RL with instantaneous hard constraints for non-star-convex decision spaces.

## 2. Problem formulation

In this study, we focus on an episodic constrained MDP, denoted as $(\mathcal{S}, \mathcal{A}, H, \mathbb{P}, r, c)$, operating in an online setting over $K \in \mathbb{N}$ episodes. Here, $\mathcal{S}$ represents the state space, which may contain an infinite number of states; $\mathcal{A}$ denotes a continuous action space; and $H \in \mathbb{N}$ determines the number of steps within each episode. Additionally, $\{\mathbb{P}\}_{h=1}^H$, $\{r\}_{h=1}^H$, and $\{c\}_{h=1}^H$ correspond to the state transition probability kernel, reward function, and cost function at each step, respectively. During episode $k \in [K]$ and time step $h \in [H]$, the learner is in state $s_h^k \in \mathcal{S}$, interacting with the environment by selecting an action $a_h^k \in \mathcal{A}$. Subsequently, the learner observes a noisy reward $\hat{r}_h^k(s_h^k, a_h^k) = r_h(s_h^k, a_h^k) + \eta_h^k$, where $r_h(.) : \mathcal{S} \times \mathcal{A} \to [0, 1]$ represents an unknown function, and $\eta_h^k$ denotes a zero-mean $\sigma$-sub-Gaussian random variable. In addition, it observes a corresponding noisy cost $\hat{c}_h^k(s_h^k, a_h^k) = c_h(s_h^k, a_h^k) + \zeta_h^k$, where $c_h(.) : \mathcal{S} \times \mathcal{A} \to [0, 1]$ is an unknown cost function, and $\zeta_h^k$ is a zero-mean $\sigma$-sub-Gaussian random variable as well. Finally, the transition to the next state, $s_{h+1}^k$, under the action $a_h^k$ at state $s_h^k$, is determined by an unknown transition probability kernel denoted as $\mathbb{P}_h(.|s_h^k, a_h^k)$. The episode terminates at step $H + 1$.

**Instantaneous hard constraint.** In each episode $k$ and at each step $h$, the learner is required to adhere to a hard constraint: $c_h(s_h^k, a_h^k) \leq \tau$, where $\tau$ is a known positive constant that serves as the safety threshold. For each state $s \in \mathcal{S}$, the corresponding safe set of actions is given by $\mathcal{A}_h^{\text{safe}}(s) = \{a \in \mathcal{A} : c_h(s, a) \leq \tau\}$. Our framework naturally extends to settings involving multiple constraints by defining the estimated safe set as $\mathcal{A}_h^k(s) = \bigcap_{j=1}^M \mathcal{A}_h^{k,j}(s)$, where each $\mathcal{A}_h^{k,j}(s)$ represents the safe action set associated with the $j$-th constraint.

**Safe policy.** A deterministic policy is defined as a mapping $\pi(s, h) : \mathcal{S} \times [H] \to \mathcal{A}$. Thus, a policy $\pi$ is called safe, if, for every state $s \in \mathcal{S}$ and time step $h$, it satisfies the instantaneous constraint. Therefore, the set of all safe policies is defined by $\Pi^{\text{safe}} = \{\pi(.) : \pi(s, h) \in \mathcal{A}_h^{\text{safe}}(s), \forall(s, h) \in \mathcal{S} \times [H]\}$.

**Performance metric.** Given a policy $\pi$, the corresponding $V$-value and the $Q$-value are defined as follows: $V_h^\pi(s) = E[\Sigma_{h'=h}^H r_{h'}(s_{h'}, \pi(h', s_{h'}))|s_h = s]$, and $Q_h^\pi(s, a) = r_h(s, a) + E[\Sigma_{h'=h+1}^H r_{h'}(s_{h'}, \pi(h', s_{h'}))|s_h = s, a_h = a]$. Assume that each episode starts with a fixed initial state $s_1 \in \mathcal{S}$. The agent, at each episode $k \in [K]$, employs the policy $\pi^k$ to interact with the environment. We then evaluate the performance of the set of policies $\{\pi^k\}_{k=1}^K$ by the well-studied regret metric, defined as $\text{Regret}(K) \triangleq \sum_{k=1}^K [V_1^{\pi^*}(s_1) - V_1^{\pi^k}(s_1)]$. Note that $\pi^*$ is the optimal safe policy that maximizes $V$-value function, while remaining safe $\pi^* \in \Pi^{\text{safe}}$.

**Linear MDP.** To handle the large and potentially infinite number of states and actions, we concentrate on linear MDPs. This choice enables us to employ linear function approximation methods to solve our problem effectively.

**Assumption 2.1.** (Linear MDP (Ghosh et al., 2022b), (Jin et al., 2020)) Consider an episodic constrained MDP denoted as $(\mathcal{S}, \mathcal{A}, H, \mathbb{P}, r, c)$, which is assumed to be a linear MDP with a feature function $\phi : \mathcal{S} \times \mathcal{A} \to \mathcal{F} \subset \mathbb{R}^d$. Specifically, for each $h \in [H]$, there exist $d$ unknown measures $\mu_h = \{\mu_h^1, \ldots, \mu_h^d\}$ over the state space $\mathcal{S}$ and unknown vectors $\theta_h^*, \gamma_h^* \in \mathbb{R}^d$ such that for any $(s, a, s') \in \mathcal{S} \times \mathcal{A} \times \mathcal{S}$, the transition probabilities, cost function, and reward function are given by: $\mathbb{P}_h(s'|s, a) = \langle \phi(s, a), \mu_h(s') \rangle$, $r_h(s, a) = \langle \phi(s, a), \theta_h^* \rangle$, and $c_h(s, a) = \langle \phi(s, a), \gamma_h^* \rangle$, respectively. Additionally, we assume without loss of generality that for all $(s, a) \in \mathcal{S} \times \mathcal{A}$, we have $\|\phi(s, a)\| \leq L$ for some $L \in (0, 1]$, and $\max(\|\mu(\mathcal{S})\|, \|\theta_h^*\|, \|\gamma_h^*\|) \leq \sqrt{d}$, where $d$ is the dimension of the feature space.

Assumption 2.1 encapsulates the linear relationship of the transition probabilities, costs, and rewards with the feature map. It is important to note that despite the linearity, the feature map $\phi(.)$ itself may potentially be non-linear or even non-convex. Note that, the linearity of the transition

probabilities, results in the fact that all features points are located on a $d-1$ dimensional hyper plane as explained in the following proposition:

**Proposition 2.2.** *(Proposition A.1. from Jin et al. (2020)). Let $\mathcal{F}_s \triangleq \{\phi(s,a) \in \mathbb{R}^d \mid a \in \mathcal{A}\}$, and $\mathcal{F} \triangleq \{\phi(s,a) \in \mathbb{R}^d \mid (a,s) \in \mathcal{A} \times \mathcal{S}\}$. Then, there exists a vector $\mu^* \in \mathbb{R}^d$ such that $\mathcal{F}_s \subset \mathcal{F} \subset \mathcal{H}$, where $\mathcal{H}$ is a $d-1$ dimensional hyper plane determined by $\mathcal{H} \triangleq \{x \in \mathbb{R}^d \mid \langle x, \mu^* \rangle = 1\}$.*

**Initial safe action.** Designing a safe RL algorithm that achieves sublinear regret requires at least one known safe action per state $s \in \mathcal{S}$, as shown in Theorem 3 of (Shi et al., 2023). This assumption is often valid in real-world scenarios where a known, albeit suboptimal, safe strategy exists (Amani et al., 2019; 2021; Khezeli & Bitar, 2020). In this study, we also adopt a similar assumption as stated bellow.

**Assumption 2.3.** For each state $s \in \mathcal{S}$, there exists a safe action $a_s^0$ that incurs a zero cost, i.e., $\langle \phi(s, a_s^0), \gamma_h^* \rangle = 0$.

*Remark* 2.4. We highlight that for problems where the initial action results in a non-zero cost $\tau_0$, the original problem can be converted to an equivalent one that satisfies Assumption 2.3 through a simple translation. In the new problem, the safety threshold is adjusted to $\tau - \tau_0$.

**Notations.** For any positive semi-define matrix $A$ and vector $v$, the operator $\|v\|_A$ defines the weighted norm as $\|v\|_A \triangleq \sqrt{v^T A v}$. The symbol $\|.\|$ denotes the $l_2$ norm. Also, we define $\mathbb{P}_h V_{h+1}^\pi(s,a) \triangleq \mathbb{E}_{s' \sim \mathbb{P}_h(\cdot \mid s,a)} V_{h+1}^\pi(s')$. The ball $\mathbb{B}_\epsilon(v)$ represents the set of all points within distance $\epsilon$ from $v$.

## 3. Non-Convex Feature Spaces

Non-convexity often occurs in real-world problems due to non-convex constraints such as traffic rules, collision avoidance, and kinematic restrictions in autonomous vehicles, as well as in feature selection and function approximation in modern RL methods. Here, we define two key assumptions in the feature space $\mathcal{F}_s$—*star-convexity* and the *Local Point Assumption*-commonly found in these applications and will later provide near-optimal solutions for them. To start, we define the notion of star-convexity, introduced in Amani et al. (2021):

**Assumption 3.1. (Star-Convex Sets)** For each $s \in \mathcal{S}$, the feature space $\mathcal{F}_s$ is star-convex around $\phi(s, a_s^0)$. That is, for all $\mathbf{x} \in \mathcal{F}_s$ and $\alpha \in [0,1]$: $\alpha \mathbf{x} + (1-\alpha) \phi(s, a_s^0) \in \mathcal{F}_s$.

Although star-convexity simplifies the theoretical analysis, many real-world applications—including autonomous driving—cannot satisfy this assumption as we explain the following example:

**Example (Autonomous Driving):** Consider an autonomous vehicle approaching an intersection and facing a merging

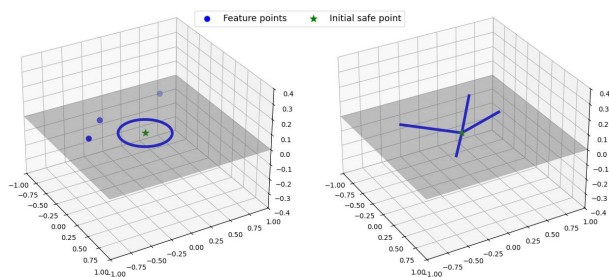

*Figure 1.* An illustrative example of Assumptions 3.2 and 3.1. The left figure demonstrates the Local Point Assumption, where a sphere exists around the initial safe point. The right figure depicts a Star-Convex Set, where all points are connected to the initial safe point by a line segment.

decision (Figure 2). The car can either (i) drive slowly to let oncoming traffic pass or (ii) accelerate rapidly to merge before other vehicles arrive. These two action modes induce disjoint feasible regions, violating Assumption 3.1 because no straight line connecting the points in the slow mode to the points in high acceleration mode resides entirely in the decision space. Note that these distinct action subsets naturally emerge from collision avoidance constraints and results in a non-convex decision set.

To address the limitations of star-convex spaces, we introduce a new class of non-convex sets, termed the *Local Point Assumption*, which satisfies conditions relevant to autonomous vehicles and robotics. These sets feature localized properties around $\phi(s, a_s^0)$ and near constraint boundaries within the feature space, while allowing arbitrary structures outside these regions.

**Assumption 3.2. (Local Point Assumption)** Let $\mathcal{H}$ be the $(d-1)$-dimensional hyperplane containing $\mathcal{F}$. There exists $\epsilon \in (0, \frac{\tau}{\sqrt{d}})$ such that $\mathbb{B}_\epsilon(\phi(s, a_s^0)) \cap \mathcal{H} \subset \mathcal{F}_s$ for all $s \in \mathcal{S}$. Moreover, for any $(s, a, h)$, if $\tau - \iota \leq \langle \phi(s,a), \gamma_h^* \rangle \leq \tau$ for some $\iota < \tau$, then $\{\nu \phi(s,a) + (1-\nu) \phi(s, a_s^0) \mid \nu \in [\frac{\tau - \iota}{\langle \phi(s,a), \gamma_h^* \rangle}, 1]\} \subset \mathcal{F}_s$.

**Comparison with Assumption 3.1.** The key difference between Assumption 3.1 and Assumption 3.2 is that star-convexity imposes global requirements on all points in $\mathcal{F}_s$, whereas the Local Point Assumption focuses solely on local properties. Intuitively, the Local Point Assumption demands that we can slightly perturb the initial safe action $a_s^0$ and still remain safe, enabling sampling from a small neighborhood around $a_s^0$. For instance, in the autonomous driving scenario described earlier, if a particular speed $v_0$ is considered safe, then speeds within the interval $[v_0 - \epsilon, v_0 + \epsilon]$ must also be safe. Furthermore, the second condition in the Local Point Assumption requires local connectivity near the constraint boundary: if an optimal action $v^*$ lies exactly at the constraint boundary, then actions in the range $[v^* - \iota, v^*]$

must also remain within the safe region. However, the Local Point Assumption is not always less restrictive, as it requires the existence of a hypersphere on the plane $\mathcal{H}$, a condition not mandated by star-convexity. In practice, though, it is an effective assumption for handling complex decision spaces, particularly in applications such as autonomous vehicles and robotics. For an illustrative example of star-convex sets, see Figure 1. Consequently, these structural differences lead to distinct regret bounds under the two assumptions (see Section 5).

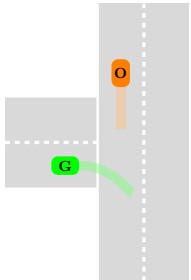 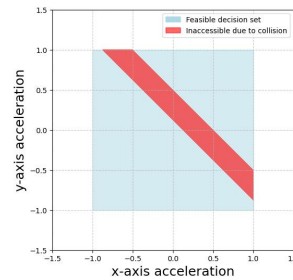

*Figure 2.* **Left figure:** The green car (G) must decide whether to stop at the intersection for the approaching orange car (O) or accelerate to pass before O arrives. **Right figure:** The green car's decision space, where the red region is inaccessible due to the collision avoidance module.

## 4. Our Approach

We now describe our proposed algorithm, Non-Convex Safe Least Square Value Iteration (NCS-LSVI), detailed in Algorithm 1. This algorithm extends the standard SLUCB-QVI from (Amani et al., 2021) to address safe RL problems in non-star-convex environments. A key feature of our approach is the ability to handle large covering numbers in non-star-convex environments using the pure exploration phase. The algorithm consists of two main phases: **pure safe exploration** and **safe exploitation-exploration**, implemented over $K$ episodes. At a high level, NCS-LSVI ensures safe learning by leveraging a Recursive Least Squares (RLS) framework to construct an estimated safe set $\mathcal{A}_h^k(s)$. Subsequently, it employs an optimistic $Q$-function estimation to guide policy updates. Detailed explanations of the algorithm's components are provided below.

**Lines** $1-6$ **(Pure safe exploration).** In this phase the agent samples uniformly from a set $\mathcal{D}^\epsilon(s)$ defined as follows:

$$\mathcal{D}^\epsilon(s) \triangleq \{a \in \mathcal{A} \mid \|\phi(s,a) - \phi(s,a_s^0)\| = \epsilon\}. \quad (1)$$

According to Assumption 3.2, $\mathcal{D}^\epsilon(s)$ represents the boundary of $\mathbb{B}_{\epsilon'}(\phi(s,a_s^0)) \cap \mathcal{H}$. Note that given Assumption 2.3 and the linearity of the problem (Assumption 2.1), the Cauchy-Schwarz inequality can be applied to confirm that

---

**Algorithm 1** Non-Convex Safe Least Square Value Iteration (NCS-LSVI)

**Require:** $\epsilon$, $K'$, $K$, $\nu$, $s_1$
1: **Pure safe exploration**
2: **for** $k = 1, \ldots, K'$ **do**
3:     **for** $h = 1, \ldots, H$ **do**
4:        At state $s$ take action $a_h^k$ randomly from $\mathcal{D}^\epsilon(s)$ according to Eq. (1).
5:     **end for**
6: **end for**
7: **Safe exploitation-exploration**
8: **for** episode $k = K' + 1, \ldots, K$ **do**
9:     **for** step $h = H, \ldots, 1$ **do**
10:       Compute $\Lambda_h^{k,Q}, \Lambda_h^{k,\gamma}, \gamma_h^k, w_h^k$ according to Eqs. (2-3)
11:       Compute estimated safe set, $\forall s \in \mathcal{S}$ :

$$\mathcal{A}_h^k(s) \triangleq \{a \in \mathcal{A} : \langle \phi(s,a) - \phi(s,a_s^0), \gamma_h^k \rangle$$
$$+ \beta_2 \|\phi(s,a) - \phi(s,a_s^0)\|_{(\Lambda_h^{k,\gamma})^{-1}} \leq \tau\}$$

12:       Compute $Q_h^k(s,a) := \langle \phi(s,a), w_h^k \rangle + b_h^k(s,a)$.
13:       Take action $a_h^k = argmax_{a \in \mathcal{A}_h^k(s)} Q_h^k(s,a)$.
14:     **end for**
15:     **for** step $h = 1$ to $H$ **do**
16:       Play $a_h^k$ and observe its reward $r_h^k$ and cost $c_h^k$.
17:     **end for**
18: **end for**

---

all actions during the pure exploration phase are safe, i.e., $\mathcal{D}^\epsilon(s) \subset \mathcal{A}_h^{\text{safe}}(s)$.

**Role of pure exploration.** In non-convex environments, small changes in constraint parameters can dramatically alter the estimated safe set, leading to large covering numbers (as shown by our toy example in Section 5.2). However, once the pure exploration phase is complete, under the Local Point Assumption (Assumption 3.2), the safe set near the constraint boundary becomes smooth and stable. Consequently, minor perturbations no longer cause drastic changes, allowing us to control the covering number effectively. Under Star-Convexity (Assumption 3.1), this global smoothness and stability is inherent, we prove that the pure exploration phase is not needed to achieve a bounded covering number (see Section 6).

**Lines 7-14 (Safe exploitation-exploration).** Once the pure exploration phase has been finished, we start the safe exploitation-exploration which contains two main stages: Safe set construction and $Q$- function estimation.

**Safe-set estimation.** We use Recursive Least Squares (RLS) for safe-set construction, with the modification that we consider the difference in features relative to the initial starting point. In fact, to satisfy the safety condition, it is crucial to

determine how far we can deviate from the initial starting point. Therefore, we formulate the following RLS problem:

$$
\begin{aligned}
\Lambda_h^{k,\gamma} &\triangleq \Sigma_{\tau=1}^{k-1}(x_h^\tau)(x_h^\tau)^\top + \lambda I \\
\gamma_h^k &= (\Lambda_h^{k,\gamma})^{-1}\Sigma_{\tau=1}^{k-1}(x_h^\tau)c_h^\tau(s_h^\tau, a_h^\tau)
\end{aligned}
\tag{2}
$$

where $x_h^\tau \triangleq \phi(s_h^\tau, a_h^\tau) - \phi(s_h^\tau, a_{s_h}^0)$ Note that Considering Assumption 2.3, then Theorem 2 from Abbasi-Yadkori et al. (2011) demonstrate that for any $\delta \in (0, 1)$, the choice of $\beta_2 = \sigma\sqrt{d\log\left(\frac{2+\frac{2KH}{\lambda}}{\delta}\right)} + \sqrt{\lambda d}$ ensures that $\mathcal{A}_h^k(s) \subset \mathcal{A}_h^{\text{safe}}(s)$ holds with probability of at least $1 - \delta$.

**Q-function estimation:** According to Proposition 2.3 in Jin et al. (2020), for a linear MDP, the $Q$-value of the optimal policy $\pi^*$ can be expressed as $Q_h^{\pi^*}(s, a) = \langle \phi(s,a), w_h^{\pi^*}\rangle$. Thus, in line 12 of Algorithm 1, we utilize Regularized Least Squares (RLS) to estimate $w_h^{\pi^*}$ as follows:

$$
w_h^k = (\Lambda_h^{k,Q})^{-1}\Sigma_{\tau=1}^{k-1}\phi(s_h^\tau, a_h^\tau)(y_h^{\tau,k}),
\tag{3}
$$

where $y_h^{\tau,k} \triangleq r_h^\tau(s_h^\tau, a_h^\tau) + V_{h+1}^k(s_{h+1}^\tau)$, $\Lambda_h^{k,Q} = \Sigma_{\tau=1}^{k-1}\phi(s_h^\tau, a_h^\tau)\phi(s_h^\tau, a_h^\tau)^\top + \lambda I$, and $V_{h+1}^k \triangleq \min\{\max_{a\in\mathcal{A}_h^k(s)} Q_h^k(s,a), H\}$.

**Bonus design and geometric assumptions.** Following Amani et al. (2021), we include a bonus term in Step 5 to encourage exploring unexplored safe actions: $b_h^k(s, a) \triangleq \beta_1\|\phi(s,a)\|_{(\Lambda_h^k)^{-1}} + g_{h,\nu}^k(s,a)$, where the first term is a standard exploration bonus from unconstrained RL (Jin et al., 2020), and the second term is defined as follows:

$$
g_{h,\nu}^k(s,a) \triangleq \nu \times \left(\beta_2\|\phi(s,a) - \phi(s,a_s^0)\|_{(\Lambda_h^{k,\gamma})^{-1}}\right) H.
\tag{4}
$$

$g_{h,\nu}^k$ accounts for the distance between the optimal action $\pi^*(s,h)$ and the estimated safe set $\mathcal{A}_h^k(s)$. Under star-convex sets (Assumption 3.1), Amani et al. (2021) show that there exists a scaling factor $\alpha \geq (1 - \frac{\tau}{\tau+2\beta_2\|\phi(s,\pi^*(s,h))-\phi(s,a_s^0)\|_{(\Lambda_h^{k,\gamma})^{-1}}})$ such that $\alpha\phi(s, \pi^*(s,h)) + (1-\alpha)\phi(s, a_s^0) \in \phi(s, \mathcal{A}_h^k(s))$. In non-star-convex settings satisfying our Local Point Assumption 3.2, we show that this property holds for $k \geq K'$, i.e., after the pure exploration phase. This ensures that $g_{h,\nu}^k$ compensates for the remaining distance between $\alpha\phi(s, \pi^*(s,h)) + (1-\alpha)\phi(s, a_s^0)$ and $\phi(s, \pi^*(s,h))$, while encouraging the algorithm to expand the estimated safe set toward the optimal action.

**Lines 15-17 (Environment interaction).** Finally, in steps 15-17, the algorithm plays the selected actions and observes their corresponding rewards and costs, which are stored for use in the next round. The steps 8-17 is repeated for $K - K'$ iterations.

## 5. Analysis

In this section, we first present our main theoretical results for star-convex cases. Then, we explain that ( quite interestingly), in contrast to the unconstrained RL, the geometry of the decision set, i.e. $\mathcal{F}_s$, can affect the covering number of the class of value functions in constrained RL.

### 5.1. Star-Convex Results

**Theorem 5.1.** *Regret in Star-Convex Spaces (Refined version of Theorem 1 in Amani et al. (2021)) Under assumptions 2.1, 2.3, and 3.1, there exists a constant $c_\beta > 0$ such that for any $\delta \in (0, \frac{1}{3})$, by setting $K' = 0$, $\beta_1 = c_\beta dH\sqrt{\log(\frac{dK}{\tau\delta})}$, $\nu = \frac{2}{\tau}$ and $\lambda = 1$, with the probability of at least $1 - 3\delta$ Algortihm 1 remains safe, $\mathcal{A}_h^k(s) \subset \mathcal{A}_h^{safe}(s)$, $\forall(h,k) \in [H] \times [K]$. Moreover, with probability at least $1 - 3\delta$, Algorithm 1 achieves: $\text{Regret}(K) \leq 2H\sqrt{KH \log(\frac{d(K)H}{\delta})} + 2(\beta_1 H + \frac{3\beta_2 H^2}{\tau L})\sqrt{2d(K) \log(1 + \frac{K}{d\lambda})}$.*

**Comparison with Theorem 1 in Amani et al. (2021).**

Compared to our Theorem 5.1, the regret bound in Theorem 1 of Amani et al. (2021) lacks the $\sqrt{\log(\frac{1}{\tau})}$ factor. This omission arises due to a mistake in their Theorem 2, which relies on a covering-number argument designed for unconstrained RL (Lemma D.6 in Jin et al. (2020)), an approach that is not valid in our setting under instantaneous hard constraints. To clarify, note that a $\kappa$-covering of the value functions ensures that every possible $V_h^k$ can be approximated to within $\kappa$. For the unconstrained setting, Jin et al. (2020) demonstrated that a $\kappa$-covering of the $Q_h^k$-functions directly induces a $\kappa$-covering of the $V_h^k$-functions. This is because $V_h^k$ is obtained by maximizing $Q_h^k$ over a fixed action set $\mathcal{A}$, and the contraction property of the max operator ensures that small changes in $Q_h^k$ lead to small changes in $V_h^k$ (Ghosh et al., 2022b). In our setting, however, $V_h^k$ is defined by maximizing $Q_h^k$ over a *data-dependent* safe set $\mathcal{A}_h^k$. Consequently, a covering of the $Q_h^k$-functions does not directly imply a covering of the $V_h^k$-functions, as changes in $\mathcal{A}_h^k$ can cause significant differences in $V_h^k$. To resolve this issue, we develop a novel technique called OCD for bounding the covering number in the star-convex setting (see Section 6). Specifically, we show that when $\mathcal{F}_s$ is star-convex, the changes in $\mathcal{A}_h^k$ can be controlled by variations in the safety parameters $\gamma_h^k$ and $\Lambda_h^{k,\gamma}$, enabling an effective bound on the covering number. Once this bound is established, we provide a corrected version of Theorem 2 in Amani et al. (2021) for the event $\mathcal{E}_2$, which now includes the missing $\sqrt{\log(\frac{1}{\tau})}$ term, as detailed below:

**Theorem 5.2** (Corrected covering-number result)**.** *Under the setting of Theorem 5.1, for any fixed policy*

$\pi$, let the event $\mathcal{E}_2$ be defined as follows: $\mathcal{E}_2 \triangleq$
$\{ \left| \langle \mathbf{w}_h^k, \phi(s,a) \rangle - Q_h^\pi(s,a) + \mathbb{P}_h \left( V_{h+1}^\pi - V_{h+1}^k \right)(s,a) \right|$
$\leq \beta \| \phi(s,a) \|_{(\mathcal{A}_h^{k,Q})^{-1}}, \forall (a,s,h,k) \in \mathcal{A} \times \mathcal{S} \times [H] \times [K] \},$

where $\beta_1 = c_\beta dH \sqrt{ log(\frac{dK}{\tau\delta}) }$ for some constant $c_\beta$. Then, the event $\mathcal{E}_2$ holds with the probability of at least $1 - \delta$.

## 5.2. The Challenge of Covering Numbers in Non-star-Convex Safe RL

Now, we can illustrate why a dedicated pure exploration phase is critical for controlling the covering number when $\mathcal{F}_s$ is not star-convex. Suppose we remove the pure-exploration component from Algorithm 1 and start the second phase of the algorithm, *safe-exploration-exploitation*. The immediate issue, in contrast star-convex settings, is that we would require a large $\kappa$-covering of the value functions. The key distinction between these two settings lies in the inability to control changes in $\mathcal{A}_h^k$ through variations in the safety parameters $\gamma_h^k$ and $\Lambda_h^{k,\gamma}$. In non-star-convex settings, even small variations in these safety parameters can drastically alter $\mathcal{A}_h^k$, leading to substantial shifts in $V_h^k$. Thus, we cannot bound the covering number in the same way as in star-convex settings. In fact, the following formalizes this phenomenon in a simple one-dimensional example, showing that the covering number can grow at least as large as the number of states, which can be prohibitively large in environments with very large or continuous state spaces.

**Lemma 5.3.** *(Covering number in non-star-convex settings) Consider a one-dimensional setting where $Q(s,a) = a$ for $a \in \mathbb{R}$, with a finite state space $\mathbb{S} = \bigcup_{i=1}^{|\mathcal{S}|} \{i\} \subset \mathbb{N}$. Define the class of parameterized functions $\mathcal{V} \triangleq \{ V_\gamma \mid \|\gamma\| \leq 1 \}$, where $V_\gamma(s) \triangleq \max_{a \in \mathcal{A}_\gamma(s)} Q(s,a)$, and $\mathcal{A}_\gamma(s) \triangleq \{ a \mid a \in [0, \frac{1}{3}] \cup [\frac{2}{3}, 1], \gamma \cdot s \cdot a \leq \tau \}$. Now, for an arbitrary positive real number $\kappa < \frac{1}{6}$, let the set $\mathcal{V}_\kappa \subset \mathcal{V}$ be a $\kappa$-covering for the function class $\mathcal{V}$. Then, the following inequality holds: $|\mathcal{S}| \leq |\mathcal{V}_\kappa|$.*

Despite the fundamental challenge highlighted in Lemma 5.3, we show that under the Local Point Assumption 3.2, the pure exploration phase in Algorithm 1 enables us to control the covering number during the second phase of Algorithm 1. This is because, by the end of the pure exploration phase, the agent's estimated safe set remains stable under small variations in the safety parameter with high probability. Below, we present our main result:

**Theorem 5.4.** *(Regret under Local Point Assumption) Under assumptions 2.1, 2.3, and 3.2, there exists a constant $c_\beta > 0$ such that for any $\delta \in (0, \frac{1}{3})$, by setting $\beta_1 = c_\beta dH \sqrt{ log(\frac{dK}{\tau\delta}) }$ and $\nu = \frac{2}{\tau}$ and $\lambda = 1$, with the probability of at least $1 - 3\delta$ Algortihm 1 remains safe, $\mathcal{A}_h^k(s) \subset \mathcal{A}_h^{safe}(s), \quad \forall (h,k) \in [H] \times [K]$. Moreover, with probability at least $1 - 3\delta$, Algorithm 1 achieves*

$Regret(K) \leq K'H + 2H\sqrt{ (K-K')H\ log(\frac{d(K-K')H}{\delta}) } + 2(\beta_1 H + \frac{\beta_2 H^2}{\tau}) H \sqrt{ 2(K-K')d \log((\frac{d\lambda + 2KL^2}{\lambda d})) }, for all$

$K' \geq \max\{ \frac{8d}{\epsilon^2} \log(\frac{dH}{\delta}), \frac{2d}{\epsilon^2}(\frac{16\beta_2^2}{\iota^2} - \lambda) \}$, where $\epsilon$ and $\iota$ are defined in Assumption 3.2.

**Comparison with Star-Convex Case.** The regret dependence on $K$ in Theorem 5.4 is $\tilde{O}(\sqrt{K})$. The key distinction between Theorems 5.4 (non-star-convex) and 5.1 (star-convex) arises due to the necessary pure exploration phase in the non-star-convex setting. Specifically, while Theorem 5.1 achieves a regret of $\tilde{O}(\sqrt{K})$ without an explicit exploration phase ($K' = 0$), Theorem 5.4 introduces a pure exploration phase with length $K' = O(\frac{\log(K)}{\epsilon^2 \iota^2})$, which stems from the Local Point Assumption and reflects the added complexity of non-star-convex settings compared to star-convex ones. This term also appears explicitly in the regret bound, capturing the cost of ensuring safe exploration in such environments.

## 6. Outline of the Proof

The proof steps presented here hold for both Theorem 5.4 and Theorem 5.1, albeit with different choices of $K'$. For Theorem 5.4, $K'$ corresponds to the termination of the pure exploration phase, while for Theorem 5.1, $K'$ can be set to 0 as no pure exploration phase is needed in the star-convex case. Our proof involves three main steps: (1) decomposing the sum of regret and ensuring the constraint satisfaction, (2) bounding the covering number of the value function class, and (3) applying uniform concentration tools to achieve sublinear regret. The key challenge lies in controlling the covering number of the value function class, which critically depends on how the geometry of the decision space.

**Step 1: Decomposition.** Following a similar approach as in Ghosh et al. (2022b), we first establish a decomposition that upper bounds the sum of regret: $Regret(K) \leq K'H +$

$$\underbrace{\sum_{k=K'}^{K} \left( V_0^{\pi^*}(s_0) - V_0^k(s_0) \right)}_{\mathcal{T}_1} + \underbrace{\sum_{k=K'}^{K} \left( V_0^k(s_0) - V_0^{\pi^k}(s_0) \right)}_{\mathcal{T}_2}.$$

To control $\mathcal{T}_1$ and $\mathcal{T}_2$, we need to carefully compare the estimated value functions $V_h^k$ produced by our algorithm at each episode $k$ with the corresponding value functions $V_h^\pi$ induced by a given policy $\pi$. A critical challenge is dealing with the adaptive nature of the learning process, where the chosen actions depend on previously observed transitions and rewards. This adaptivity can make standard self-normalized concentration inequalities inapplicable, as the future sample distribution depends on the current value estimates. To address this, we rely on value-aware uniform concentration arguments. By fixing a suitable function class $\mathcal{V}$ of candidate value functions in advance and ensuring

that each $V_h^k$ belongs to $\mathcal{V}$, we leverage the polynomial log-covering number of $\mathcal{V}$ to obtain high-probability bounds uniformly over all functions in the class. Concretely, for each $(k, h) \in [K] \times [H]$, we aim to show that, with high probability:

$$\left\| \sum_{\tau=1}^{k-1} \phi(s_h^\tau, a_h^\tau) \left[ V_{h+1}^k(s_{h+1}^\tau) - \mathbb{P}_h V_{h+1}^k(s_h^\tau, a_h^\tau) \right] \right\|_{(\Lambda_h^k)^{-1}}$$
$$\leq O(d\sqrt{\log K}), \tag{5}$$

where $\mathbb{P}_h$ is the transition operator at stage $h$. Achieving such a bound ensures that our estimated value functions are stable and close to the true policy-based value functions, facilitating the desired sublinear regret guarantees.

**Step 2: Controlling the Covering Number.** We begin by defining our class of value functions $\mathcal{V}$. Consider parameters $\theta, \gamma \in \mathbb{R}^d$ and positive semi-definite matrices $A, A'$, with bounded norms (e.g., $\|\theta\| \leq \sqrt{d}$, $\|\gamma\| \leq \sqrt{d}$, $\|A\|_F \leq \frac{\sqrt{d}B^2}{\lambda}$, and $\|A'\|_F \leq \frac{\sqrt{d}(B')^2}{\lambda}$). Given these parameters, the value function can be written as $V(s) = \min\{V'(s), H\}$, where:

$$V'(s) = \max_{a \in \mathcal{A}} \langle \phi(s, a), w \rangle + \|\phi(s, a)\|_A + g_{A', 1}(s, a),$$

s.t. $\langle \phi(s, a) - \phi(s, a_s^0), \gamma \rangle + \|\phi(s, a) - \phi(s, a_s^0)\|_{A'} \leq \tau,$

where $g_{A', 1}(s, a) \triangleq \frac{2}{\tau} \|\phi(s, a) - \phi(s, a_s^0)\|_{A'}$. We collect all such value functions into the following class: $\mathcal{V} \triangleq \{V_{w, \gamma, A, A'} \mid \|w\| \leq L_w, \|\gamma\| \leq \sqrt{d}, \|A\|_F \leq \frac{\sqrt{d}B^2}{\lambda}, \|A'\|_F \leq \frac{\sqrt{d}(B')^2}{\lambda}\}$. In contrast to unconstrained RL, where the value function $V(.)$ is determined by maximizing the objective over the entire action space $\mathcal{A}$, our setup in Eq. (6) introduces constraints through the parameters $(\gamma, A')$, which directly affect the feasible decision set. This distinction highlights the need to consider both the objective and the constraints when constructing a log-polynomial-sized covering for $\mathcal{V}$, as we discuss next.

**OCD: Objective–Constraint Decomposition.** Here we introduce our novel idea for bounding the covering number. Consider two arbitrary value functions $V_1 = \min\{V_1'(s), H\}$ and $V_2 = \{V_2'(s), H\}$ from our class $\mathcal{V}$, where:

$$V_i'(s) = \max_{a \in \mathcal{A}} \langle \phi(s, a), w_i \rangle + \|\phi(s, a)\|_{A_i} + g_{A_i', 1}(s, a)$$

s.t. $\langle \phi(s, a) - \phi(s, a_s^0), \gamma_i \rangle + \|\phi(s, a) - \phi(s, a_s^0)\|_{A_i'} \leq \tau,$ (6)

for all $i \in \{1, 2\}$. To relate $V_1$ and $V_2$, we introduce an intermediate value function $V_3$ with objective parameters from $V_1$ and constraint parameters from $V_2$. Specifically,

$V_3(s) = \min\{V_3'(s), H\}$, where:

$$V_3'(s) = \max_{a \in \mathcal{A}} \langle \phi(s, a), w_1 \rangle + \|\phi(s, a)\|_{A_1} + g_{A_1', 1}(s, a)$$

s.t. $\langle \phi(s, a) - \phi(s, a_s^0), \gamma_2 \rangle + \|\phi(s, a) - \phi(s, a_s^0)\|_{A_2'} \leq \tau.$
(7)

Now, we have: $|V_1(s) - V_2(s)| \leq |V_1(s) - V_3(s)| + |V_2(s) - V_3(s)|$. This inequality represents a *decomposition of the distance between $V_1$ and $V_2$ into contributions from differences in objectives and constraints*. Specifically, $V_1$ and $V_3$ differ only in their objective parameters, while $V_2$ and $V_3$ differ in their constraints. Now, since $V_2$ and $V_3$ share the same constraint parameters $(\gamma_2, A_2')$, they are maximizing over the same feasible action set. Thus, bounding $|V_2 - V_3|$ reduces to a scenario akin to the unconstrained case, where comparing two linear–quadratic forms over the same domain is straightforward. On the other hand, bounding $|V_1 - V_3|$ is *more challenging* because $V_1$ and $V_3$ have *different feasible decision sets*.

**Bounding $|V_1 - V_3|$ in Star-Convex Setting.** When $\mathcal{F}_s$ is star-convex, we prove that small perturbations in $\gamma$ or $A'$ do not cause large, discontinuous changes in the feasible action set (see Lemma B.7 in the Appendix). Specifically, we show that if $a_1$ is feasible for $V_1$, then there exists a scalar $\alpha_3 \in \left[ \frac{\tau}{\tau + \Delta}, 1 \right]$ and an action $a_3$ feasible for $V_3$ such that $\phi(s, a_3) = \alpha_3 \phi(s, a_1) + (1 - \alpha_3) \phi(s, a_s^0)$, where $\Delta = \|\gamma_2 - \gamma_1\| + \sqrt{\|A_1' - A_2'\|_F}$. As $\Delta$ approaches zero, $\phi(s, a_3)$ converges to $\phi(s, a_1)$, implying that every feasible feature of $V_1$ has a close counterpart in $V_3$, and vice versa, with the difference controlled by $\Delta$. Consequently, $|V_1 - V_3|$ can also be controlled by $\Delta$. Hence, constructing a polynomial-size cover of the parameter space induces a corresponding cover for the value function class $\mathcal{V}$.

*Remark* 6.1. When $\tau$ is small, a smaller $\Delta$ is required to ensure that $\alpha_3 \in [\frac{\tau}{\tau + \Delta}, 1]$ remains close to 1, which in turn ensures that $\phi(s, a_3)$ stays sufficiently close to $\phi(s, a_1)$. As a result, a larger covering net is needed to account for smaller variations in the parameter space. This additional complexity is reflected in the $\sqrt{\log(\frac{1}{\tau})}$ term in Theorem 5.2.

**Non-Star-Convex setting.** In non-star-convex settings, our OCD technique cannot be directly applied without a pure exploration phase, as small changes in the safety parameters $\gamma$ or $A'$ can cause drastic shifts in the feasible action sets, leading to significant differences in $V_1$ and $V_3$ (see Lemma 5.2). To understand why this occurs in non-star-convex settings, note that when $\mathcal{F}_s$ is not star-convex, we cannot guarantee that $\alpha \phi(s, a_1^*) + (1 - \alpha) \phi(s, a_s^0) \in \mathcal{F}_s$ for $\alpha \in [\frac{\tau}{\tau + \Delta}, 1]$. This is the main reason why our approach for the star-convex case does not extend directly to the non-star-convex setting. However, once $V_1 = V_h^k$, where $V_h^k$ is the value function generated by Algorithm 1 after the pure exploration phase, we show that when $\Delta$

is sufficiently small, with high probability, all actions in $\mathcal{A}_h^\iota(s) = \{a \in \mathcal{A} \mid \langle \phi(s,a), \gamma_h^* \rangle \leq \tau - \iota\}$ are feasible for both $V_1$ and $V_3$. Moreover, under the Local Point Assumption, any feasible action of $V_1$ outside $\mathcal{A}_h^\iota(s)$ has a close feasible counterpart in $V_3$ within a $\Delta$-neighborhood, and vice versa (see Lemma B.3 in the Appendix). This closeness of the feasible sets of $V_1$ and $V_3$ enables us to reapply the ideas underlying OCD and effectively bound the covering number in the second phase of Algorithm 1.

**Step 3: Final assembly.** With these log-polynomial bound for the covering number in hand—achieved, we can apply standard self-normalized concentration inequalities. Thus, we can prove the following Lemmas:

**Lemma 6.2.** *(Optimism) With probability of at least $1 - \frac{\delta}{2}$, we have:* $\mathcal{T}_1 \leq 0$.

**Lemma 6.3.** *(Bounding $\mathcal{T}_2$) With the probability of at least $1 - \frac{\delta}{2}$, $\mathcal{T}_2 \leq \sum_{k=K'}^{K} \sum_{h=1}^{H} \zeta_h^k + \sum_{k=K'}^{K} \sum_{h=1}^{H} 2\beta_1 \|\phi(s_h^k, a_h^k)\|_{(\Lambda_h^{k,Q})^{-1}} + g_h^k(s_h^k, a_h^k)$, where $\zeta_{h+1}^k \triangleq \mathbb{P}_h(V_{h+1}^k - V_{h+1}^{\pi^k})(s_h^k, a_h^k) - \delta_{h+1}^k$ and $\delta_{h+1}^k \triangleq V_h^k(s_h^k) - V_h^{\pi^k}(s_h^k)$.*

Now, combining the last two steps and upper bounding the normalized term obtained in step 3, we can apply Lemma 11 from Abbasi-Yadkori et al. (2011) to derive the final result. Additionally, Theorem 2 from Abbasi-Yadkori et al. (2011) can be applied to show that our approach ensures high-probability safety.

# 7. Experiment

We consider an autonomous-vehicle path-planning task in a merging scenario (Figure 2) where the goal is to navigate safely and efficiently (see Section 3 for details). A trained module (**Collav**) handles collision avoidance by masking infeasible actions, so the RL agent focuses on satisfying lane-keeping constraints. Given an initial safe policy $\pi^{\text{safe}}$, the feature space is non-star-convex due to **Collav** but still satisfies the Local Point Assumption. Our objective is to learn an optimal safe policy with sublinear regret. We implement **NCS-LSVI** (Algorithm 1) with different values of $K'$ and report the resulting regret in Figure 3. For $K' = 2000$, regret remains sublinear, indicating successful learning of the optimal safe policy after sufficient pure exploration. Moreover, across all $K$, our algorithm never violates the safety constraint, ensuring lane-keeping is always satisfied. See Appendix A for further details and additional baseline comparisons.

# 8. Conclusion

In this paper, we studied Safe RL with instantaneous hard constraints in both star-convex and non-star-convex settings.

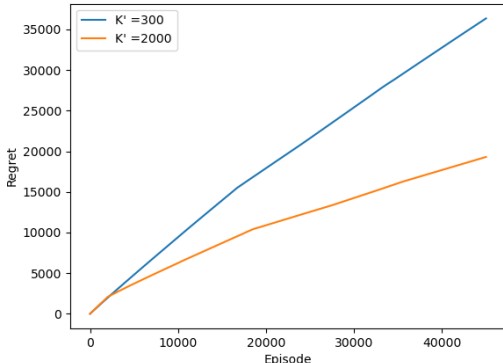

*Figure 3.* Regret vs. episodes for NCS-LSVI in an autonomous vehicle merging scenario.

A key challenge in these settings is bounding the covering number of the value-function class, which is critical for achieving value-aware uniform concentration in model-free function approximation. For the star-convex setting, we introduced a novel technique, OCD, to effectively bound the covering number. This result also resolves an error in previous work on constrained RL. For non-star-convex scenarios, we proposed a new two-phase algorithm, NCS-LSVI, which effectively addresses the challenge of bounding the covering number in these settings. Our analysis demonstrates that our methods attain $\tilde{\mathcal{O}}(\sqrt{K})$ regret with zero safety violations in both the star-convex and non-star-convex settings. Numerical simulations on an autonomous driving scenario demonstrated the practical effectiveness of our method. While our regret bound for the non-star-convex setting under the Local Point Assumption includes a dependence of order $\frac{1}{\epsilon^2 \iota^2}$, it remains an open question whether this dependence is fundamental, and resolving it is an important direction for future work. Additionally, future research will focus on moving beyond linear-MDP structures, relaxing assumptions such as the local-point condition, and designing algorithms capable of maintaining safety and near-optimal regret in more complex, nonlinear environments.

# Acknowledgments

This work has been supported in part by the U.S. National Science Foundation under the grants: NSF AI Institute (AIEDGE) 2112471, 2312836, 2106933, NJIT start-up fund index number 172884, Army Research Office W911NF-24-2-0205, and was sponsored by the Army Research Laboratory under Cooperative Agreement Number W911NF-23-2-0225. The views and conclusions contained in this document are those of the authors and should not be interpreted as representing the official policies, either expressed

or implied, of the Army Research Laboratory or the U.S. Government. The U.S. Government is authorized to reproduce and distribute reprints for Government purposes notwithstanding any copyright notation herein. The authors are also grateful to Dr. Xingyu Zhou for pointing out an error in the analysis by Amani et al. (2021).

## Impact Statement

This paper contributes to the advancement of Machine Learning and Learning Theory by addressing fundamental technical challenges. While the societal impacts of our work align with those typically associated with progress in these fields, we do not foresee any immediate or unique ethical concerns that require specific attention in this context.

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

The appendix is organized as follows. Appendix A contains the simulation details presented in the main part of the paper. Appendix B includes the proofs for bounding the covering number in both star-convex and non-star-convex settings, as stated in Theorems 5.4 and 5.1. Appendix C contains the detailed proof steps for Theorem 5.4 in the non-star-convex setting. Finally, Appendix D provides the proof steps for Theorem 5.1 in the star-convex setting.

## A. Experiment Details and Star-Convex Simulations

### A.1. Detailed Setup for Section 7 Experiments

**Problem Statement** We consider a merging scenario at an intersection where an autonomous vehicle must navigate safely and efficiently. The intersection consists of a main road with another vehicle already traveling on it and a merging lane where our autonomous vehicle starts. The autonomous vehicle needs to decide whether to wait for the car on the main road to pass or to accelerate and merge into the main road before the other car arrives. Please see Figure 2 for a sketch of the scenario. The objective of the autonomous vehicle is to maximize the traversed path toward a goal point within a finite time horizon while adhering to the following constraints: 1. Avoid collisions with the car on the main road. 2. Keep the vehicle within the lane boundaries. In our experiment we assume that, the car includes a trained collision avoidance module (referred to as **Collav**), which identifies infeasible actions that could lead to collisions, threfore the RL agent does not need to learn collision avoidance constraint. However, lane-keeping is a constraint that the RL agent must learn during its interaction with the environment. Please see Figure 5 for a brief diagram of our problem.

Accelerating to merge into the main road is the optimal choice in this scenario, as it allows the vehicle to traverse a greater distance toward the goal point within the finite time horizon. However, due to the unknown dynamics of the car, the RL agent must adopt a conservative strategy, denoted as $\pi^{\text{safe}}$, to avoid losing control when accelerating.

**Vehicle Dynamics** The vehicle's dynamics are governed by the following equations:

$$\begin{bmatrix} x_{t+1} \\ y_{t+1} \\ v_{t+1}^x \\ v_{t+1}^y \end{bmatrix} = \begin{bmatrix} x_t + v_{t+1}^x \\ y_t + v_{t+1}^y \\ f(v_t^x + \alpha_1 u_t^x) \\ f(v_t^y + \alpha_2 u_t^y) \end{bmatrix}, \tag{8}$$

where $x_t, y_t$ represent the vehicle's position, and $v_t^x, v_t^y$ are its velocities. The control inputs $u_t^x, u_t^y$ affect the velocity through unknown parameters $\alpha_1, \alpha_2$. The function $f(\cdot)$ models nonlinear dynamics, which are linear in most regions but saturate at the car's speed limits as illustrated in Figure 4. For simulation purposes, we set $(\alpha_1, \alpha_2) = (0.5, 0.5)$.

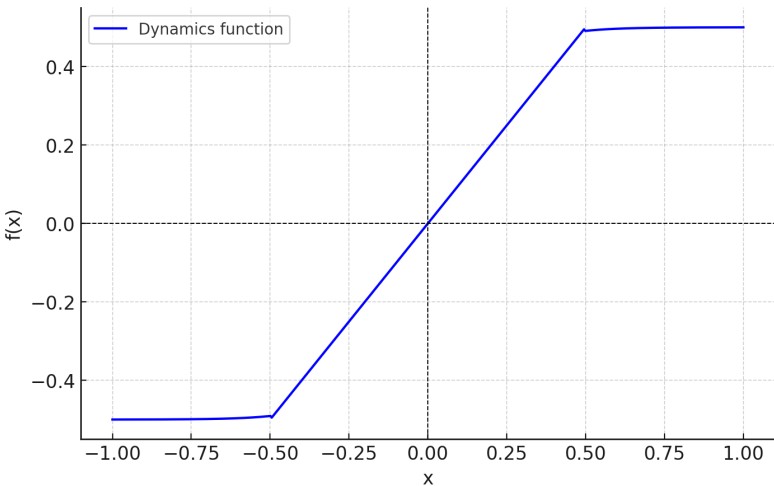

*Figure 4.* Plot of $f(x)$ showing the dynamics function behavior.

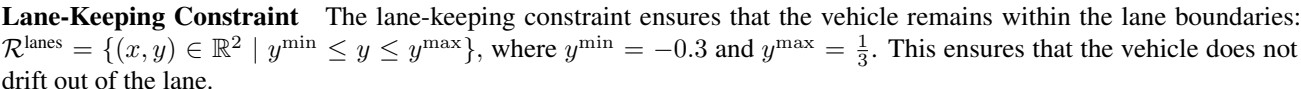

**Lane-Keeping Constraint** The lane-keeping constraint ensures that the vehicle remains within the lane boundaries: $\mathcal{R}^{\text{lanes}} = \{(x, y) \in \mathbb{R}^2 \mid y^{\min} \leq y \leq y^{\max}\}$, where $y^{\min} = -0.3$ and $y^{\max} = \frac{1}{3}$. This ensures that the vehicle does not drift out of the lane.

**Collision Avoidance** The collision avoidance module **Collav** restricts unsafe actions at the start of each episode ($h = 0$). Specifically, actions that result in a speed within $\frac{1}{16} \leq |v_t^x + v_t^y| \leq \frac{1}{4}$ are prohibited. For all other timesteps ($h \in [H]$), the module does not impose any restrictions. At $h = 0$, the resulting restricted action space is shown in Figure 2. The time horizon for the simulation is $H = 3$.

**Initial Safe Policy.** The initial safe policy, $\pi^{\text{safe}}$, is defined as: $\pi^{\text{safe}}(s_t) = \left[ \frac{v^{\text{ref}} - v_t^x}{\kappa_1}, \frac{v^{\text{ref}} - v_t^y}{\kappa_2} \right]^\top$, where $v^{\text{ref}} = 0.001$ is a small conservative reference speed, and $\kappa_1 = 100000$ and $\kappa_2 = 0.5$ determine the reaction strength to deviations.

**Algorithm Implementation** We implement Algorithm 1 for this setup using the feature vector: $\phi(s, a) = [x, y, v^x, v^y, u^x, u^y]^\top$, and express the lane-keeping constraint as: $y^{\min} \leq \langle \phi(s, a), \gamma^* \rangle \leq y^{\max}$, where $\gamma^*$ is an unknown parameter vector. The total number of episodes is set to $K = 1000$, with $\epsilon = 0.1$ chosen to satisfy Assumption 3.2. The pure exploration parameter $K'$ is selected from $\{1, 300, 2000\}$. We discretize the action space into grids of size $\frac{1}{8} \times \frac{1}{8}$ to address optimization challenges in the non-convex setup. However, our state space is continueous. Moreover, in order to speed-up our Algorithm, we update the $Q$-function's parameter ($w_h^k$) every $2^k$ epochs.

**Results** Figure 3 demonstrates that after an initial exploration phase, the agent converges to the optimal policy, achieving sublinear regret. Additionally, Figure 3 shows that the $y_t$ values remain within the lane boundaries throughout both exploration and exploitation phases.

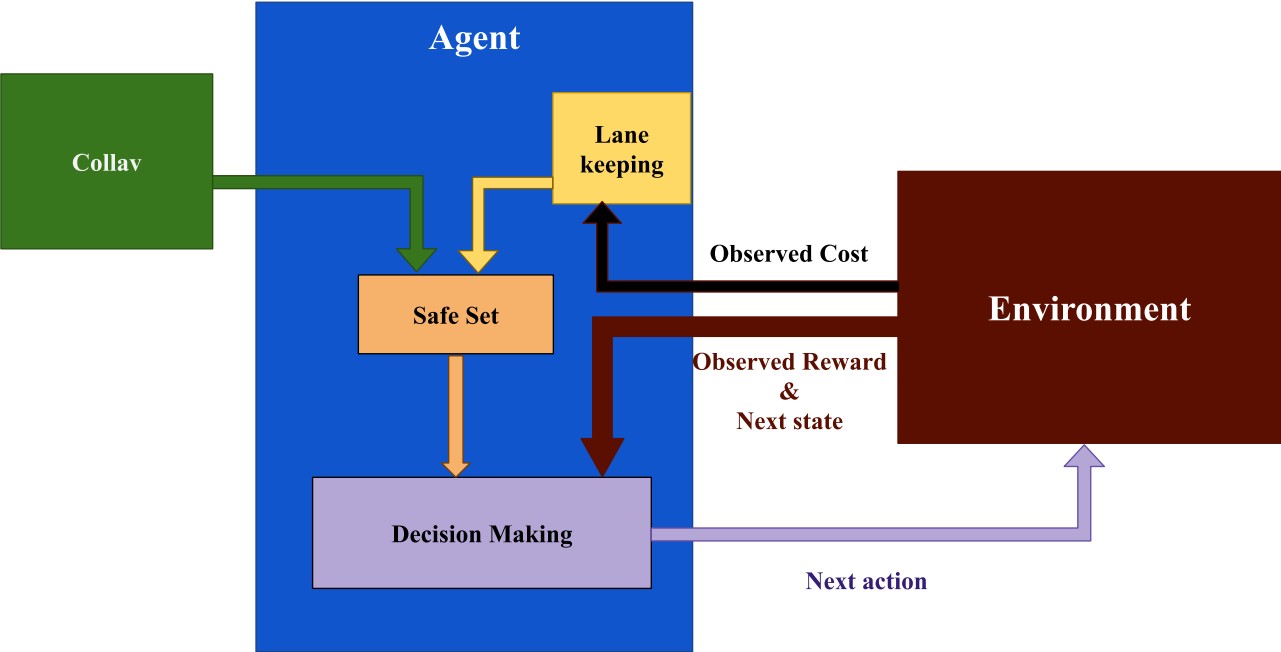

*Figure 5.* The diagram of autonomous vehicle example: Agent interacts with the environment and observe feedbacks on its location and speed. It utilizes the feedback to imporove the estimation of lane keeping. Then, using lane keeping and a trained collav module it provides the safe set of actions. The decision making module uses the feedback to enhance the estimation on $Q$ function, and then utilizes the saf set to make the next decision. Note that the Collav block is trained a prioir and we are not learning it, but lane keeping and Decision Making are the blocks that RL agent needs to learn.

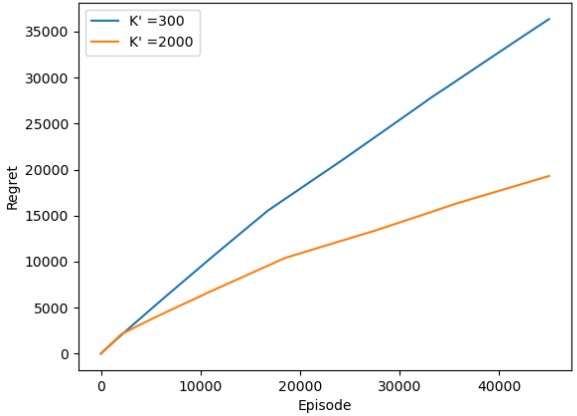

*Figure 6.* Regret of our method (NCS-LSVI) over more episodes, showing sublinear behavior for $K' = 2000$.

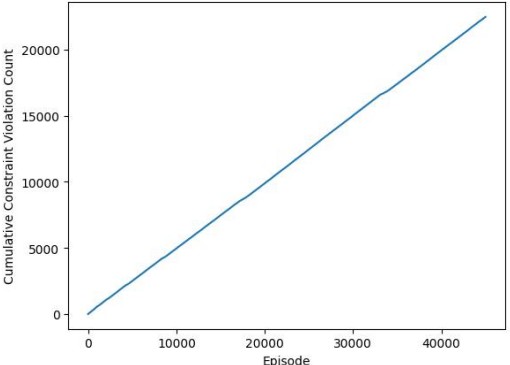

*Figure 7.* Regret of a sub-optimal but safe baseline that stays within an $\epsilon$-neighborhood of the initial safe policy.

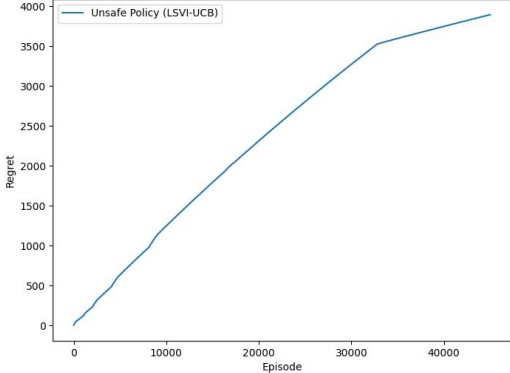
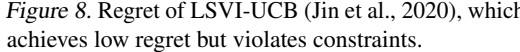

*Figure 8.* Regret of LSVI-UCB (Jin et al., 2020), which achieves low regret but violates constraints.

*Figure 9.* Cumulative constraint violations for LSVI-UCB, showing linear growth. The other two methods have zero violations, so no violation plots are included.

### A.2. Baseline comparisons.

Figure 6 shows the regret of NCS-LSVI over more episodes, demonstrating sublinear growth for $K' = 2000$. Figure 7 shows the regret of a sub-optimal but safe baseline constrained to an $\epsilon$-neighborhood of the initial policy.

Figure 8 shows the regret of LSVI-UCB (Jin et al., 2020), which achieves lower regret but violates constraints, as shown in Figure 9, where cumulative violations grow linearly. The other two methods have zero violations, so no violation plots are included. We will add these results in the final version.

### A.3. Experiment in the Star-Convex Setting

Based on Lemma 5.3, we observe that in non-star-convex settings, without a pure exploration phase, the covering number is lower bounded by the cardinality of the state space. However, Theorem 5.1 establishes that in star-convex settings, the covering number can be properly bounded without a pure exploration phase, ensuring sublinear regret. To validate this argument, we conduct numerical experiments in our autonomous vehicle setting, where the state space is continuous and has infinite cardinality.

We retain the same setup as in the non-star-convex scenario but remove the collision avoidance module, making the decision space star-convex. Specifically, we assume that the red car in Figure 2 is no longer present. According to Theorem 5.1, we expect that skipping the pure exploration phase in Algorithm 1 (i.e., setting $K' = 0$) will still allow the RL agent to achieve sublinear regret. The regret graph in Figure 10 confirms this expectation, showing that Algorithm 1 successfully attains sublinear regret. Moreover, the agent consistently satisfies the safety constraint, never deviating from its lane throughout learning.

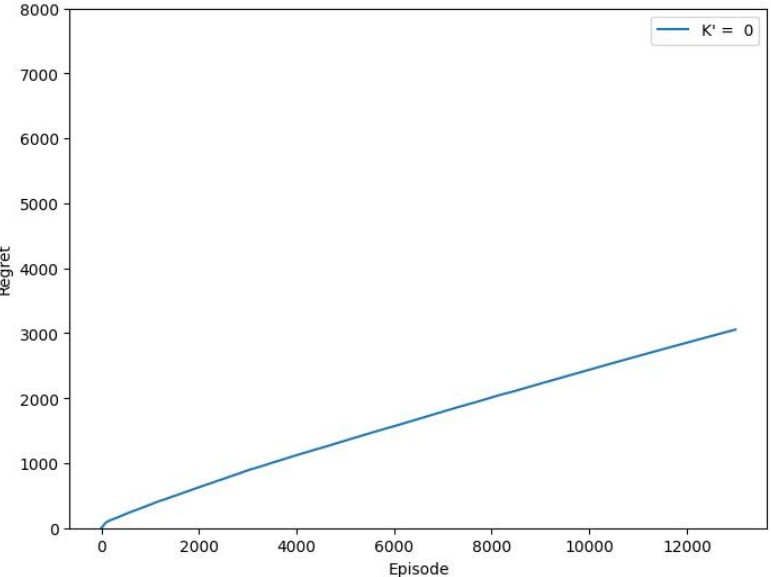

*Figure 10.* Regret vs. episodes for NCS-LSVI in an star-convex autonomous vehicle merging scenario.

### A.4. Effectiveness of Linear MDPs in Practice

Linear MDPs are widely studied and effective in practice. Zhang et al. (2022) demonstrate state-of-the-art performance on MuJoCo and DeepMind Control benchmarks, and Jin et al. (2020) show that linear MDP solutions offer regret guarantees even when the true MDP is nonlinear. Also, a safe sub-optimal policy can often be identified offline using domain knowledge (Amani et al., 2019; Khezeli & Bitar, 2020; Shi et al., 2023).

## B. Bounding Covering Number in Theorems 5.4 and 5.1

In Appendix B.1, we present our results for bounding the covering number of the function class of individual value functions in the non-star-convex setting defined in Theorem 5.4. Similarly, Appendix B.2 contains our results for the star-convex setting defined in Theorem 5.1. Appendix B.3 provide our proof for Lemma 5.3. Additionally, Appendix B.4 provides proofs for auxiliary lemmas used in the previous sections.

Before presenting the proof, we define the function class of value functions used in our work below.

**Definition B.1.** Let $\mathcal{V}$ denotes a class of functions, $\min\{V(.), H\}$, where $V(.)$ is mapping from $\mathcal{S}$ to $\mathbb{R}$ with the following parametric form:

$$V(s) \triangleq \min\{\max_{a \in \mathcal{A}}(\langle\phi(s,a), w\rangle + \beta\|\phi(s,a)\|_{\Lambda^{-1}}) + g_{\Lambda^{-1}}(s,a), \ H\}$$
$$s.t : \langle\phi(s,a) - \phi(s,a_s^0), \gamma\rangle + \beta'\|\phi(s,a) - \phi(s,a_s^0)\|_{(\Lambda')^{-1}} \leq \tau, \tag{9}$$

where the parameters $(w, \gamma, \beta, \beta', \Lambda)$ satisfy $\|w\| \leq L_w$, $\|\gamma\| \leq \sqrt{d}$, $\lambda_{\min}(\Lambda) \geq \lambda$, $\beta \in [0, B]$, and $\beta' \in [0, B']$. Also, $g_{\Lambda^{-1}, \beta'}$ is defined as follows:

$$g_{\Lambda^{-1}, \beta'}(s,a) \triangleq \frac{2}{\tau}\beta'\|\phi(s,a) - \phi(s,a_s^0)\|_{\Lambda^{-1}} \tag{10}$$

.

### B.1. Covering Number in Linear MDPs with Instantaneous Hard Constraints under Local Point Assumption

Now we state the next Lemma that provide a proper upper bound for $\mathcal{N}_\kappa$:

**Lemma B.2.** *(Covering number in linear MDPs with instantaneous hard constraints under Local Point Assumption). Consider the setting of Theorem 5.4, and let $\kappa \leq \frac{\iota}{2}$. Then there exists a finite set of functions $\mathcal{V}_\kappa \subset \mathcal{V}$ such that for all $k \geq K'$, there exists a $V \in \mathcal{V}_\kappa$ such that $dist(V, V_h^k) \leq \kappa$. Moreover, let $\mathcal{N}_\kappa$ be the cardinality for $\mathcal{V}_\kappa$, then with probability of at least $1 - \delta$, we will have the following:*

$$\log(\mathcal{N}_\kappa) \leq d\left[\log(1 + \frac{4L_w}{\kappa}) + \log(1 + \frac{8(\frac{d^{\frac{1}{4}}B}{\sqrt{\lambda}} + H)}{\kappa\tau})\right]$$
$$+ d^2\left[\log(1 + \frac{32\sqrt{d}B^2}{\lambda\kappa^2}) + \log(1 + \frac{32\sqrt{d}(B')^2(\frac{d^{\frac{1}{4}}B}{\sqrt{\lambda}} + 3H)^2}{\lambda\tau^2\kappa^2})\right]. \tag{11}$$

**Proof:** We equivalently define $A = \beta^2\Lambda^{-1}$ and $A' = \beta'^2\Lambda^{-1}$ and reparameterize the class function $\mathcal{V}$ as follows:

$$V(.) = \min\{\max_{a \in \mathcal{A}}\langle\phi(.,a), w\rangle + \|\phi(.,a)\|_A + g_{A',1}(.,a), \ H\}$$
$$s.t : \langle\phi(.,a) - \phi(.,a_.^0), \gamma\rangle + \|\phi(.,a) - \phi(.,a_s^0)\|_{A'} \leq \tau, \tag{12}$$

where $\|A\| \leq \frac{B^2}{\lambda}$ and $\|A'\| \leq \frac{(B')^2}{\lambda}$.

Now, for $k \geq K'$, let $V_1 = V_h^k$ be the value function generated by Algorithm 1 during the exploration-exploitation phase (secon phase), and $V_2$ is an arbitrary function in $\mathcal{V}$. Then, there exist parameters $(w_1, \gamma_1, A_1, A_1')$ and $(w_2, \gamma_2, A_2, A_2')$ such

that:

$$V_1(s) = \min\{\max_{a \in \mathcal{A}}\langle\phi(s,a), w_1\rangle + \|\phi(s,a)\|_{A_1} + g_{A'_1,1}(s,a), H\}$$
$$s.t : \langle\phi(s,a) - \phi(s,a_s^0), \gamma_1\rangle + \|\phi(s,a) - \phi(s,a_s^0)\|_{A'_1} \leq \tau$$

(13)

$$V_2(s) = \min\{\max_{a \in \mathcal{A}}\langle\phi(s,a), w_2\rangle + \|\phi(s,a)\|_{A_2}) + g_{A'_2,1}(s,a), H\}$$
$$s.t : \langle\phi(s,a) - \phi(s,a_s^0), \gamma_2\rangle + \|\phi(s,a) - \phi(s,a_s^0)\|_{A'_2} \leq \tau$$

Now, define $V_3$ as follows:

$$V_3(s) = \min\{\max_{a \in \mathcal{A}}\langle\phi(s,a), w_1\rangle + \|\phi(s,a)\|_{A_1}) + g_{A'_1,1}(s,a), H\}$$
$$s.t : \langle\phi(s,a) - \phi(s,a_s^0), \gamma_2\rangle + \|\phi(s,a) - \phi(s,a_s^0)\|_{A'_2}) \leq \tau$$

(14)

Now, using triangle inequlity, for all $s \in \mathcal{S}$, we have:

$$|V_1(s) - V_2(s)| \leq |V_1(s) - V_3(s)| + |V_2(s) - V_3(s)|$$

(15)

Now, applying the triangle inequality for all $s \in \mathcal{S}$, we have:

$$|V_1(s) - V_2(s)| \leq |V_1(s) - V_3(s)| + |V_2(s) - V_3(s)|.$$

To bound $|V_1(s) - V_2(s)|$, it suffices to separately bound $|V_1(s) - V_3(s)|$ and $|V_2(s) - V_3(s)|$. Notably, $V_2$ and $V_3$ share the same constraint parameters, resulting in identical feasible sets. Therefore, their difference can be bounded by the difference in their objective parameters. On the other hand, $V_1$ and $V_3$ have differing constraint parameters, leading to different feasible decision sets. In the following, we provide bounds for each of these terms.

**Bounding** $|V_1(s) - V_3(s)|$  Note that $V_1(s) = V_h^k$ after pure exploration phase of Algorithm 1, i.e., $k \geq K'$. Thus, we show that if the differnce between the constraint parameters of $V_1$ and $V_3$ are small enough then their feasible set also does not change drastically as explained in the following Lemma:

**Lemma B.3.** *(No dramatic changes in the estimated action set). Let $\Delta = \|\gamma_2 - \gamma_1\| + \sqrt{\|A'_1 - A'_2\|_F}$, where $\|\cdot\|_F$ denotes the Frobenius norm for matrices, and assume $\Delta \leq \frac{\iota}{2}$. For each $s \in \mathcal{S}$, let $\phi(s, a_1^*)$ represent the solution of the constrained optimization problem associated with $V_1 = V_h^k$ for $k \geq K'$. Then, with probability of at least $1 - \delta$, there exists $\alpha_3 \in [\frac{\tau}{\tau + \Delta}, 1]$ such that $\alpha_3\phi(s, a_1^*) + (1 - \alpha_3)\phi(s, a_s^0)$ is feasible for the constrained optimization problem associated with $V_3$. Specifically, there exists an action $a_3 \in \mathcal{A}$ such that $\phi(s, a_3) = \alpha_3\phi(s, a_1^*) + (1 - \alpha_3)\phi(s, a_s^0)$, and the following holds:*

$$\langle(\phi(s,a_3) - \phi(s,a_s^0), \gamma_2\rangle + \|\phi(s,a_3) - \phi(s,a_s^0)\|_{A'_2} \leq \tau$$

The proof of the above Lemma is provided in Section B.4.

Using the above lemma, we can immediately derive the following result:

**Lemma B.4.** *For each $s \in \mathcal{S}$, with the probability of at least $1 - \delta$, the following inequality is satisfied:*

$$V_1(s) - V_3(s) \leq \frac{\Delta}{\tau + \Delta}(H + \sqrt{\|A_1\|_F}).$$

The proof of this lemma can be found in Section B.4.

Now, simliar steps can be applied to get $V_3(s) - V_1(s) \leq \frac{\Delta}{\tau+\Delta}(H + \sqrt{\|A_1\|_F})$. Thus, we will have:

$$|V_1(s) - V_3(s)| \leq \frac{\Delta}{\tau + \Delta}\left(H + \sqrt{\|A_1\|_F}\right) \leq \frac{\Delta}{\tau}\left(H + \sqrt{\|A_1\|_F}\right),$$

(16)

where the last inequality obtained by the fact that $\frac{\Delta}{\tau+\Delta} \leq \frac{\Delta}{\tau}$. Now, substituiting $\Delta = \|\gamma_1 - \gamma_2\| + \sqrt{\|A'_1 - A'_2\|_F}$ in Eq.(16) we will have:

$$|V_1(s) - V_3(s)| \leq \left( H + \frac{d^{\frac{1}{4}} B}{\sqrt{\lambda}} \right) \frac{\|\gamma_1 - \gamma_2\| + \sqrt{\|A_1' - A_2'\|_F}}{\tau}, \tag{17}$$

where we used the fact that $\|A_1\|_F \leq \frac{\sqrt{d} B^2}{\sqrt{\lambda}}$.

**Bounding** $|V_2(s) - V_3(s)|$   To bound $|V_2(s) - V_3(s)|$, we follow a similar approach to Lemma **D**.6 in (Jin et al., 2020):

$$
\begin{aligned}
&|V_2(s) - V_3(s)| \\
&\leq \sup_{a \in \mathcal{A}} \Big| \langle \phi(s,a), w_1 \rangle + \|\phi(s,a)\|_{A_1} + g_{A_1',1}(s,a) \\
&\qquad\qquad - \langle \phi(s,a), w_2 \rangle - \|\phi(s,a)\|_{A_2} - g_{A_2',1}(s,a) \Big|
\end{aligned}
\tag{18}
$$

$$\leq \|w_1 - w_2\| + \sqrt{\|A_1 - A_2\|_F} + \sup_{a \in \mathcal{A}} |g_{A_1',1}(s,a) - g_{A_2',1}(s,a)|,$$

where the first inequality follows from the fact that $V_2$ and $V_3$ are maximized over the same action space (see Equations (13) and (14)), and the optimal solution to the constrained problem is upper bounded by the optimal solution to the unconstrained problem.

Now, it remained to bound $\sup_{a \in \mathcal{A}} |g_{A_1',1}(s,a) - g_{A_2',1}(s,a)|$. Thus, we state the following helpful Lemma:

**Lemma B.5.** *Given PSD matrices $A_1'$ and $A_2'$, the following inequity holds:*

$$\sup_{s,a} |g_{A_1',1}(s,a) - g_{A_2',1}(s,a)| \leq \frac{2H}{\tau} \sqrt{\|A_1' - A_2'\|_F}$$

**Proof:** The proof of the above Lemma is provided in Section B.4.

Now, by applying Lemma B.5 to Equation (18), we can bound $|V_2(s) - V_3(s)|$ as follows:

$$|V_2(s) - V_3(s)| \leq \|w_1 - w_2\| + \sqrt{\|A_1 - A_2\|_F} + \frac{2H}{\tau} \sqrt{\|A_1' - A_2'\|_F} \tag{19}$$

**Bounding** $|V_1(s) - V_2(s)|$**:**   Combining Eq. (17) and Eq. (19), with the proability of at least $1 - \delta$, we have:

$$
\begin{aligned}
|V_1(s) - V_2(s)| &\leq |V_1(s) - V_3(s)| + |V_2(s) - V_3(s)| \\
&\leq \left( H + \frac{d^{\frac{1}{4}} B}{\sqrt{\lambda}} \right) \frac{\|\gamma_1 - \gamma_2\| + \sqrt{\|A_1' - A_2'\|_F}}{\tau} + \|w_1 - w_2\| + \sqrt{\|A_1 - A_2\|_F} + \frac{2H}{\tau} \sqrt{\|A_1' - A_2'\|_F}
\end{aligned}
\tag{20}
$$

**Final step.**   Let $\mathcal{C}_w$ be the $\frac{\kappa}{4}$-cover for $\{w \in \mathbb{R}^d | \|w\| \leq L_w\}$, and $\mathcal{C}_\gamma$ be the $\frac{\kappa\tau}{4(\frac{d^{\frac{1}{4}} B}{\sqrt{\lambda}} + H)}$-cover for $\{\gamma \in \mathbb{R}^d | \|\gamma\| \leq \sqrt{d}\}$.

Also, let $\mathcal{C}_A$ be the $\frac{\kappa^2}{16}$-cover for $\{A \in \mathbb{R}^{d \times d} | \|A\|_F \leq \frac{\sqrt{d} B^2}{\lambda}\}$, and let $\mathcal{C}_{A'}$ be the $\frac{\kappa^2 \tau^2}{16(\frac{d^{\frac{1}{4}} B}{\sqrt{\lambda}} + 3H)^2}$-cover for $\{A' \in \mathbb{R}^{d \times d} | \|A'\|_F \leq \frac{\sqrt{d} B^2}{\lambda}\}$. By Lemma **D**.5 in (Jin et al., 2020), we have:

$$
\begin{aligned}
|\mathcal{C}_w| &\leq (1 + \frac{8 L_w}{\kappa})^d, \qquad |\mathcal{C}_\gamma| \leq (1 + \frac{8(\frac{d^{\frac{1}{4}} B}{\sqrt{\lambda}} + H)}{\kappa\tau})^d \\
|\mathcal{C}_A| &\leq (1 + \frac{32 \sqrt{d} B^2}{\lambda \kappa^2})^{d^2}, \qquad |\mathcal{C}_{A'}| \leq (1 + \frac{32 \sqrt{d} (B')^2 (\frac{d^{\frac{1}{4}} B}{\sqrt{\lambda}} + 3H)^2}{\lambda \tau^2 \kappa^2})^{d^2}
\end{aligned}
\tag{21}
$$

Now, by Eq. (20), for any $V_1 = V_h^k$ generated by Algorithm 1 and $k \geq K'$, there exist $w_2 \in \mathcal{C}_w$, $\gamma_2 \in \mathcal{C}_\gamma$, $A_2 \in \mathcal{C}_A$, and $A_2' \in \mathcal{C}_{A'}$ such that:

$$dist(V_1, V_2) = \sup_s |V_1(s) - V_2(s)| \leq \kappa$$

Hence, it holds that $\mathcal{N}_\kappa \leq |\mathcal{C}_w| \, |\mathcal{C}_\gamma| \, |\mathcal{C}_A| \, |\mathcal{C}_{A'}|$, which yields:

$$\log(\mathcal{N}_\kappa) \leq \log(|\mathcal{C}_w|) + \log(|\mathcal{C}_\gamma|) + \log(|\mathcal{C}_A|) + \log(|\mathcal{C}_{A'}|)$$

$$\leq d\left[ \log(1 + \frac{8L_w}{\kappa}) + \log(1 + \frac{8(\frac{d^{\frac{1}{4}}B}{\sqrt{\lambda}} + H)}{\kappa\tau})\right]$$

$$+ d^2[\log(1 + \frac{32\sqrt{d}B^2}{\lambda\kappa^2}) + \log(1 + \frac{32\sqrt{d}(B')^2(\frac{d^{\frac{1}{4}}B}{\sqrt{\lambda}} + 3H)^2}{\lambda\tau^2\kappa^2})\right]. \quad \square$$

## B.2. Covering Number in Linear MDPs with Instantaneous Hard Constraints under Star-Convexity Assumtion (Theorem 5.1)

Here, we bound the covering number for star-convex cases. The main difference from our proof for the Local Point Assumption lies in Lemma B.3, where we show that in the star-convex case, even when $K' = 0$, the estimated safe set remains stable under small variations in the safety parameters.

**Lemma B.6.** *(Covering number in linear MDPs with instantaneous hard constraints under Star-Convex Assumption). Consider the setting of Theorem 5.1. Let $\kappa$ be a positive number. Then there exists a finite set of functions $\mathcal{V}_\kappa \subset \mathcal{V}$ such that for all $k \in [K]$, there exists a $V \in \mathcal{V}_\kappa$ such that $dist(V, V_h^k) \leq \kappa$. Moreover, let $\mathcal{N}_\kappa$ be the cardinality for $\mathcal{V}_\kappa$, then we will have the following:*

$$\log(\mathcal{N}_\kappa) \leq d\left[ \log(1 + \frac{4L_w}{\kappa}) + \log(1 + \frac{8(\frac{d^{\frac{1}{4}}B}{\sqrt{\lambda}} + H)}{\kappa\tau})\right]$$

$$+ d^2\left[ \log(1 + \frac{32\sqrt{d}B^2}{\lambda\kappa^2}) + \log(1 + \frac{32\sqrt{d}(B')^2(\frac{d^{\frac{1}{4}}B}{\sqrt{\lambda}} + 3H)^2}{\lambda\tau^2\kappa^2})\right]. \tag{22}$$

**Proof:** We equivalently define $A = \beta^2\Lambda^{-1}$ and $A' = \beta'^2\Lambda^{-1}$ and reparameterize the class function $\mathcal{V}$ as follows:

$$V(.) = \min\{\max_{a \in \mathcal{A}}\langle\phi(.,a), w\rangle + \|\phi(.,a)\|_A + g_{A',1}(.,a), H\}$$

$$s.t : \langle\phi(.,a) - \phi(.,a^0), \gamma\rangle + \|\phi(.,a) - \phi(.,a^0)\|_{A'}) \leq \tau, \tag{23}$$

where $\|A\| \leq \frac{B^2}{\lambda}$ and $\|A'\| \leq \frac{(B')^2}{\lambda}$.

Now, let $V_1 = V_h^k$ be the value function generated by Algorithm 1 during the exploration-exploitation phase, and $V_2$ is an arbitrary function in $\mathcal{V}$. Then, there exist parameters $(w_1, \gamma_1, A_1, A_1')$ and $(w_2, \gamma_2, A_2, A_2')$ such that:

$$V_1(s) = \min\{\max_{a \in \mathcal{A}}\langle\phi(s,a), w_1\rangle + \|\phi(s,a)\|_{A_1} + g_{A_1',1}(s,a), \ H\}$$

$$s.t : \langle\phi(s,a), \gamma_1\rangle + \|\phi(s,a)\|_{A_1'} \leq \tau$$

$$V_2(s) = \min\{\max_{a \in \mathcal{A}}\langle\phi(s,a), w_2\rangle + \|\phi(s,a)\|_{A_2}) + g_{A_2',1}(s,a), \ H\}$$

$$s.t : \langle\phi(s,a), \gamma_2\rangle + \|\phi(s,a)\|_{A_2'} \leq \tau \tag{24}$$

Now, define $V_3$ as follows:

$$V_3(s) \triangleq \min\{\max_{a \in \mathcal{A}}\langle\phi(s,a), w_1\rangle + \|\phi(s,a)\|_{A_1}) + g_{A_1',1}(s,a), \ H\}$$

$$s.t : \langle\phi(s,a), \gamma_2\rangle + \|\phi(s,a)\|_{A_2'}) \leq \tau \tag{25}$$

Similar to our Proof for Lemma B.2, we can write $|V_1(s) - V_2(s) \leq |V_1(s) - V_3(s)| + |V_2(s) - V_3(s)|$. Thus, to bound $|V_1(s) - V_2(s)|$, it suffices to separately bound $|V_1(s) - V_3(s)|$ and $|V_2(s) - V_3(s)|$.

**Bounding $|V_1(s) - V_3(s)|$.** Now, we provide the counter part of the Lemma B.3 for the Star-Convex setting as stated below:

**Lemma B.7.** *(No dramatic changes in the estimated safe set under Star-Convexity). Let $\Delta = \|\gamma_2 - \gamma_1\| + \sqrt{\|A_1' - A_2'\|_F}$, where $\|\cdot\|_F$ denotes the Frobenius norm for matrices. For each $s \in \mathcal{S}$, let $\phi(s, a_1^*)$ represent the solution of the constrained optimization problem associated with $V_1 = V_h^k$, for $k \in [K]$. Then, there exists $\alpha_3 \in [\frac{\tau}{\tau + \Delta}, 1]$ such that $\alpha_3 \phi(s, a_1^*) + (1 - \alpha_3)\phi(s, a_s^0)$ is feasible for the constrained optimization problem associated with $V_3$. Specifically, there exists an action $a_3 \in \mathcal{A}$ such that $\phi(s, a_3) = \alpha_3 \phi(s, a_1^*) + (1 - \alpha_3)\phi(s, a_s^0)$, and the following holds:*

$$\langle \phi(s, a_3) - \phi(s, a_s^0), \gamma_2 \rangle + \|\phi(s, a_3) - \phi(s, a_s^0)\|_{A_2'} \leq \tau \tag{26}$$

The proof of the above Lemma is provided in Section B.5.

*Remark* B.8. The key difference between Lemma B.7 and Lemma B.3 lies in their applicability: Lemma B.7 holds for all $k \in [K]$, whereas Lemma B.3 is valid only for $k \geq K'$ with high probability. This distinction is the primary reason why the Pure Exploration phase in Algorithm 1 is unnecessary in star-convex settings.

The remaining steps are identical to the proof steps outlined in Lemma B.2. In fact, using Lemma B.4 we will have:

$$|V_1(s) - V_3(s)| \leq \left(H + \frac{d^{\frac{1}{4}}B}{\sqrt{\lambda}}\right) \frac{\|\gamma_1 - \gamma_2\| + \sqrt{\|A_1' - A_2'\|_F}}{\tau}, \tag{27}$$

**Bounding $|V_2(s) - V_3(s)|$** Now, applying Lemma B.5 on Eq. (18) we get the following:

$$|V_2(s) - V_3(s)| \leq \|w_1 - w_2\| + \sqrt{\|A_1 - A_2\|_F} + \frac{2H}{\tau}\sqrt{\|A_1' - A_2'\|_F} \tag{28}$$

**Bounding $|V_1(s) - V_2(s)|$:** Combining Eq. (27) and Eq. (28), we have:

$$
\begin{aligned}
|V_1(s) - V_2(s)| &\leq |V_1(s) - V_3(s)| + |V_2(s) - V_3(s)| \\
&\leq \left(H + \frac{d^{\frac{1}{4}}B}{\sqrt{\lambda}}\right) \frac{\|\gamma_1 - \gamma_2\| + \sqrt{\|A_1' - A_2'\|_F}}{\tau} + \|w_1 - w_2\| + \sqrt{\|A_1 - A_2\|_F} + \frac{2H}{\tau}\sqrt{\|A_1' - A_2'\|_F}
\end{aligned} \tag{29}
$$

**Final step.** Let $\mathcal{C}_w$ be the $\frac{\kappa}{4}$-cover for $\{w \in \mathbb{R}^d | \|w\| \leq L_w\}$, and $\mathcal{C}_\gamma$ be the $\frac{\kappa\tau}{4(\frac{d^{\frac{1}{4}}B}{\sqrt{\lambda}} + H)}$-cover for $\{\gamma \in \mathbb{R}^d | \|\gamma\| \leq \sqrt{d}\}$. Also, let $\mathcal{C}_A$ be the $\frac{\kappa^2}{16}$-cover for $\{A \in \mathbb{R}^{d \times d} | \|A\|_F \leq \frac{\sqrt{d}B^2}{\lambda}\}$, and let $\mathcal{C}_{A'}$ be the $\frac{\kappa^2\tau^2}{16(\frac{d^{\frac{1}{4}}B}{\sqrt{\lambda}} + 3H)^2}$-cover for $\{A' \in \mathbb{R}^{d \times d} | \|A'\|_F \leq \frac{\sqrt{d}B^2}{\lambda}\}$. By Lemma D.5 in (Jin et al., 2020), we have:

$$
\begin{aligned}
|\mathcal{C}_w| &\leq (1 + \frac{8L_w}{\kappa})^d, \qquad |\mathcal{C}_\gamma| \leq (1 + \frac{8(\frac{d^{\frac{1}{4}}B}{\sqrt{\lambda}} + H)}{\kappa\tau})^d \\
|\mathcal{C}_A| &\leq (1 + \frac{32\sqrt{d}B^2}{\lambda\kappa^2})^{d^2}, \qquad |\mathcal{C}_{A'}| \leq (1 + \frac{32\sqrt{d}(B')^2(\frac{d^{\frac{1}{4}}B}{\sqrt{\lambda}} + 3H)^2}{\lambda\tau^2\kappa^2})^{d^2}
\end{aligned} \tag{30}
$$

Now, by Eq. (29), for any $V_1 = V_h^k$ generated by Algorithm 1 and $k \in [K]$, there exist $w_2 \in \mathcal{C}_w$, $\gamma_2 \in \mathcal{C}_\gamma$, $A_2 \in \mathcal{C}_A$, and $A_2' \in \mathcal{C}_{A'}$ such that:

$$dist(V_1, V_2) = \sup_s |V_1(s) - V_2(s)| \leq \kappa$$

Hence, it holds that $\mathcal{N}_\kappa \leq |\mathcal{C}_w| \, |\mathcal{C}_\gamma| \, |\mathcal{C}_A| \, |\mathcal{C}_{A'}|$, which yields:

$$\log(\mathcal{N}_\kappa) \leq \log(|\mathcal{C}_w|) + \log(|\mathcal{C}_\gamma|) + \log(|\mathcal{C}_A|) + \log(|\mathcal{C}_{A'}|)$$

$$\leq d\left[\log(1 + \frac{8L_w}{\kappa}) + \log(1 + \frac{8(\frac{d^{\frac{1}{4}}B}{\sqrt{\lambda}} + H)}{\kappa\tau})\right]$$

$$+ d^2\left[\log(1 + \frac{32\sqrt{d}B^2}{\lambda\kappa^2}) + \log(1 + \frac{32\sqrt{d}(B')^2(\frac{d^{\frac{1}{4}}B}{\sqrt{\lambda}} + 3H)^2}{\lambda\tau^2\kappa^2})\right]. \quad \square$$

### B.3. Proof of Lemma 5.3

**Proof:** Let $\mathcal{V}'' \triangleq \{v_\gamma \in \mathcal{V} \mid \gamma \in \cup_{i=1}^n\{\frac{1}{i}\}\}$. Firsly, we show that for all $v_1'', v_2'' \in \mathcal{V}''$ with $v_1'' \neq v_2''$, we have $\sup_{s\in\mathcal{S}}|v_1''(s) - v_2''(s)| \geq \frac{1}{3}$. To prove this, note that there exists $i, j \in \mathbb{N}$ such that $v_1'' = v_{\frac{1}{i}}$ and $v_2'' = v_{\frac{1}{j}}$. Now, without loss of generality assume that $i < j$ Thus, for $s = j$ we will have: $v_1''(j) = \frac{1}{3}$ but $v_2''(j) = \frac{2}{3}$. Thus,

$$\forall v_1'', v_2'' \in \mathcal{V}'' \text{ such that } v_1'' \neq v_2'' : \sup_{s\in\mathbb{S}}|v_1''(s) - v_2''(s)| \geq |\frac{2}{3} - \frac{1}{3}| = \frac{1}{3}. \tag{31}$$

Now, to complete our proof, we use a contradiction strategy. Assume that $\mathcal{V}'$ is a $\kappa$-covering for $\mathcal{V}$ such that $|\mathcal{V}'| \leq |\mathbb{S}| - 1$ and $\kappa < \frac{1}{6}$. Since $\mathcal{V}'$ is a $\kappa$-covering, for all $v'' \in \mathcal{V}''$, there exists a $v' \in \mathcal{V}'$ such that

$$\sup_{s\in\mathbb{S}}|v''(s) - v'(s)| < \kappa \leq \frac{1}{6}.$$

However, since $\mathcal{V}'$ has one fewer element than $\mathcal{V}''$, there must exist two functions $v_1'', v_2'' \in \mathcal{V}''$ and one function $v_{1,2}' \in \mathcal{V}'$ such that

$$\sup_{s\in\mathbb{S}}|v_1''(s) - v_{1,2}'(s)| \leq \kappa \quad \text{and} \quad \sup_{s\in\mathbb{S}}|v_2''(s) - v_{1,2}'(s)| \leq \kappa.$$

However, this implies:

$$\sup_{s\in\mathbb{S}}\left|v_1''(s) - v_2''(s)\right| = \sup_{s\in\mathbb{S}}\left|v_1''(s) - v_{1,2}'(s) + \left(v_{1,2}'(s) - v_2''(s)\right)\right|$$

$$\leq \sup_{s\in\mathbb{S}}|v_1''(s) - v_{1,2}'(s)| + \sup_{s\in\mathbb{S}}|v_2''(s) - v_{1,2}'(s)| \leq 2\kappa < \frac{1}{3} \tag{32}$$

Thus, combining Equations 31 and 32 we will have:

$$\frac{1}{3} \leq \sup_{s\in\mathbb{S}}\left|v_1''(s) - v_2''(s)\right| < \frac{1}{3},$$

which is a contraditction and it completes the proof $\square$

### B.4. Proof of Lemmas B.3 -B.5

Before we delve into the proof we state the following helpful lemma:

**Lemma B.9.** *For any $\delta \in (0, 1)$, let $K' \geq \frac{8d}{\epsilon^2}\log(\frac{Hd}{\delta})$. Then, with probability at least $1 - \delta$, the following inequality holds for all $(s, a)$, and $k \geq K'$:*

$$(\phi(s, a) - \phi(s, a_s^0))^T(\Lambda_h^{k,\gamma})^{-1}(\phi(s, a) - \phi(s, a_s^0)) \leq \frac{1}{\lambda + \frac{\lambda_{-k}}{2}}\|\phi(s, a) - \phi(s, a_s^0)\|^2.$$

Using Lemma B.9 we can immdiately prove the following Lemma as well.

**Lemma B.10.** *Let $K' = \max\{\frac{8d}{\epsilon^2}\log(\frac{dH}{\delta}), \frac{2d}{\epsilon^2}(\frac{16\beta_2^2}{\iota^2} - \lambda)\}$, and define $\mathcal{A}_h^{\frac{\iota}{2}}(s) \triangleq \{a \in \mathcal{A} \mid \langle\phi(s, a), \gamma_h^*\rangle \leq \tau - \frac{\iota}{2}\}$. For all $K \geq K'$, with probability at least $1 - \delta$, we have $\mathcal{A}_h^{\frac{\iota}{2}}(s) \subset \mathcal{A}_h^k(s)$.*

The proof of Lemma B.10 is provided in Appendix C.4.

Now, we are ready for the main proof:

**Proof of Lemma B.3**

**Bounding difference between safety values** First of all, we want to bound the safety values of $V_1$ and $V_3$:

$$
\begin{aligned}
&\forall(s,a): |\langle \phi(s,a) - \phi(s,a_s^0), \gamma_2\rangle + \|\phi(s,a) - \phi(s,a_s^0)\|_{A_2'} - \langle \phi(s,a) - \phi(s,a_s^0), \gamma_1\rangle + \|\phi(s,a) - \phi(s,a_s^0)\|_{A_1'}| \\
&\leq |\langle \phi(s,a) - \phi(s,a_s^0), \gamma_2 - \gamma_1\rangle| + |\|\phi(s,a) - \phi(s,a_s^0)\|_{A_2'} - \|\phi(s,a) - \phi(s,a_s^0)\|_{A_1'}| \\
&\leq \|\phi(s,a) - \phi(s,a_s^0)\| \left(\|\gamma_1 - \gamma_2\| + \sqrt{\|A_1' - A_2'\|}\right) \\
&\leq 2\left(\|\gamma_1 - \gamma_2\| + \sqrt{\|A_1' - A_2'\|}\right) = 2\Delta
\end{aligned}
\tag{33}
$$

where the last inequality obtained by Assumption 3.2 that $\|\phi\| \leq 1$.

**Existence of the feasible feature for $V_3$:** Let $\mathcal{A}_h^\iota(s) \triangleq \{a \in \mathcal{A} \mid \langle \phi(s,a), \gamma_h^*\rangle \leq \tau - \iota\}$. Then, we decompose the proof into two sub-cases:

- **Case 1:** If $a_1^* \in \mathcal{A}_h^\iota(s)$, then we show that $a_1^*$ is feasible for the optimization problem of $V_3$. Using Eq.(33) we will have:

$$
\langle \phi(s,a_1^*) - \phi(s,a_s^0), \gamma_2\rangle + \|\phi(s,a_1^*) - \phi(s,a_s^0)\|_{A_2'} \leq \langle \phi(s,a_1^*) - \phi(s,a_s^0), \gamma_1\rangle + \|\phi(s,a_1^*) - \phi(s,a_s^0)\|_{A_1'} + \Delta
\tag{34}
$$

Now, on the event $\mathcal{E}_1$, since $V_1$ is the estimated value function computed by Algorithm 1 (recall that $V_1 = V_h^k$), we have:

$$
\langle \phi(s,a_1^*) - \phi(s,a_s^0), \gamma_1\rangle + \|\phi(s,a_1^*) - \phi(s,a_s^0)\|_{A_1'} + \Delta \leq \langle \phi(s,a_1^*) - \phi(s,a_s^0), \gamma_h^*\rangle + 2\|\phi(s,a_1^*) - \phi(s,a_s^0)\|_{A_1'} + 2\Delta
\tag{35}
$$

But by Lemma B.9, for all $K \geq K'$ we will have:

$$
\langle \phi(s,a_1^*) - \phi(s,a_s^0), \gamma_h^*\rangle + 2\|\phi(s,a_1^*) - \phi(s,a_s^0)\|_{A_1'} + \Delta \leq \langle \phi(s,a_1^*) - \phi(s,a_s^0), \gamma_h^*\rangle + \frac{\iota}{2} + \Delta
\tag{36}
$$

Now, by the Assumption of Lemma B.3, we have $\Delta \leq \frac{\iota}{2}$, using Assumption 2.3, and since $a_1^* \in \mathcal{A}_h^\iota(s)$ we will have:

$$
\langle \phi(s,a_1^*) - \phi(s,a_s^0), \gamma_h^*\rangle + 2\|\phi(s,a_1^*) - \phi(s,a_s^0)\|_{A_1'} + \Delta \leq \tau - \iota + \frac{\iota}{2} + \frac{\iota}{2} = \tau
\tag{37}
$$

Now, combining Equations (34-37) yields:

$$
a_1^* \in \mathcal{A}_h^\iota(s) \implies \langle \phi(s,a_1^*) - \phi(s,a_s^0), \gamma_2\rangle + \|\phi(s,a_1^*) - \phi(s,a_s^0)\|_{A_2'} \leq \tau
\tag{38}
$$

which implies that $a_3 = a_1^*$ is feasible for the constrained optimization problem for $V_3$ and completes the proof for case 1.

- **Case 2:** Now, we only need to show the proof for the case that $a_1^* \in (\mathcal{A}_h^\iota(s))^C \cap \mathcal{A}_h^k(s)$, where $(\mathcal{A}_h^\iota(s))^C$ is the complement of the set $\mathcal{A}_h^\iota(s)$, and $\mathcal{A}_h^k(s)$ is the feasible set for the optimization problem for $V_1$ (recall that $V_1 = V_h^k$). Note that $a_1^* \in (\mathcal{A}_h^\iota(s))^C \cap \mathcal{A}_h^k(s)$ implies that $\tau - \iota \leq \langle \phi(s,a_1^*), \gamma_h^*\rangle \leq \tau$, which implies $\langle \frac{\tau-\iota}{\langle \phi(s,a_1^*), \gamma_h^*\rangle}\phi(s,a_1^*), \gamma_h^*\rangle = \tau - \iota$. Then, by Assumption 3.2 we have $\alpha\phi(s,a_1^*) + (1-\alpha)\phi(s,a_s^0) \in \mathcal{F}_s$, where $\alpha = \frac{\tau-\iota}{\langle \phi(s,a_1^*), \gamma_h^*\rangle}$, i.e., there exists

an $a' \in \mathcal{A}$ such that $\phi(s, a') = \alpha\phi(s, a_1^*) + (1 - \alpha)\phi(s, a_s^0)$. Now, since $a' \in \mathcal{A}_h^\iota(s)$, from Case 1 (Eq. (38)) we have that:

$$\langle \phi(s, a') - \phi(s, a_s^0), \gamma_2 \rangle + \|\phi(s, a') - \phi(s, a_s^0)\|_{A_2'} \leq \tau, \tag{39}$$

which implies that $a'$ is feasible for the constrained optimization problem of $V_3$.

On the other hand, let $\alpha' = \frac{\tau}{\tau + \Delta}$, then we will have:

$$\left\langle \left(\alpha'\phi(s, a_1^*) + (1 - \alpha')\phi(s, a_1^*)\right) - \phi(s, a_1^*), \gamma_2 \right\rangle + \left\|\left(\alpha'\phi(s, a_1^*) + (1 - \alpha')\phi(s, a_s^0)\right) - \phi(s, a_s^0)\right\|_{A_2'}$$

$$= \alpha' \left( \langle \phi(s, a_1^*) - \phi(s, a_s^0), \gamma_2 \rangle + \|\phi(s, a_1^*) - \phi(s, a_s^0)\|_{A_2'} \right) \tag{40}$$

$$\leq \alpha'(\langle \phi(s, a_1^*) - \phi(s, a_s^0), \gamma_1 \rangle + \|\phi(s, a_1^*) - \phi(s, a_s^0)\|_{A_1'} + \Delta) \leq \tau$$

where the last ineuqlity obtained by the fact that $a_1^*$ is feasible for the constrained problem of $V_1$, and $\alpha' = \frac{\tau}{\tau + \Delta}$.

Now, let $\alpha_3 = \max\{\alpha, \alpha'\}$. Since $\frac{\tau - \iota}{\langle \phi(s, a_1^*), \gamma_h^* \rangle} = \alpha \leq \alpha_3 \leq 1$, we can apply Assumption 3.2 to find that $\alpha_3\phi(s, a_1^*) + (1 - \alpha_3)\phi(s, a_s^0) \in \mathcal{F}_s$, i.e. there exists an $a_3 \in \mathcal{A}$ such that $\phi(s, a_3) = \alpha_3\phi(s, a_1^*) + (1 - \alpha_3)\phi(s, a_s^0)$. From Eq.(39) and Eq.(40), it follows that $\phi(s, a_3)$ is feasible for the constrained problem of $V_3$, i.e. $\langle \phi(s, a_3) - \phi(s, a_s^0), \gamma_2 \rangle + \|\phi(s, a_3) - \phi(s, a_s^0)\|_{A_2'} \leq \tau$. $\square$

**Proof of Lemma B.4** Using Lemma B.3, there exists an action $a_3$ which is feasible for $V_3(.)$, and $\phi(s, a_3) = \alpha_3\phi(s, a_1^*) + (1 - \alpha_3)\phi(s, a_s^0)$ for some $\alpha \in [\frac{\tau}{\tau + \Delta}, 1]$. Thus, we have:

$$V_3(s) \geq \min\{\langle \phi(s, a_3), w_1 \rangle + \|\phi(s, a_3)\|_{A_1} + g_{A_1', 1}(s, a_3), \ H\}$$

$$= \min \left\{ \langle \alpha_3\phi(s, a_1^*) + (1 - \alpha_3)\phi(s, a_s^0), w_1 \rangle + \|\alpha_3\phi(s, a_1^*) + (1 - \alpha_3)\phi(s, a_s^0)\|_{A_1} \right.$$

$$\left. + \|\left(\alpha_3\phi(s, a_1^*) + (1 - \alpha_3)\phi(s, a_s^0)\right) - \phi(s, a_s^0)\|_{A_1'}, \ H \right\} \tag{41}$$

$$\geq \min \left\{ \alpha_3\left(\langle \phi(s, a_1^*), w_1 \rangle + \|\phi(s, a_1^*)\|_{A_1} + \phi(s, a_1^*)\|_{A_1'}\right) - (1 - \alpha_3)\|\phi(s, a_1^*)\|_{A_1}, \ H \right\}$$

$$\geq \min \left\{ \alpha_3\left(\langle \phi(s, a_1^*), w_1 \rangle + \|\phi(s, a_1^*)\|_{A_1} + \phi(s, a_1^*)\|_{A_1'}\right), \ H \right\} - (1 - \alpha_3)\|\phi(s, a_1^*)\|_{A_1}$$

$$\geq \alpha_3 V_1(s) - (1 - \alpha_3)\|\phi(s, a_1^*)\|_{A_1},$$

where the last inequality obtained by the fact that $\alpha_3 \leq 1$. Thus:

$$V_1(s) - V_3(s) \leq (1 - \alpha_3)\left(V_1(s) + \|\phi(s, a_1^*)\|_{A_1}\right) \tag{42}$$

Now, since $V_1 \leq H$ we can continue Eq.(42) as follows:

$$V_1(s) - V_3(s) \leq \frac{\Delta}{\tau + \Delta}\left(H + \|\phi(s, a_1^*)\|_{A_1}\right) \leq \frac{\Delta}{\tau + \Delta}\left(H + \sqrt{\|A_1\|_F}\right), \tag{43}$$

where $\|A_1\|_F$ is the Frobenius norm of the matrix $A_1$. $\square$

**Proof of Lemma B.5:** Consider an arbitrary pair $(s, a)$, then we have:

$$|g_{A_1', 1}(s, a) - g_{A_2', 1}(s, a)| = 2H\frac{\left| \|\phi(s, a) - \phi(s, a_s^0)\|_{A_1'} - \|\phi(s, a) - \phi(s, a_s^0)\|_{A_2'} \right|}{\tau} \leq 2H\frac{\sqrt{\|A_1' - A_2'\|}}{\tau} \tag{44}$$

where, the last inequality is obtained by the fact that $|\sqrt{x} - \sqrt{y}| \leq \sqrt{|x - y|}$ for any $x, y \geq 0$. Now, using the fact that Frobenius norm is larger than matrix-norm we can continue:

$$|g_{A_1', 1}(s, a) - g_{A_2', 1}(s, a)| \leq 2H\frac{\sqrt{\|A_1' - A_2'\|}}{\tau} \leq 2H\frac{\sqrt{\|A_1' - A_2'\|_F}}{\tau} \tag{45}$$

The last inequality completes the proof. $\square$

### B.5. Proof of Lemma B.7

**Proof:**

**Bounding difference between safety values**   First of all, we want to bound the safety values of $V_1$ and $V_3$:

$$
\begin{aligned}
\forall (s,a) : & \left| \langle \phi(s,a) - \phi(s,a_s^0), \gamma_2 \rangle + \|\phi(s,a) - \phi(s,a_s^0)\|_{A_2'} - \langle \phi(s,a) - \phi(s,a_s^0), \gamma_1 \rangle + \|\phi(s,a) - \phi(s,a_s^0)\|_{A_1'} \right| \\
& \leq \left| \langle \phi(s,a) - \phi(s,a_s^0), \gamma_2 - \gamma_1 \rangle \right| + \left| \|\phi(s,a) - \phi(s,a_s^0)\|_{A_2'} - \|\phi(s,a) - \phi(s,a_s^0)\|_{A_1'} \right| \\
& \leq \|\phi(s,a) - \phi(s,a_s^0)\| \left( \|\gamma_1 - \gamma_2\| + \sqrt{\|A_1' - A_2'\|} \right) \\
& \leq 2\|\gamma_1 - \gamma_2\| + \sqrt{\|A_1' - A_2'\|} = 2\Delta
\end{aligned}
\tag{46}
$$

where the last inequality obtained by Assumption 3.2 that $\|\phi\| \leq 1$.

**Existence of the feasible feature for $V_3$:**   By Assumption 3.1 we have $\alpha\,\phi(s,a_1^*) + (1-\alpha)\phi(s,a_s^0) \in \mathcal{F}_s$, where $\alpha = \frac{\tau}{\tau+\Delta}$, i.e., there exists an $a' \in \mathcal{A}$ such that $\phi(s,a') = \alpha\phi(s,a_1^*) + (1-\alpha)\phi(s,a_s^0)$. Now, we show that $a'$ is feasible for the constrained problem $V_3$. In fact, we have:

$$
\langle \phi(s,a') - \phi(s,a_s^0), \gamma_2 \rangle + \|\phi(s,a') - \phi(s,a_s^0)\|_{A_2'} = \alpha \left( \langle \phi(s,a_1^*) - \phi(s,a_s^0), \gamma_2 \rangle + \|\phi(s,a_1^*) - \phi(s,a_s^0)\|_{A_2'} \right)
\tag{47}
$$

Now, using Eq.(46) we can continue Eq.(47) as follows:

$$
\begin{aligned}
\langle \phi(s,a'), \gamma_2 \rangle + \|\phi(s,a')\|_{A_2'} &= \alpha \left( \langle \phi(s,a_1^*) - \phi(s,a_s^0), \gamma_2 \rangle + \|\phi(s,a_1^*) - \phi(s,a_s^0)\|_{A_2'} \right) \\
& \leq \alpha \left( \langle \phi(s,a_1^*) - \phi(s,a_s^0), \gamma_1 \rangle + \|\phi(s,a_1^*) - \phi(s,a_s^0)\|_{A_1'} + \Delta \right) \\
& \leq \frac{\tau}{\tau+\Delta}(\tau + \Delta) = \tau,
\end{aligned}
\tag{48}
$$

where the last inequality obtained by the fact that $a_1^*$ is feasible for the constrained problem of $V_1$, and $\alpha = \frac{\tau}{\tau+\Delta}$. Thus, $\langle \phi(s,a'), \gamma_2 \rangle + \|\phi(s,a')\|_{A_2'} \leq \tau$, which implies that $a'$ is feasible for the constrained problem of $V_3$. Now, setting $a_3 = a'$ and $\alpha_3 = \alpha'$ concludes the proof. $\square$

## C. Proof Steps of Theorem 5.4

We start with the following definition:

**Definition C.1.**   For any fixed policy, the event $\mathcal{E}_2$ is defined as:

$$
\begin{aligned}
\mathcal{E}_2 \triangleq \{ & |\langle \phi(s,a), w_h^k \rangle - Q_h^\pi(s,a) + E_{s' \sim \mathbb{P}_h(.|s,a)}[V_{h+1}^\pi(s') - V_{h+1}^k(s')]| \leq \beta_1 \|\phi(s,a)\|_{(\Lambda_h^k)^{-1}}, \\
& \forall (a,s,h,k) \in \mathcal{A} \times \mathcal{S} \times [H] \times [K] \},
\end{aligned}
\tag{49}
$$

where $\beta_1 = c_\beta \cdot dH\sqrt{\log(\frac{2dT}{\delta\tau})}$, and $c_\beta$ is a constant.

**Lemma C.2.**   *Under the setup defined in Theorem 5.4, for all $K \geq K'$, there exists a constant $c_\beta > 0$, such that for any fixed $\delta \in (0,1)$, the event $\mathcal{E}_2$ holds with probability at least $1 - \delta$.*

**Proof:**   Similar to the proof steps of Lemma B.4. in (Jin et al., 2020) we will have:

$$
\langle \phi(s,a), w_h^k \rangle - Q_h^\pi(s,a) = \langle \phi(s,a), w_h^k - w_h^\pi \rangle =
$$

$$
\underbrace{\langle \phi(s,a), -\lambda(\Lambda_h^k)^{-1}\mathbf{w}_h^\pi \rangle}_{q_1} + \underbrace{\langle \phi(s,a), (\Lambda_h^k)^{-1} \sum_{\tau=1}^{k-1} \phi_h^\tau \left[ V_{h+1}^k(s_{h+1}^\tau) - \mathbb{P}_h V_{h+1}^k(s_h^\tau, a_h^\tau) \right] \rangle}_{q_2}
\tag{50}
$$

$$
\underbrace{\langle \phi(s,a), (\Lambda_h^k)^{-1} \sum_{\tau=1}^{k-1} \phi_h^\tau \mathbb{P}_h \left( V_{h+1}^k - V_{h+1}^\pi \right) (s_h^\tau, a_h^\tau) \rangle}_{q_3}.
$$

We start with bound ing $|q_2|$, which is the term related to the covering number.

**Bounding $|q_2|$**    For our problem,we cannot directly utilize Lemma B.2 from (Jin et al., 2020) to bound $|q_2|$, since in our case Value function is obtained by optimization over $\mathcal{A}_h^k(s)$ instead of $\mathcal{A}$, and $\mathcal{A}_h^k(s)$ varies over the time. This makes bounding the covering number challenging. Thus, we first utilize Theorem D.4 from (Jin et al., 2020) to bound $|q_2|$ in terms of the covering number of Value function in our problem, then, we apply Lemma B.2 to get the counterpart result of Lemma B.2 from (Jin et al., 2020).

We start by applying Lemma D.4 from (Jin et al., 2020) on $|q_2|$, to get that with the probability at least $1 - \frac{\delta}{2}$ we will have:

$$
\begin{aligned}
|q_2| &\leq \|\phi(s,a)(\Lambda_h^k)^{-1}\| \ \|\sum_{\tau=1}^{k-1} \phi_h^\tau \left[ V_{h+1}^k(s_{h+1}^\tau) - \mathbb{P}_h V_{h+1}^k(s_h^\tau, a_h^\tau) \right]\| \\
&\leq \sqrt{\phi(s,a)^\top (\Lambda_h^k)^{-1} \phi(s,a)} \ (4H^2 \left[ \frac{d}{2} \log\left(\frac{k+\lambda}{\lambda}\right) + \log \frac{2\mathcal{N}_\kappa}{\delta} \right] + \frac{8k^2\kappa^2}{\lambda}),
\end{aligned}
\tag{51}
$$

where, $\mathcal{N}_\kappa$ is the cardinality of the set of pre-fixed functions $\mathcal{V}_\kappa \subset \mathcal{V}$ defined in Lemma B.2. Now, considering the fact that Lemma B.2 from (Jin et al., 2020) implies that $\|w_h^k\| \leq 2H\sqrt{\frac{dk}{\lambda}}$. Thus, using Lemma B.2 we can continue Eq.(51) as follows:

$$
\begin{aligned}
|q_2| &\leq \sqrt{\phi(s,a)^\top (\Lambda_h^k)^{-1} \phi(s,a)} \ \Bigg( 4H^2 \Big( \frac{d}{2} \log(\frac{k+\lambda}{\lambda}) + d \Big[ \log(1 + \frac{8H}{\kappa}\sqrt{\frac{dk}{\lambda}}) + \log(1 + \frac{8((2\sqrt{\frac{dk}{\lambda}}+1)H + \frac{B}{\sqrt{\lambda}})}{\kappa\tau}) \Big] \\
&\quad + d^2 \Big[ \log(1 + \frac{32\sqrt{d}B^2}{\lambda\kappa^2}) + \log(1 + \frac{32\sqrt{d}B^2(2H\sqrt{\frac{dk}{\lambda}} + \frac{B}{\sqrt{\lambda}} + 3H)^2}{\lambda\tau^2\kappa^2}) \Big] + \log(\frac{2}{\delta}) \Big) + \frac{8k^2\kappa^2}{\lambda} \Bigg)^{\frac{1}{2}},
\end{aligned}
\tag{52}
$$

where $k \geq K' \geq \frac{2dH}{\iota}$. Thus, by choosing $\kappa = \frac{dH}{k} \leq \frac{\iota}{2}$, and taking $B = \beta_1 = c_\beta dH\sqrt{\log(\frac{2dT}{\delta\tau})}$, and $B' = \beta_2 = \mathbf{O}(\log(1 + kH))$ we will have:

$$
|q_2| \leq CdH \log\left[ 2(c_\beta + 1) \frac{dKH}{\delta\tau} \right] \sqrt{\phi(s,a)^\top (\Lambda_h^k)^{-1} \phi(s,a)},
\tag{53}
$$

where $C$ is a constant.

**Bounding $q_1$ and $q_3$:**    Similar to Lemma B.4 from (Jin et al., 2020) we can bound terms $q_1$ and $q_3$ as follows:

$$
\begin{aligned}
&|\langle \phi(s,a), w_h^k \rangle - Q_h^\pi(s,a) - \mathbb{P}_h \left( V_{h+1}^k - V_{h+1}^\pi \right)(s,a)| \\
&\leq (2H\sqrt{d\lambda} + \sqrt{\lambda}\|w_h^\pi\|)\sqrt{\phi(s,a)^\top (\Lambda_h^k)^{-1} \phi(s,a)} + |q_2|.
\end{aligned}
\tag{54}
$$

**Final step**    Combining Equations 53 and 54 yields the following:

$$
\begin{aligned}
&|\langle \phi(s,a), w_h^k \rangle - Q_h^\pi(s,a) - \mathbb{P}_h \left( V_{h+1}^k - V_{h+1}^\pi \right)(s,a)| \\
&\leq \left( 2H\sqrt{d\lambda} + \sqrt{\lambda}\|w_h^\pi\| + CdH\sqrt{\log\left(2(c_\beta + 1)\frac{dKH}{\delta\tau}\right)} \right) \sqrt{\phi(s,a)^\top (\Lambda_h^k)^{-1} \phi(s,a)}
\end{aligned}
\tag{55}
$$

Now, by Lemma B.1 from (Jin et al., 2020), we have $\|w_h^\pi\| \leq 2H\sqrt{d}$. Therefore, there exists an absolute constant $c_\beta$ such that the following holds:

$$
|\langle \phi(s,a), w_h^k \rangle - Q_h^\pi(s,a) - \mathbb{P}_h \left( V_{h+1}^k - V_{h+1}^\pi \right)(s,a)| \leq c_\beta \cdot dH\sqrt{\log(\frac{2dT}{\delta\tau})}\sqrt{\phi(s,a)^\top (\Lambda_h^k)^{-1} \phi(s,a)} \qquad \square
\tag{56}
$$

Now, we define another important event that ensures safety, i.e., it guarantees with high probability that the estimated safe set by the agent lies within the actual safe set:

**Definition C.3.** The event $\mathcal{E}_1$ is defined as: $\mathcal{E}_1 := \{\mathcal{A}_h^k(s) \subset \mathcal{A}_h^{\text{safe}}(s) \; \forall (h, k) \in [H] \times [K]\}$.

Our intersts lies in the events that both event $\mathcal{E}_1$ and $\mathcal{E}_2$ holds, i.e., the actual safe set and value function are approximated properly. Therefore, we provide the following important Lemma:

**Lemma C.4.** *Under Assumption 2.1, for all $K \geq K'$, there exists a constant $c_\beta > 0$, such that for any fixed $\delta \in (0, \frac{1}{3})$, the event $\mathcal{E} := \mathcal{E}_1 \cap \mathcal{E}_2$ holds with probability at least $1 - 2\delta$.*

**Proof:** Theorem 2 from (Abbasi-Yadkori et al., 2011) can be directly applied to our case to show that the event $\mathcal{E}_1$ holds with a probability of at least $1 - \delta$. Similarly, Lemma C.2 establishes that the event $\mathcal{E}_2$ holds with a probability of at least $1 - \delta$. By applying the union bound, the final result follows. $\square$

### C.1. Proof of Lemma 6.2 (Optimism)

Having Lemma C.4 in hand, we are ready to prove that Algorithm 1 satisfies the optimism property stated in Lemma 6.2, which is an essential step in the final proof of regret's upper upper bound. We first provide a helpful lemma in C.1.1, then in C.1.2 we provide the main proof.

#### C.1.1. HELPFUL LEMMAS

**Lemma C.5.** *Let $k \geq K'$, where $K'$ is specified in Theorem 5.4. Then, there exists an action $a' \in \mathcal{A}_h^k(s)$ such that $\phi(s, a') = \alpha\phi(s, a_s^*) + (1 - \alpha)\phi(s, a_s^0)$, for some $\alpha \in [\frac{\tau}{\tau + 2\beta_2\|\phi(s, a_s^*) - \phi(s, a_s^0)\|_{(\Lambda_h^{k,\gamma})^{-1}}}, 1]$.*

**Proof:** Condiontioned on the event $\mathcal{E}$, when $a_s^* \in \mathcal{A}_h^{\frac{k}{2}}(s)$, then by Lemma B.10 we find that $a^* \in \mathcal{A}_h^{\frac{k}{2}}(s) \subset \mathcal{A}_h^k(s)$, which completes the proof. Thus, it remains to only prove the case that $a^* \in (\mathcal{A}_h^{\frac{k}{2}}(s))^c \cap \mathcal{A}_h^{\text{safe}}(s)$. Let us assume that $a^* \in (\mathcal{A}_h^{\frac{k}{2}}(s))^c \cap \mathcal{A}_h^{\text{safe}}(s)$. Then, one can verify that $\alpha = \max\{\frac{\tau - \frac{\iota}{2}}{\langle\phi(s, a_s^*), \gamma_h^*\rangle}, \frac{\tau}{\tau + 2\beta_2\|\phi(s, a_s^*) - \phi(s, a_s^0)\|_{(\Lambda_h^{k,\gamma})^{-1}}}\} \in [\frac{\tau - \iota}{\langle\phi(s, a_s^*), \gamma_h^*\rangle}, 1]$. Thus, by Assumption 3.2 we can argue that there exists an action $a' \in \mathcal{A}$ such that $\phi(s, a') = \alpha\phi(s, a_s^*) + (1 - \alpha)\phi(s, a_s^0)$. Now, we need to show that $a' \in \mathcal{A}_h^k(s)$ as well. Note that, when $\alpha = \frac{\tau - \frac{\iota}{2}}{\langle\phi(s, a_s^*), \gamma_h^*\rangle}$, then:

$$\begin{aligned}
\langle\phi(s, a'), \gamma_h^*\rangle &= \alpha\langle\phi(s, a_s^*), \gamma_h^*\rangle + (1 - \alpha)\langle\phi(s, a_s^0), \gamma_h^*\rangle \\
&= \alpha\langle\phi(s, a_s^*), \gamma_h^*\rangle + 0 = \tau - \frac{\iota}{2}
\end{aligned} \tag{57}$$

where in the second equality we used Assumption 2.3, and the last equality is obtained by substituting $\alpha = \frac{\tau - \frac{\iota}{2}}{\langle\phi(s, a_s^*), \gamma_h^*\rangle}$. Now, Equation (57) implies that $a' \in \mathcal{A}_h^{\frac{k}{2}}(s)$, and by Lemma B.10 we have $a' \in \mathcal{A}_h^k(s)$. Now, for the other case that $\alpha = \frac{\tau}{\tau + 2\beta_2\|\phi(s, a_s^*) - \phi(s, a_s^0)\|_{(\Lambda_h^{k,\gamma})^{-1}}}$, conditioned on the event $\mathcal{E} = \mathcal{E}_1 \cap \mathcal{E}_2$, we can follow the below steps:

$$\begin{aligned}
0 &\leq \langle\phi(s, a') - \phi(s, a_s^0), \gamma_h^k\rangle + \beta_2\|\phi(s, a) - \phi(s, a_s^0)\|_{(\Lambda_h^{k,\gamma})^{-1}} \\
&= \alpha\big(\langle\phi(s, a_s^*) - \phi(s, a_s^0), \gamma_h^k\rangle + \beta_2\|\phi(s, a_s^*) - \phi(s, a_s^0)\|_{(\Lambda_h^{k,\gamma})^{-1}}\big) \\
&\leq \alpha\big(\langle\phi(s, a_s^*) - \phi(s, a_s^0), \gamma_h^*\rangle + 2\beta_2\|\phi(s, a_s^*) - \phi(s, a_s^0)\|_{(\Lambda_h^{k,\gamma})^{-1}}\big)
\end{aligned} \tag{58}$$

where the last inequality is obtained by conditioning on the event $\mathcal{E}_1 \subset \mathcal{E}$. Now, substituting $\alpha$ with $\frac{\tau}{\tau + 2\beta_2\|\phi(s, a_s^*) - \phi(s, a_s^0)\|_{(\Lambda_h^{k,\gamma})^{-1}}}$ in Eq.(58) yields:

$$\begin{aligned}
0 &\leq \langle\phi(s, a') - \phi(s, a_s^0), \gamma_h^k\rangle + \beta_2\|\phi(s, a) - \phi(s, a_s^0)\|_{(\Lambda_h^{k,\gamma})^{-1}} \\
&\leq \frac{\tau}{\tau + 2\beta_2\|\phi(s, a_{s,h}^*) - \phi(s, a_s^0)\|_{(\Lambda_h^{k,\gamma})^{-1}}}(\langle\phi(s, a_{s,h}^*), \gamma_h^*\rangle + 2\beta_2\|\phi(s, a_{s,h}^*)\|_{(\Lambda_h^{k,\gamma})^{-1}})
\end{aligned} \tag{59}$$

Now, since $a_{s,h}^*$ is the optimal safe solution, it should be feasible as well, i.e., $\langle\phi(s, a_{s,h}^*), \gamma_h^*\rangle \leq \tau$. Thus, we can continue

Eq. (58) as follows:

$$0 \leq \langle \phi(s, a') - \phi(s, a_s^0), \gamma_h^k \rangle + \beta_2 \| \phi(s, a) - \phi(s, a_s^0) \|_{(\Lambda_h^{k,\gamma})^{-1}}$$
$$\leq \frac{\tau}{\tau + 2\beta_2 \| \phi(s, a_{s,h}^*) - \phi(s, a_s^0) \|_{(\Lambda_h^{k,\gamma})^{-1}}} (\tau + 2\beta_2 \| \phi(s, a_{s,h}^*) - \phi(s, a_s^0) \|_{(\Lambda_h^{k,\gamma})^{-1}}) \leq \tau \tag{60}$$

The last inequality implies that $a' \in \mathcal{A}_h^k(s)$. This, completes the proof of this Lemma. $\square$

### C.1.2. MAIN PROOF OF LEMMA 6.2

**Proof:** We employ an induction strategy to establish this lemma. Initially, we define the value functions at time $H + 1$ as $V_{H+1}^k(s) = V_{H+1}^{\pi^*}(s) = 0$ for any state $s \in \mathcal{S}$, confirming the optimism for step $H + 1$. Then, by the induction hypothesis, we assume that for an arbitrary $h \in [H]$, $V_{h+1}^{\pi^*}(s) \leq V_{h+1}^k(s)$ holds for all states $s \in \mathcal{S}$. We now need to prove that $V_h^{\pi^*}(s) \leq V_h^k(s)$ also holds for all states $s \in \mathcal{S}$.

Consider the case where $H \leq \max_{a \in \mathcal{A}_h^k(s)} \langle \phi(s, a), w_h^k \rangle + b_h^k(s, a)$. Then, $V_h^{\pi^*}(s) \leq H = \min\{\langle \phi(s, a), w_h^k \rangle + b_h^k(s, a), H\} = V_h^k(s)$ holds, which completes the proof for this case.

It remains to prove the result for the case where $\max_{a \in \mathcal{A}_h^k(s)} \langle \phi(s, a), w_h^k \rangle + b_h^k(s, a) \leq H$, which implies $V_h^k(s) = \max_{a \in \mathcal{A}_h^k(s)} Q_h^k(s, a) = \max_{a \in \mathcal{A}_h^k(s)} \langle \phi(s, a), w_h^k \rangle + b_h^k(s, a)$.

This brings us to analyze two sub-cases. First of all recall the definition of $\mathcal{A}_h^\iota(s) \triangleq \{a \in \mathcal{A} \mid \langle \phi(s, a), \gamma_h^* \rangle \leq \tau - \iota\}$, then we will have the following sub-cases:

**Sub-case one:** $a_{s,h}^* \in \mathcal{A}_h^{\frac{\iota}{2}}(s)$

For this case, by Lemma B.10, we will have: $a_{s,h}^* \in \mathcal{A}^{\frac{\iota}{2}}(s) \subset \mathcal{A}_h^k(s)$, which implies that $a_{s,h}^* \in \mathcal{A}_h^k(s)$ for all $k \geq K'$. Now, we have:

$$V_h^k(s) = \max_{a \in \mathcal{A}_h^k(s)} Q_h^k(s, a) \geq Q_h^k(s, a_{s,h}^*) = \langle \phi(s, a_{s,h}^*), w_h^k \rangle + b_h^k(s, a_{s,h}^*). \tag{61}$$

Then, conditioned on the event $\mathcal{E}_2$ we have:

$$\langle \phi(s, a_{s,h}^*), w_h^k \rangle + b_h^k(s, a_{s,h}^*)$$
$$\geq Q_h^{\pi^*}(s, a_{s,h}^*) + E_{s' \sim \mathbb{P}_h(.|s, a^{s,h})}[V_{h+1}^k(s') - V_{h+1}^{\pi^*}(s')] - \beta_1 \| \phi(s, a_{s,h}^*) \|_{(\Lambda_h^{k,Q})^{-1}} + b_h^k(s, a_{s,h}^*). \tag{62}$$
$$\geq Q_h^{\pi^*}(s, a_{s,h}^*) + E_{s' \sim \mathbb{P}_h(.|s, a^{s,h})}[V_{h+1}^k(s') - V_{h+1}^{\pi^*}(s')]$$

where the last inequality obtained by the fact that $b_h^k(s, a_{s,h}^*) = \beta_1 \| \phi(s, a_{s,h}^*) \|_{(\Lambda_h^{k,Q})^{-1}} + g_h^k(s, a_{s,h}^*)$ and $g \geq 0$. Now, by induction hypothesis, for all $s$, we have: $V_{h+1}^k(s) \geq V_{h+1}^{\pi^*}(s)$. Thus, we can continue Equation(62) as follows:

$$\langle \phi(s, a_{s,h}^*), w_h^k \rangle + b_h^k(s, a_{s,h}^*)$$
$$\geq Q_h^{\pi^*}(s, a_{s,h}^*) + E_{s' \sim \mathbb{P}_h(.|s, a^{s,h})}[V_{h+1}^k(s') - V_{h+1}^{\pi^*}(s')] \geq Q_h^{\pi^*}(s, a_{s,h}^*) = V_h^{\pi^*}(s) \tag{63}$$

. Equations 61 and 63 together complete the proof for sub-case one.

**Sub-case two:** $a_{s,h}^* \in (\mathcal{A}_h^{\frac{\iota}{2}}(s))^c \cap \mathcal{A}_h^{\text{safe}}(s)$: In this case by Lemma C.5, there exists an action $a' \in \mathcal{A}_h^k(s)$ such that $\phi(s, a') = \alpha \phi(s, a_s^*)$ for some $\alpha \in [\frac{\tau}{\tau + 2\beta_2 \| \phi(s, a_s^*) - \phi(s, a_s^0) \|_{(\Lambda_h^{k,\gamma})^{-1}}}, 1]$. Thus we will have:

$$V_h^k(s) = \max_{a \in \mathcal{A}_h^k(s)} Q_h^k(s, a) \geq Q_h^k(s, a') = \langle \phi(s, a'), w_h^k \rangle + b_h^k(s, a'). \tag{64}$$

Then, conditioned on the event $\mathcal{E}_2$ we have:

$$
\begin{aligned}
\langle \phi(s, a'), w_h^k \rangle &+ b_h^k(s, a') \\
&\geq Q_h^{\pi^*}(s, a') + E_{s' \sim \mathbb{P}_h(.|s,a')}[V_{h+1}^k(s') - V_{h+1}^{\pi^*}(s')] - \beta_1 \|\phi(s, a')\|_{(\Lambda_h^{k,Q})^{-1}} + b_h^k(s, a'). \\
&\geq Q_h^{\pi^*}(s, a') + E_{s' \sim \mathbb{P}_h(.|s,a')}[V_{h+1}^k(s') - V_{h+1}^{\pi^*}(s')] + g_h^k(s, a') \geq Q_h^{\pi^*}(s, a') + g_h^k(s, a')
\end{aligned}
\tag{65}
$$

, where in the last inequality we utilized the induction hypothesis that $V_{h+1}^k(.) \geq V_{h+1}^{\pi^*}(.)$. By considering the fact that $\phi(s, a') = \alpha \phi(s, a_s^*)$ we can continue Equation (65) as follows:

$$
V_h^k(s) \geq \langle \phi(s, a'), w_h^k \rangle + b_h^k(s, a') \geq \alpha \langle \phi(s, a_s^*), w_h^{\pi^*} \rangle + g_h^k(s, a'),
\tag{66}
$$

Now, according to the Equation (4) we will have:

$$
g_h^k(s, a') = \nu \times \left( \beta_2 \|\phi(s, a') - \phi(s, a_s^0)\|_{(\Lambda_h^{k,\gamma})^{-1}} \right) H = (\alpha \nu) \times \left( \beta_2 \|\phi(s, a_s^*) - \phi(s, a_s^0)\|_{(\Lambda_h^{k,\gamma})^{-1}} \right) H
\tag{67}
$$

Now, by setting $\nu = \frac{2}{\tau}$, we can continue Eq. (67) as follows:

$$
g_h^k(s, a') = \nu \times \left( 1 - \frac{\tau}{\tau + 2\beta_2 \|\phi(s, a_s^*) - \phi(s, a_s^0)\|_{(\Lambda_h^{k,\gamma})^{-1}}} \right) H \geq (1 - \alpha) H
\tag{68}
$$

Now, combining Equations (68) and (66), and considering the fact that $V^{\pi^*} \leq H$, we obtain the following:

$$
V_h^k(s) \geq \langle \phi(s, a'), w_h^k \rangle + b_h^k(s, a') \geq \alpha \langle \phi(s, a_s^*), w_h^{\pi^*} \rangle + (1 - \alpha) H \geq V_h^{\pi^*}(s),
\tag{69}
$$

where in the second inequality we used the fact that when rewards are non-negative, then according to Proposition 2.3 from (Jin et al., 2020) we will have: $0 \leq Q_h^{\pi^*}(s, a_s^0) = \langle \phi(s, a_s^0), w_h^{\pi^*} \rangle$. The Equation (69) completes the proof. $\square$

**Final Step:** Now, according to the sub-case 1 and sub-case 2, we have $V_h^{\pi^*} \leq V_h^k$ with the probability of at least $1 - \delta$. Therefore, we can say with the probability of at least $1 - \delta$ the following holds:

$$
\mathcal{T}_1 = \sum_{k=K'}^{K} V_0^{\pi^*}(s_0) - V_0^k(s_0) \leq 0. \qquad \square
$$

### C.2. Proof of Lemma 6.3 (Bounding $\mathcal{T}_2$):

**Lemma C.6.** *Conditioned on the event $\mathcal{E}$, the following inequality holds:*

$$
\mathcal{T}_2 \leq \underbrace{\sum_{k=K'}^{K} \sum_{h=1}^{H} \zeta_h^k}_{\mathcal{T}_{2,1}} + \underbrace{2\beta_1 \sum_{k=K'}^{K} \sum_{h=1}^{H} \|\phi(s_h^k, a_h^k)\|_{(\Lambda_h^k)^{-1}}}_{\mathcal{T}_{2,2}} + \underbrace{\sum_{k=K'}^{K} \sum_{h=1}^{H} (g_h^k(s_h^k, a_h^k))}_{\mathcal{T}_{2,3}},
\tag{70}
$$

*where $\zeta_{h+1}^k := \mathbb{E}_{s' \sim \mathbb{P}_h(.|s_h^k, a_h^k)}[V_{h+1}^k(s') - V_{h+1}^{\pi^k}(s')] - \delta_{h+1}^k$ and $\delta_{h+1}^k := V_h^k(s_h^k) - V_h^{\pi^k}(s_h^k)$.*

**Proof:** First, by the definition of $V_h^k$ and $Q_h^k$ we have:

$$
\begin{aligned}
\delta_{h+1}^k = V_h^k(s_h^k) - V_h^{\pi^k}(s_h^k) &= \min\{Q_h^k(s_h^k, a_h^k), H\} - Q_h^{\pi^k}(s_h^k, a_h^k) \\
&\leq Q_h^k(s_h^k, a_h^k) - Q_h^{\pi^k}(s_h^k, a_h^k) \\
&= \langle \phi(s_h^k, a_h^k), w_h^k \rangle + \beta_1 \|\phi(s_h^k, a_h^k)\|_{(\Lambda_h^k)^{-1}} + g_h^k(s_h^k, a_h^k) - Q_h^{\pi^k}(s_h^k, a_h^k).
\end{aligned}
\tag{71}
$$

Then, on the event $\mathcal{E}$, we have:

$$
\begin{aligned}
&\langle \phi(s_h^k, a_h^k), w_h^k \rangle + \beta_1 \|\phi(s_h^k, a_h^k)\|_{(\Lambda_h^k)^{-1}} + g_h^k(s_h^k, a_h^k) - Q_h^{\pi^k}(s_h^k, a_h^k) \\
&\leq \mathbb{E}_{s' \sim \mathbb{P}_h(.|s_h^k, a_h^k)}[V_{h+1}^k(s') - V_{h+1}^{\pi^k}(s')] \\
&= \delta_{h+1}^k + \zeta_{h+1}^k + 2\beta_1 \|\phi(s_h^k, a_h^k)\|_{(\Lambda_h^k)^{-1}} + g_h^k(s_h^k, a_h^k).
\end{aligned}
\tag{72}
$$

Now, note that we have:

$$
\mathcal{T}_2 = \sum_{k=K'}^{K} V_1^k(s_1^k) - V_1^{\pi^k}(s_1^k) = \sum_{k=K'}^{K} \delta_1^k.
\tag{73}
$$

Note that Lemma 6.2 ensures that on the event $\mathcal{E}$, $0 \leq \delta_h^k$ holds. Consequently, by applying Equations (72) and (73), we can conclude:

$$
\mathcal{T}_2 \leq \sum_{k=K'}^{K} \sum_{h=1}^{H} \zeta_h^k + 2\beta_1 \sum_{k=K'}^{K} \sum_{h=1}^{H} \|\phi(s_h^k, a_h^k)\|_{(\Lambda_h^k)^{-1}} + \sum_{k=K'}^{K} \sum_{h=1}^{H} g_h^k(s_h^k, a_h^k). \qquad \square
\tag{74}
$$

### C.3. Proof of Theorem 5.4

**Proof of Theorem 5.4**   We start by bounding the term $\mathcal{T}_2$:

**Bounding $\mathcal{T}_2$**   To bound the term $\mathcal{T}_2$. Applying Lemma 6.3 we will have:

$$
\mathcal{T}_2 \leq \sum_{k=K'}^{K} \sum_{h=1}^{H} \zeta_h^k + 2\beta_1 \sum_{k=K'}^{K} \sum_{h=1}^{H} \|\phi(s_h^k, a_h^k)\|_{(\Lambda_h^{k,Q})^{-1}} + \left(\frac{2\beta_2 H}{\tau}\right) \sum_{k=K'}^{K} \sum_{h=1}^{H} \|\phi(s_h^k, a_h^k) - \phi(s_h^k, a_{s_h^k}^0)\|_{(\Lambda_h^{k,\gamma})^{-1}}
\tag{75}
$$

In order to bound the first term on the right hand side of Equation (75) note that $\zeta_h^k$ forms a martingale difference sequence with a bounded norm, $|\zeta_h^k| \leq H$. Thus, we can apply Azuma-Hoeffding's inequality:

$$
\mathbb{P}\left( \sum_{k=K'}^{K} \sum_{h=1}^{H} \zeta_h^k \leq 2H \sqrt{(K - K')H \, log(\frac{d(K - K')H}{\delta})} \right) \geq 1 - \delta
\tag{76}
$$

Therefore, with a probability of at least $1 - \delta$ we have:

$$
\sum_{k=K'}^{K} \sum_{h=1}^{H} \zeta_h^k \leq 2H \sqrt{(K - K')H \, log(\frac{d(K - K')H}{\delta})}
\tag{77}
$$

Now, to bound the rest of the right hand side of Equation(75) we can Follow the steps outlined in the proof of Theorem 3 in (Abbasi-Yadkori et al., 2011), as follows:

$$
\begin{aligned}
&2\beta_1 \sum_{k=K'}^{K} \sum_{h=1}^{H} \|\phi(s_h^k, a_h^k)\|_{(\Lambda_h^{k,Q})^{-1}} + \left(\frac{2\beta_2 H}{\tau}\right) \sum_{k=K'}^{K} \sum_{h=1}^{H} \|\phi(s_h^k, a_h^k) - \phi(s_h^k, a_{s_h^k}^0)\|_{(\Lambda_h^{k,\gamma})^{-1}} \\
&= 2\beta_1 \sum_{h=1}^{H} \sum_{k=K'}^{K} \|\phi(s_h^k, a_h^k)\|_{(\Lambda_h^{k,Q})^{-1}} + \left(\frac{2\beta_2 H}{\tau}\right) \sum_{h=1}^{H} \sum_{k=K'}^{K} \|\phi(s_h^k, a_h^k) - \phi(s_h^k, a_{s_h^k}^0)\|_{(\Lambda_h^{k,\gamma})^{-1}} \\
&\leq 2\beta_1 \sum_{h=1}^{H} \sqrt{(K - K') \sum_{k=K'}^{K} \|\phi(s_h^k, a_h^k)\|_{(\Lambda_h^{k,Q})^{-1}}} + \frac{2\beta_2 H}{\tau} \sum_{h=1}^{H} \sqrt{(K - K') \sum_{k=K'}^{K} \|\phi(s_h^k, a_h^k) - \phi(s_h^k, a_{s_h^k}^0)\|_{(\Lambda_h^{k,\gamma})^{-1}}}
\end{aligned}
\tag{78}
$$

By Assumption 2.1, given that $L \leq 1$ and $\lambda = 1$ in Algorithm 1, it can be shown that $\|\phi(s_h^k, a_h^k)\|_{(\Lambda_h^{k,Q})^{-1}} \leq 1$, and $\|\phi(s_h^k, a_h^k) - \phi(s_h^k, a_{s_h^k}^0)\|_{(\Lambda_h^{k,\gamma})^{-1}} \leq 2$. Consequently, the following inequality holds:

$$
\begin{aligned}
&\|\phi(s_h^k, a_h^k)\|^2_{(\Lambda_h^{k,Q})^{-1}} \leq 2\log(1+\|\phi(s_h^k, a_h^k)\|^2_{(\Lambda_h^k)^{-1}}), \\
&\|\phi(s_h^k, a_h^k) - \phi(s_h^k, a_{s_h^k}^0)\|^2_{(\Lambda_h^{k,\gamma})^{-1}} \leq 2\log\left(1+\|\phi(s_h^k, a_h^k) - \phi(s_h^k, a_{s_h^k}^0)\|^2_{(\Lambda_h^{k,\gamma})^{-1}}\right)
\end{aligned}
\tag{79}
$$

Thus, by Equations(78) and (79) we have:

$$
\begin{aligned}
& 2\beta_1 \sum_{h=1}^H \sqrt{(K-K') \sum_{k=K'}^K \|\phi(s_h^k, a_h^k)\|^2_{(\Lambda_h^{k,Q})^{-1}}} + \frac{2\beta_2 H}{\tau} \sum_{h=1}^H \sqrt{(K-K') \sum_{k=K'}^K \|\phi(s_h^k, a_h^k) - \phi(s_h^k, a_{s_h^k}^0)\|_{(\Lambda_h^{k,\gamma})^{-1}}} \\
& \leq 2\beta_1 \sum_{h=1}^H \sqrt{2(K-K') \sum_{k=K'}^K \log(1+\|\phi(s_h^k, a_h^k)\|^2_{(\Lambda_h^k)^{-1}})} \\
& \quad + \frac{2\beta_2 H}{\tau} \sum_{h=1}^H \sqrt{(K-K') \sum_{k=K'}^K \log\left(1+\|\phi(s_h^k, a_h^k) - \phi(s_h^k, a_{s_h^k}^0)\|^2_{(\Lambda_h^{k,\gamma})^{-1}}\right)} \\
& = 2\beta_1 \sum_{h=1}^H \sqrt{2(K-K')(\log(\det(\Lambda_h^{K,Q})) - \log(\lambda^d))} + \frac{2\beta_2 H}{\tau} \sum_{h=1}^H \sqrt{2(K-K')(\log(\det(\Lambda_h^{K,\gamma})) - \log(\lambda^d))}
\end{aligned}
\tag{80}
$$

where the last inequality is obtained by Lemma 11 from (Abbasi-Yadkori et al., 2011). Now, considering that $\|\phi(s_h^k, a_h^k)\| \leq L$, it follows that the trace of $\Lambda_h^{K,Q}$ is upper bounded by $d\lambda + KL^2$, and the trace of $\Lambda_h^{K,\gamma}$ is upper bounded by $d\lambda + 2KL^2$. Since $\Lambda_h^{K,Q}$ and $\Lambda_h^{K,\lambda}$ are positive definite matrices, the determinant of them can be bounded as follows:

$$
\begin{aligned}
det(\Lambda_h^{K,Q}) &\leq \left(\frac{\text{trace}(\Lambda_h^{K,Q})}{d}\right)^d \leq \left(\frac{d\lambda + (K)L^2}{d}\right)^d, \\
det(\Lambda_h^{K,\gamma}) &\leq \left(\frac{\text{trace}(\Lambda_h^{K,\gamma})}{d}\right)^d \leq \left(\frac{d\lambda + 2(K)L^2}{d}\right)^d.
\end{aligned}
$$

Combining all together yields:

$$
\begin{aligned}
& 2\beta_1 \sum_{h=1}^H \sqrt{2(K-K')(\log(\det(\Lambda_h^{K,Q})) - \log(\lambda^d))} + \frac{2\beta_2 H}{\tau} \sum_{h=1}^H \sqrt{2(K-K')(\log(\det(\Lambda_h^{K,\gamma})) - \log(\lambda^d))} \\
& \leq 2\beta_1 \sum_{h=1}^H \sqrt{2(K-K')(\log((\frac{d\lambda + KL^2}{d})^d) - \log(\lambda^d))} + \frac{2\beta_2 H}{\tau} \sum_{h=1}^H \sqrt{2(K-K')(\log((\frac{d\lambda + 2KL^2}{d})^d) - \log(\lambda^d))} \\
& \leq 2(\beta_1 + \frac{\beta_2 H}{\tau}) \sum_{h=1}^H \sqrt{2(K-K')d\log((\frac{d\lambda + 2KL^2}{\lambda d}))} = 2(\beta_1 + \frac{\beta_2 H}{\tau})H\sqrt{2(K-K')d\log((\frac{d\lambda + 2KL^2}{\lambda d}))}
\end{aligned}
\tag{81}
$$

Given that $\lambda = 1$ and utilizing the Equations (78) through (81), we conclude that:

$$
\begin{aligned}
& 2\beta_1 \sum_{k=K'}^K \sum_{h=1}^H \|\phi(s_h^k, a_h^k)\|_{(\Lambda_h^{k,Q})^{-1}} + \left(\frac{2\beta_2 H}{\tau}\right) \sum_{k=K'}^K \sum_{h=1}^H \|\phi(s_h^k, a_h^k) - \phi(s_h^k, a_{s_h^k}^0)\|_{(\Lambda_h^{k,\gamma})^{-1}} \\
& \leq 2(\beta_1 + \frac{\beta_2 H}{\tau})H\sqrt{2(K-K')d\log((\frac{d\lambda + 2KL^2}{\lambda d}))}
\end{aligned}
\tag{82}
$$

Now, by integrating the bounds from Eq.(77) and Eq. (82), we establish the desired upper bound on $\mathcal{T}_2$:

$$\mathcal{T}_2 \leq 2H\sqrt{(K-K')H\,log(\frac{d(K-K')H}{\delta})} + 2(\beta_1 + \frac{\beta_2 H}{\tau})H\sqrt{2(K-K')d\log((\frac{d\lambda + 2KL^2}{\lambda d}))} \tag{83}$$

**Final step**  Note that by Lemma 6.2, we have $\mathcal{T}_1 \leq 0$. Thus, by applying union bound we will have the desired result as follows:

$$\text{Regret}(K) \leq K'H + \mathcal{T}_1 + \mathcal{T}_2 \leq K'H + \mathcal{T}_2 \leq$$

$$K'H + 2H\sqrt{(K-K')H\,log(\frac{d(K-K')H}{\delta})} + 2(\beta_1 H + \frac{\beta_2 H^2}{\tau})\sqrt{2(K-K')d\log((\frac{d\lambda + 2KL^2}{\lambda d}))}.$$

**Safety**  The safety of our method is guaranteed by Theorem 2 from Abbasi-Yadkori et al. (2011), as stated below:

**Lemma C.7.** *(Theorem 2 in (Abbasi-Yadkori et al., 2011)) Let $\delta \in (0, \frac{1}{H})$. Then, with probability at least $1 - H\delta$, the chosen actions by Algorithm 1 satisfy the safety constraints for all episodes. In other words, for all $(h,k) \in [H] \times [K]$, $\mathcal{A}_h^k(s) \subseteq \mathcal{A}_h^{safe}(s)$.*

$\square$

## C.4. Proof of Lemmas B.9 and B.10

**Definition C.8.** Let $\lambda_- \triangleq \frac{d}{\epsilon^2}$.

**Proof of Lemma B.9:**  Our main strategy is to project the problem into $\mathbb{R}^{d-1}$ (Recall that $\mathcal{F}$ is a $d-1$ dimensional hyper plane and then we apply Lemma 1 from (Amani et al., 2019) to achieve the final result.

**Reltation of the higher dimensional problem to the lower dimensional space, i.e., $\mathbb{R}^{d-1}$**  Recall that in Algortihm 1 the agent samples uniformly from the set of safe actions $\mathcal{D}^\epsilon(s)$ that we rewrite its definition here for the convenience of the reader:

$$\mathcal{D}^\epsilon(s) \triangleq \{a \in \mathcal{A} \mid \|\phi(s,a) - \phi(s, a_s^0)\| = \epsilon\}. \tag{84}$$

According to Proposition 2.2, by Assumption 3.2, we can define the image of the set $\mathcal{D}^\epsilon(s)$ under the feature transformation $\phi(s,.)$ as follows:

$$\phi(s, \mathcal{D}^\epsilon(s)) \triangleq \{\phi(s, a_s^0) + \epsilon \times \sum_{i=1}^{d-1} \alpha_i v_i \mid \sum_{i=1}^{d-1} \alpha_i^2, \ \alpha_i \in \mathbb{R}\}, \tag{85}$$

where $\{v_i\}_{i=1}^{d-1}$ are orthonormal vectors such that $\mathcal{F}_s = \phi(s, a_s^0) + \text{Span}(\{v_i\}_{i=1}^{d-1})$ (Note that $\{v_i\}_{i=1}^{d-1}$ are naturaly orthogonal to the vector $\mu^*$ defined in Proposition 2.2). Thus, random sampling from $\mathcal{D}^\epsilon(s)$ is equivalent to random sampling from the following set:

$$\mathcal{W} \triangleq \{w \in \mathbb{R}^{d-1} \mid \|w\|_2 = \epsilon\} \tag{86}$$

**Equivalency of the result in lower dimensional space**  Considering the Eq.(85), we have: $\phi(s_h^k, a_h^k) = \phi(s_h^k, a_{s_h^k}^0) + \epsilon \times \sum_{i=1}^{d-1} \alpha_i^{k,h} v_i$, and $\phi(s,a) = \phi(s, a_s^0) + \epsilon \times \sum_{i=1}^{d-1} \alpha_i^{s,a} v_i$. Now, considering the fact that $\Lambda_h^{k,\gamma} \triangleq \sum_{\tau=1}^{k-1}(\phi(s_h^\tau, a_h^\tau) - \phi(s_h^\tau, a_{s_h}^0))(\phi(s_h^\tau, a_h^\tau) - \phi(s_h^\tau, a_{s_h}^0))^\top + \lambda I$, we will have:

$$(\phi(s,a) - \phi(s,a_s^0))^\top \Lambda_h^{k,\gamma}(\phi(s,a) - \phi(s,a_s^0)) =$$

$$\epsilon^2 \times (\sum_{i=1}^{d-1} \alpha_i^{s,a} v_i)^\top \left( \lambda I + \epsilon^2 \times \sum_{n=1}^{k} (\sum_{i=1}^{d-1} \alpha_i^{n,h} v_i)(\sum_{i=1}^{d-1} \alpha_i^{n,h} v_i)^\top \right) (\sum_{i=1}^{d-1} \alpha_i^{s,a} v_i). \tag{87}$$

Now, since $\{v_i\}_{i=1}^{d-1}$ are orthonormal, we can continue Eq.(87) as follows:

$$(\phi(s,a) - \phi(s,a_s^0))^\top \Lambda_h^{k,\gamma}(\phi(s,a) - \phi(s,a_s^0))\alpha_i^{s,a} \alpha_i^{n,h})^2) = (w^{s,a})^\top \left( \lambda I_{d-1} + \sum_{n=1}^{k} (w_h^n)(w_h^n)^\top \right) (w^{s,a}), \tag{88}$$

where in the last inequality, $I_{d-1}$ is the identitiy matrix in $\mathbb{R}^{d-1 \times d-1}$, and $w^{a,s} \in \mathbb{R}^{d-1}$, and the $i$-th element of it is $w_i^{s,a} = \epsilon \alpha_i^{s,a}$, and simularly $w_h^n \in \mathcal{W}$, and its $i$-th element is $w_i^{n,h} = \epsilon \alpha_i^{n,h}$.

**Applying Lemma 1 from (Amani et al., 2019).** Let $\Lambda_{h,d-1}^k \triangleq \lambda I_{d-1} + \sum_{n=1}^{k} (w_h^n)(w_h^n)^\top$, where $w_h^n$ is picked randomly from $\mathcal{W}$. Therefore, using Lemma 1 from (Amani et al., 2019) we have:

$$(w^{s,a})^\top \Lambda_{h,d-1}^k (w^{s,a}) \geq \left( \lambda + \frac{\lambda_- k}{2} \right) \|w^{s,a}\|_2^2$$

Thus, using Eq.(88) we have:

$$(\phi(s,a) - \phi(s,a_s^0))^\top \Lambda_h^{k,\gamma}(\phi(s,a) - \phi(s,a_s^0)) \geq \left( \lambda + \frac{\lambda_- k}{2} \right) \|w^{s,a}\|_2^2$$

$$= \left( \lambda + \frac{\lambda_- k}{2} \right) \|\phi(s,a) - \phi(s,a_s^0)\|_2^2 \tag{89}$$

**Final Step.** Now, note that $\Lambda_h^k$ is Positive Definite matrix. Also, $\mu^*$ (defined in Proposition 2.2) is an an eigen vector of $\Lambda_h^k$, and the rest of the eigen vector of $\Lambda_h^k$ lies on the hyper-plane $\mathcal{H}$. Thus, using Eq.(89) we will have the following:

$$(\phi(s,a) - \phi(s,a_s^0))^\top \left( \Lambda_h^{k,\gamma} \right)^{-1} (\phi(s,a) - \phi(s,a_s^0)) \leq \left( \frac{1}{\lambda + \frac{\lambda_- k}{2}} \right) \|\phi(s,a) - \phi(s,a_s^0)\|_2^2 \tag{90}$$

**Proof of Lemma B.10** : Setting $K' \geq \max\{\frac{8d}{\epsilon^2} \log(\frac{dH}{\delta}), \frac{2d}{\epsilon^2}(\frac{16\beta_2^2}{\iota^2} - \lambda)\}$, then by applying Lemma B.9 we will have:

$$\beta_2 \|\phi(s,a) - \phi(s,a_s^0)\|_{(\Lambda_h^{k,\gamma})^{-1}} \leq \frac{\iota}{4} \tag{91}$$

Therefore, conditioned on the event $\mathcal{E}$ we have:

$$\langle \phi(s,a) - \phi(s,a_s^0), \gamma_h^k \rangle + \beta_2 \|\phi(s,a) - \phi(s,a_s^0)\|_{(\Lambda_h^{k,\gamma})^{-1}} \leq \langle \phi(s,a) - \phi(s,a_s^0), \gamma_h^* \rangle + 2\beta_2 \|\phi(s,a) - \phi(s,a_s^0)\|_{(\Lambda_h^{k,\gamma})^{-1}}$$

$$\leq \langle \phi(s,a) - \phi(s,a_s^0), \gamma_h^* \rangle + \frac{\iota}{2}, \tag{92}$$

where the last ineuqality implies that for all $a \in \mathcal{A}_h^{\frac{\iota}{2}}(s)$, we will have: $\langle \phi(s,a) - \phi(s,a_s^0), \gamma_h^k \rangle + \beta_2 \|\phi(s,a) - \phi(s,a_s^0)\|_{(\Lambda_h^{k,\gamma})^{-1}} \leq \tau$, which implies that $a \in \mathcal{A}_h^k(s)$. $\square$

## D. Proof of Theorem 5.1

Note that our main contribution in Star-Convex cases is bounding the covering number as provided in Lemma B.6. Once we have the result of Lemma B.6, we can apply the proof steps of Theorem 1 in (Amani et al., 2021) to obtain the desired result. However, for sake of completeness we provide the proof steps here as well.

**Lemma D.1.** *Under the setup defined in Theorem 5.4, for all $K \in [K]$, there exists a constant $c_\beta > 0$, such that for any fixed $\delta \in (0,1)$, the event $\mathcal{E}_2$ holds with probability at least $1 - \delta$.*

**Proof:** Similar to the proof steps of Lemma B.4. in (Jin et al., 2020) we will have:

$$
\begin{aligned}
&\langle \phi(s,a), w_h^k \rangle - Q_h^\pi(s,a) = \langle \phi(s,a), w_h^k - w_h^\pi \rangle = \\
&\underbrace{\langle \phi(s,a), -\lambda(\Lambda_h^k)^{-1} \mathbf{w}_h^\pi \rangle}_{q_1} + \underbrace{\langle \phi(s,a), (\Lambda_h^k)^{-1} \sum_{\tau=1}^{k-1} \phi_h^\tau \left[ V_{h+1}^k(s_{h+1}^\tau) - \mathbb{P}_h V_{h+1}^k(s_h^\tau, a_h^\tau) \right] \rangle}_{q_2} \\
&\underbrace{\langle \phi(s,a), (\Lambda_h^k)^{-1} \sum_{\tau=1}^{k-1} \phi_h^\tau \mathbb{P}_h \left( V_{h+1}^k - V_{h+1}^\pi \right)(s_h^\tau, a_h^\tau) \rangle}_{q_3}.
\end{aligned}
\tag{93}
$$

We start with bound ing $|q_2|$, which is the term related to the covering number.

**Bounding $|q_2|$**   For our problem,we cannot directly utilize Lemma B.2 from (Jin et al., 2020) to bound $|q_2|$, since in our case Value function is obtained by optimization over $\mathcal{A}_h^k(s)$ instead of $\mathcal{A}$, and $\mathcal{A}_h^k(s)$ varies over the time. This makes bounding the covering number challenging. Thus, we first utilize Theorem D.4 from (Jin et al., 2020) to bound $|q_2|$ in terms of the covering number of Value function in our problem, then, we apply Lemma B.6 to get the counterpart result of Lemma B.2 from (Jin et al., 2020).

We start by applying Lemma D.4 from (Jin et al., 2020) on $|q_2|$, to get that with the probability at least $1 - \frac{\delta}{2}$ we will have:

$$
\begin{aligned}
|q_2| &\leq \| \phi(s,a)(\Lambda_h^k)^{-1} \| \, \| \sum_{\tau=1}^{k-1} \phi_h^\tau \left[ V_{h+1}^k(s_{h+1}^\tau) - \mathbb{P}_h V_{h+1}^k(s_h^\tau, a_h^\tau) \right] \| \\
&\leq \sqrt{\phi(s,a)^\top (\Lambda_h^k)^{-1} \phi(s,a)} \left( 4H^2 \left[ \frac{d}{2} \log \left( \frac{k+\lambda}{\lambda} \right) + \log \frac{2\mathcal{N}_\kappa}{\delta} \right] + \frac{8k^2 \kappa^2}{\lambda} \right),
\end{aligned}
\tag{94}
$$

where, $\mathcal{N}_\kappa$ is the cardinality of the set of pre-fixed functions $\mathcal{V}_\kappa \subset \mathcal{V}$ defined in Lemma B.6. Now, considering the fact that Lemma B.2 from (Jin et al., 2020) implies that $\| w_h^k \| \leq 2H \sqrt{\frac{dk}{\lambda}}$. Thus, using Lemma B.6 we can continue Eq.(94) as follows:

$$
\begin{aligned}
|q_2| &\leq \sqrt{\phi(s,a)^\top (\Lambda_h^k)^{-1} \phi(s,a)} \left( 4H^2 \left( \frac{d}{2} \log(\frac{k+\lambda}{\lambda}) + d \left[ \log(1 + \frac{8H}{\kappa}\sqrt{\frac{dk}{\lambda}}) + \log(1 + \frac{8((2\sqrt{\frac{dk}{\lambda}} + 1)H + \frac{B}{\sqrt{\lambda}})}{\kappa\tau}) \right] \right. \right. \\
&\left. \left. + d^2 \left[ \log(1 + \frac{32\sqrt{d}B^2}{\lambda\kappa^2}) + \log(1 + \frac{32\sqrt{d}B^2(2H\sqrt{\frac{dk}{\lambda}} + \frac{B}{\sqrt{\lambda}} + 3H)^2}{\lambda\tau^2\kappa^2}) \right] + \log(\frac{2}{\delta}) \right) + \frac{8k^2\kappa^2}{\lambda} \right)^{\frac{1}{2}}.
\end{aligned}
\tag{95}
$$

Thus, by choosing $\kappa = \frac{dH}{k}$, and setting $B = \beta_1 = c_\beta dH \sqrt{\log(\frac{2dT}{\delta\tau})}$, and $B' = \beta_2 = \mathbf{O}(\log(1 + kH))$ we will have:

$$
|q_2| \leq CdH \log \left[ 2(c_\beta + 1)\frac{dKH}{\delta\tau} \right] \sqrt{\phi(s,a)^\top (\Lambda_h^k)^{-1} \phi(s,a)},
\tag{96}
$$

where $C$ is a constant.

**Bounding $q_1$ and $q_3$:**   Similar to Lemma B.4 from (Jin et al., 2020) we can bound terms $q_1$ and $q_3$ as follows:

$$
\begin{aligned}
&| \langle \phi(s,a), w_h^k \rangle - Q_h^\pi(s,a) - \mathbb{P}_h \left( V_{h+1}^k - V_{h+1}^\pi \right)(s,a) | \\
&\leq (2H\sqrt{d\lambda} + \sqrt{\lambda}\| w_h^\pi \|) \sqrt{\phi(s,a)^\top (\Lambda_h^k)^{-1} \phi(s,a)} + |q_2|.
\end{aligned}
\tag{97}
$$

**Final step**  Combining Equations 96 and 97 yields the following:

$$
\begin{aligned}
&|\langle \phi(s,a), w_h^k \rangle - Q_h^\pi(s,a) - \mathbb{P}_h \left( V_{h+1}^k - V_{h+1}^\pi \right)(s,a)| \\
&\qquad \leq \left( 2H\sqrt{d\lambda} + \sqrt{\lambda}\|w_h^\pi\| + CdH\sqrt{\log\left(2(c_\beta+1)\frac{dKH}{\delta\tau}\right)} \right) \sqrt{\phi(s,a)^\top (\Lambda_h^k)^{-1}\phi(s,a)}
\end{aligned}
\tag{98}
$$

Now, by Lemma B.1 from (Jin et al., 2020), we have $\|w_h^\pi\| \leq 2H\sqrt{d}$. Therefore, there exists an absolute constant $c_\beta$ such that the following holds:

$$
|\langle \phi(s,a), w_h^k \rangle - Q_h^\pi(s,a) - \mathbb{P}_h \left( V_{h+1}^k - V_{h+1}^\pi \right)(s,a)| \leq c_\beta \cdot dH\sqrt{\log(\frac{2dT}{\delta\tau})}\sqrt{\phi(s,a)^\top (\Lambda_h^k)^{-1}\phi(s,a)} \qquad \square
\tag{99}
$$

Our intersts lies in the events that both event $\mathcal{E}_1$ and $\mathcal{E}_2$ holds, i.e., the actual safe set and value function are approximed properly. Therefore, we provide the following important Lemma:

**Lemma D.2.** *Under Assumption 2.1, there exists a constant $c_\beta > 0$, such that for any fixed $\delta \in (0, \frac{1}{3})$, the event $\mathcal{E} := \mathcal{E}_1 \cap \mathcal{E}_2$ holds with probability at least $1 - 2\delta$.*

**Proof:**  Theorem 2 from (Abbasi-Yadkori et al., 2011) can be directly applied to our case to show that the event $\mathcal{E}_1$ holds with a probability of at least $1 - \delta$. Similarly, Lemma D.1 establishes that the event $\mathcal{E}_2$ holds with a probability of at least $1 - \delta$. By applying the union bound, the final result follows. $\square$

### D.1. Optimism in Star-Convex cases

Here we prove that Algorithm 1 satisfies the optimism property for the setting of Theorem 5.1, which is an essential step in the final proof of regret's upper upper bound. We first provide a helpful lemma in D.1.1, then in D.1.2 we provide the main proof.

#### D.1.1. HELPFUL LEMMAS (STAR-CONVEX)

**Lemma D.3.** *Consider the setting stated in Theorem 5.1. Then, there exists an action $a' \in \mathcal{A}_h^k(s)$ such that $\phi(s,a') = \alpha\phi(s,a_s^*) + (1-\alpha)\phi(s,a_s^0)$, for some $\alpha \in [\frac{\tau}{\tau + 2\beta_2\|\phi(s,a_s^*) - \phi(s,a_s^0)\|_{(\Lambda_h^{k,\gamma})^{-1}}}, 1]$.*

**Proof:**  Let $\alpha = \frac{\tau}{\tau + 2\beta_2\|\phi(s,a_s^*) - \phi(s,a_s^0)\|_{(\Lambda_h^{k,\gamma})^{-1}}}$. Then, conditioned on the event $\mathcal{E}$, we can follow the below steps:

$$
\begin{aligned}
0 &\leq \langle \phi(s,a') - \phi(s,a_s^0), \gamma_h^k \rangle + \beta_2\|\phi(s,a) - \phi(s,a_s^0)\|_{(\Lambda_h^{k,\gamma})^{-1}} \\
&= \alpha\big( \langle \phi(s,a_s^*) - \phi(s,a_s^0), \gamma_h^k \rangle + \beta_2\|\phi(s,a_s^*) - \phi(s,a_s^0)\|_{(\Lambda_h^{k,\gamma})^{-1}} \big) \\
&\leq \alpha\big( \langle \phi(s,a_s^*) - \phi(s,a_s^0), \gamma_h^* \rangle + 2\beta_2\|\phi(s,a_s^*) - \phi(s,a_s^0)\|_{(\Lambda_h^{k,\gamma})^{-1}} \big)
\end{aligned}
\tag{100}
$$

where the last inequality is obtained by conditioning on the event $\mathcal{E}_1 \subset \mathcal{E}$. Now, substituting $\alpha$ with $\frac{\tau}{\tau + 2\beta_2\|\phi(s,a_s^*) - \phi(s,a_s^0)\|_{(\Lambda_h^{k,\gamma})^{-1}}}$ in Eq.(100) yields:

$$
\begin{aligned}
&0 \leq \langle \phi(s,a') - \phi(s,a_s^0), \gamma_h^k \rangle + \beta_2\|\phi(s,a) - \phi(s,a_s^0)\|_{(\Lambda_h^{k,\gamma})^{-1}} \\
&\leq \frac{\tau}{\tau + 2\beta_2\|\phi(s,a_{s,h}^*) - \phi(s,a_s^0)\|_{(\Lambda_h^{k,\gamma})^{-1}}} \big( \langle \phi(s,a_{s,h}^*) - \phi(s,a_s^0), \gamma_h^* \rangle + 2\beta_2\|\phi(s,a_{s,h}^*) - \phi(s,a_s^0)\|_{(\Lambda_h^{k,\gamma})^{-1}} \big)
\end{aligned}
\tag{101}
$$

Now, since $a_{s,h}^*$ is the optimal safe solution, it should be feasible as well, i.e., $\langle \phi(s,a_{s,h}^*), \gamma_h^* \rangle \leq \tau$. Thus, using Assumption 2.3, we can continue Eq. (101) as follows:

$$
\begin{aligned}
&0 \leq \langle \phi(s,a') - \phi(s,a_s^0), \gamma_h^k \rangle + \beta_2\|\phi(s,a) - \phi(s,a_s^0)\|_{(\Lambda_h^{k,\gamma})^{-1}} \\
&\leq \frac{\tau}{\tau + 2\beta_2\|\phi(s,a_{s,h}^*) - \phi(s,a_s^0)\|_{(\Lambda_h^{k,\gamma})^{-1}}} (\tau + 2\beta_2\|\phi(s,a_{s,h}^*) - \phi(s,a_s^0)\|_{(\Lambda_h^k)^{-1}}) \leq \tau
\end{aligned}
$$

The last inequality implies that $a' \in \mathcal{A}_h^k(s)$. This, completes the proof of this Lemma. $\square$

D.1.2. MAIN PROOF OF OPTIMISM IN STAR-CONVEX CASES

**Lemma D.4.** *(Optimism): Under the setting of Theorem 5.1, conditioned on the event $\mathcal{E}$, the inequality $V_h^{\pi^*}(s) \leq V_h^k(s)$, $\forall (s,h) \in \mathcal{S} \times [H]$, and $k \in [K]$ holds.*

**Proof:** We employ an induction strategy to establish this lemma. Initially, we define the value functions at time $H+1$ as $V_{H+1}^k(s) = V_{H+1}^{\pi^*}(s) = 0$ for any state $s \in \mathcal{S}$, confirming the optimism for step $H+1$. Then, by the induction hypothesis, we assume that for an arbitrary $h \in [H]$, $V_{h+1}^{\pi^*}(s) \leq V_{h+1}^k(s)$ holds for all states $s \in \mathcal{S}$. We now need to prove that $V_h^{\pi^*}(s) \leq V_h^k(s)$ also holds for all states $s \in \mathcal{S}$.

Consider the case where $H \leq \max_{a \in \mathcal{A}_h^k(s)} \langle \phi(s,a), w_h^k \rangle + b_h^k(s,a)$. Then, $V_h^{\pi^*}(s) \leq H = \min\{\langle \phi(s,a), w_h^k \rangle + b_h^k(s,a), H\} = V_h^k(s)$ holds, which completes the proof for this case.

It remains to prove the result for the case where $\max_{a \in \mathcal{A}_h^k(s)} \langle \phi(s,a), w_h^k \rangle + b_h^k(s,a) \leq H$, which implies $V_h^k(s) = \max_{a \in \mathcal{A}_h^k(s)} Q_h^k(s,a) = \max_{a \in \mathcal{A}_h^k(s)} \langle \phi(s,a), w_h^k \rangle + b_h^k(s,a)$.

By LemmaD.3, there exists an action $a' \in \mathcal{A}_h^k(s)$ such that $\phi(s,a') = \alpha \phi(s,a_s^*) + (1-\alpha)\phi(s,a_s^0)$ for some $\alpha \in [\frac{\tau}{\tau + 2\beta_2 \|\phi(s,a_s^*) - \phi(s,a_s^0)\|_{(\Lambda_h^{k,\gamma})^{-1}}}, 1]$. Thus we will have:

$$V_h^k(s) = \max_{a \in \mathcal{A}_h^k(s)} Q_h^k(s,a) \geq Q_h^k(s,a') = \langle \phi(s,a'), w_h^k \rangle + b_h^k(s,a'). \tag{102}$$

Then, conditioned on the event $\mathcal{E}_2$ we have:

$$\begin{aligned}
&\langle \phi(s,a'), w_h^k \rangle + b_h^k(s,a') \\
&\geq Q_h^{\pi^*}(s,a') + E_{s' \sim \mathbb{P}_h(.|s,a')}[V_{h+1}^k(s') - V_{h+1}^{\pi^*}(s')] - \beta_1 \|\phi(s,a')\|_{(\Lambda_h^{k,Q})^{-1}} + b_h^k(s,a'). \\
&\geq Q_h^{\pi^*}(s,a') + E_{s' \sim \mathbb{P}_h(.|s,a')}[V_{h+1}^k(s') - V_{h+1}^{\pi^*}(s')] + g_h^k(s,a') \geq Q_h^{\pi^*}(s,a') + g_h^k(s,a')
\end{aligned} \tag{103}$$

, where in the last inequality we utilized the induction hypothesis that $V_{h+1}^k(.) \geq V_{h+1}^{\pi^*}(.)$. By considering the fact that $\phi(s,a') = \alpha \phi(s,a_s^*) + (1-\alpha)\phi(s,a_s^0)$, we can continue Equation 103 as follows:

$$V_h^k(s) \geq \langle \phi(s,a'), w_h^k \rangle + b_h^k(s,a') \geq \alpha \langle \phi(s,a_s^*), w_h^{\pi^*} \rangle + (1-\alpha)\langle \phi(s,a_s^0), w_h^{\pi^*} \rangle + g_h^k(s,a'), \tag{104}$$

Now, according to the Equation (4) we will have:

$$g_h^k(s,a') = \nu \times \left( \beta_2 \|\phi(s,a') - \phi(s,a_s^0)\|_{(\Lambda_h^{k,\gamma})^{-1}} \right) H = (\alpha\nu) \times \left( \beta_2 \|\phi(s,a_s^*) - \phi(s,a_s^0)\|_{(\Lambda_h^{k,\gamma})^{-1}} \right) H \tag{105}$$

Now, by setting $\nu = \frac{2}{\tau}$, we can continue Eq. (105) as follows:

$$g_h^k(s,a') = \nu \times \left( 1 - \frac{\tau}{\tau + 2\beta_2 \|\phi(s,a_s^*) - \phi(s,a_s^0)\|_{(\Lambda_h^{k,\gamma})^{-1}}} \right) H \geq (1-\alpha)H \tag{106}$$

Now, combining Equations (106) and (104), and considering the fact that $0 \leq V^{\pi^*} \leq H$, we obtain the following:

$$V_h^k(s) \geq \langle \phi(s,a'), w_h^k \rangle + b_h^k(s,a') \geq \alpha \langle \phi(s,a_s^*), w_h^{\pi^*} \rangle + (1-\alpha)H \geq V_h^{\pi^*}(s), \tag{107}$$

where in the second inequality we used the fact that when rewards are non-negative, then according to Proposition 2.3 from (Jin et al., 2020) we will have: $0 \leq Q_h^{\pi^*}(s,a_s^0) = \langle \phi(s,a_s^0), w_h^{\pi^*} \rangle$. The Equation (107) completes the proof. $\square$

**D.2. Regret decomposition in Star-Convex cases**

**Lemma D.5.** *Conditioned on the event $\mathcal{E}$, the following inequality holds:*

$$Regret(K) \leq \sum_{k=1}^{K} \sum_{h=1}^{H} \zeta_h^k + 2\beta_1 \sum_{k=K'}^{K} \sum_{h=1}^{H} \|\phi(s_h^k, a_h^k)\|_{(\Lambda_h^k)^{-1}} + \sum_{k=1}^{K} \sum_{h=1}^{H} (g_h^k(s_h^k, a_h^k)), \tag{108}$$

where $\zeta_{h+1}^k := \mathbb{E}_{s' \sim \mathbb{P}_h(.|s_h^k, a_h^k)}[V_{h+1}^k(s') - V_{h+1}^{\pi^k}(s')] - \delta_{h+1}^k$ and $\delta_{h+1}^k := V_h^k(s_h^k) - V_h^{\pi^k}(s_h^k)$.

**Proof:** First, we can decompose the regret as follows:

$$\text{Regret}(K) = \sum_{k=1}^{K} \left(V_0^{\pi^*}(s_0) - V_0^{\pi^k}(s_0)\right) = \sum_{k=1}^{K} \left(V_0^{\pi^*}(s_0) - V_0^k(s_0)\right) + \sum_{k=1}^{K} \left(V_0^k(s_0) - V_0^{\pi^k}(s_0)\right)$$

$$\leq \sum_{k=1}^{K} \left(V_0^k(s_0) - V_0^{\pi^k}(s_0)\right) \tag{109}$$

where, the last inequlity obtained by Lemma D.4. Now, by the definition of $V_h^k$ and $Q_h^k$ we have:

$$\delta_{h+1}^k = V_h^k(s_h^k) - V_h^{\pi^k}(s_h^k) = \min\{Q_h^k(s_h^k, a_h^k), H\} - Q_h^{\pi^k}(s_h^k, a_h^k)$$

$$\leq Q_h^k(s_h^k, a_h^k) - Q_h^{\pi^k}(s_h^k, a_h^k) \tag{110}$$

$$= \langle \phi(s_h^k, a_h^k), w_h^k \rangle + \beta_1 \|\phi(s_h^k, a_h^k)\|_{(\Lambda_h^k)^{-1}} + g_h^k(s_h^k, a_h^k) - Q_h^{\pi^k}(s_h^k, a_h^k).$$

Then, on the event $\mathcal{E}$, we have:

$$\langle \phi(s_h^k, a_h^k), w_h^k \rangle + \beta_1 \|\phi(s_h^k, a_h^k)\|_{(\Lambda_h^k)^{-1}} + g_h^k(s_h^k, a_h^k) - Q_h^{\pi^k}(s_h^k, a_h^k)$$

$$\leq \mathbb{E}_{s' \sim \mathbb{P}_h(.|s_h^k, a_h^k)}[V_{h+1}^k(s') - V_{h+1}^{\pi^k}(s')] \tag{111}$$

$$= \delta_{h+1}^k + \zeta_{h+1}^k + 2\beta_1 \|\phi(s_h^k, a_h^k)\|_{(\Lambda_h^k)^{-1}} + g_h^k(s_h^k, a_h^k).$$

Also, one can find that the following holds:

$$\sum_{k=1}^{K} V_1^k(s_1^k) - V_1^{\pi^k}(s_1^k) = \sum_{k=1}^{K} \delta_1^k. \tag{112}$$

Note that Lemma D.4 ensures that on the event $\mathcal{E}$, $0 \leq \delta_h^k$ holds. Consequently, by applying Equations (109), (111) and (112), we can conclude:

$$\text{Regret}(K) \leq \sum_{k=1}^{K} \sum_{h=1}^{H} \zeta_h^k + 2\beta_1 \sum_{k=1}^{K} \sum_{h=1}^{H} \|\phi(s_h^k, a_h^k)\|_{(\Lambda_h^k)^{-1}} + \sum_{k=1}^{K} \sum_{h=1}^{H} g_h^k(s_h^k, a_h^k). \qquad \square \tag{113}$$

### D.3. Main Proof of Theorem 5.1

**Proof:** Applying Lemma D.5 we will have:

$$\text{Regret}(K) \leq \sum_{k=1}^{K} \sum_{h=1}^{H} \zeta_h^k + 2\beta_1 \sum_{k=1}^{K} \sum_{h=1}^{H} \|\phi(s_h^k, a_h^k)\|_{(\Lambda_h^{k,Q})^{-1}} + \left(\frac{2\beta_2 H}{\tau}\right) \sum_{k=1}^{K} \sum_{h=1}^{H} \|\phi(s_h^k, a_h^k) - \phi(s_h^k, a_{s_h^k}^0)\|_{(\Lambda_h^{k,\gamma})^{-1}} \tag{114}$$

In order to bound the first term on the right hand side of Equation (114) note that $\zeta_h^k$ forms a martingale difference sequence with a bounded norm, $|\zeta_h^k| \leq H$. Thus, we can apply Azuma-Hoeffding's inequality:

$$\mathbb{P}\left(\sum_{k=1}^{K} \sum_{h=1}^{H} \zeta_h^k \leq 2H \sqrt{(K)H \, log(\frac{d(K)H}{\delta})}\right) \geq 1 - \delta \tag{115}$$

Therefore, with a probability of at least $1 - \delta$ we have:

$$\sum_{k=1}^{K}\sum_{h=1}^{H}\zeta_h^k \leq 2H\sqrt{(K)H\,log(\frac{d(K)H}{\delta})} \tag{116}$$

Now, to bound the rest of the right hand side of Equation(114) we can Follow the steps outlined in the proof of Theorem 3 in (Abbasi-Yadkori et al., 2011), as follows:

$$2\beta_1\sum_{k=1}^{K}\sum_{h=1}^{H}\|\phi(s_h^k,a_h^k)\|_{(\Lambda_h^{k,Q})^{-1}} + (\frac{2\beta_2 H}{\tau})\sum_{k=1}^{K}\sum_{h=1}^{H}\|\phi(s_h^k,a_h^k) - \phi(s_h^k,a_{s_h^k}^0)\|_{(\Lambda_h^{k,\gamma})^{-1}}$$

$$= 2\beta_1\sum_{h=1}^{H}\sum_{k=1}^{K}\|\phi(s_h^k,a_h^k)\|_{(\Lambda_h^{k,Q})^{-1}} + (\frac{2\beta_2 H}{\tau})\sum_{h=1}^{H}\sum_{k=1}^{K}\|\phi(s_h^k,a_h^k) - \phi(s_h^k,a_{s_h^k}^0)\|_{(\Lambda_h^{k,\gamma})^{-1}} \tag{117}$$

$$\leq 2\beta_1\sum_{h=1}^{H}\sqrt{K\sum_{k=1}^{K}\|\phi(s_h^k,a_h^k)\|_{(\Lambda_h^{k,Q})^{-1}}} + \frac{2\beta_2 H}{\tau}\sum_{h=1}^{H}\sqrt{K\sum_{k=1}^{K}\|\phi(s_h^k,a_h^k) - \phi(s_h^k,a_{s_h^k}^0)\|_{(\Lambda_h^{k,\gamma})^{-1}}}$$

By Assumption 2.1, given that $L \leq 1$ and $\lambda = 1$ in Algorithm 1, it can be shown that $\|\phi(s_h^k,a_h^k)\|_{(\Lambda_h^{k,Q})^{-1}}\leq 1$, and $\|\phi(s_h^k,a_h^k) - \phi(s_h^k,a_{s_h^k}^0)\|_{(\Lambda_h^{k,\gamma})^{-1}}\leq 2$. Consequently, the following inequality holds:

$$\|\phi(s_h^k,a_h^k)\|_{(\Lambda_h^{k,Q})^{-1}}^2 \leq 2\log(1+\|\phi(s_h^k,a_h^k)\|_{(\Lambda_h^k)^{-1}}^2),$$
$$\|\phi(s_h^k,a_h^k) - \phi(s_h^k,a_{s_h^k}^0)\|_{(\Lambda_h^{k,\gamma})^{-1}}^2 \leq 2\log\left(1+\|\phi(s_h^k,a_h^k) - \phi(s_h^k,a_{s_h^k}^0)\|_{(\Lambda_h^{k,\gamma})^{-1}}^2\right) \tag{118}$$

Thus, by Equations (117) and (118) we have:

$$2\beta_1\sum_{h=1}^{H}\sqrt{K\sum_{k=1}^{K}\|\phi(s_h^k,a_h^k)\|_{(\Lambda_h^{k,Q})^{-1}}^2} + \frac{2\beta_2 H}{\tau}\sum_{h=1}^{H}\sqrt{K\sum_{k=1}^{K}\|\phi(s_h^k,a_h^k) - \phi(s_h^k,a_{s_h^k}^0)\|_{(\Lambda_h^{k,\gamma})^{-1}}}$$

$$\leq 2\beta_1\sum_{h=1}^{H}\sqrt{2K\sum_{k=1}^{K}\log(1+\|\phi(s_h^k,a_h^k)\|_{(\Lambda_h^k)^{-1}}^2)}$$

$$+ \frac{2\beta_2 H}{\tau}\sum_{h=1}^{H}\sqrt{K\sum_{k=1}^{K}\log\left(1+\|\phi(s_h^k,a_h^k) - \phi(s_h^k,a_{s_h^k}^0)\|_{(\Lambda_h^{k,\gamma})^{-1}}^2\right)} \tag{119}$$

$$= 2\beta_1\sum_{h=1}^{H}\sqrt{2K(\log(\det(\Lambda_h^{K,Q})) - \log(\lambda^d))} + \frac{2\beta_2 H}{\tau}\sum_{h=1}^{H}\sqrt{2K(\log(\det(\Lambda_h^{K,\gamma})) - \log(\lambda^d))}$$

where the last inequality is obtained by Lemma 11 from (Abbasi-Yadkori et al., 2011). Now, considering that $\|\phi(s_h^k,a_h^k)\| \leq L$, it follows that the trace of $\Lambda_h^{K,Q}$ is upper bounded by $d\lambda + KL^2$, and the trace of $\lambda_h^{k,\gamma}$ is upper bounded by $d\lambda + 2KL^2$. Since $\Lambda_h^{K,Q}$ and $\Lambda^{K,\gamma}$ are positive definite matrices, the determinant of $\Lambda_h^{k,Q}$ and $\Lambda_h^{k,\gamma}$ can be bounded by:

$$det(\Lambda_h^{K,Q}) \leq (\frac{\text{trace}(\Lambda_h^{K,Q})}{d})^d \leq (\frac{d\lambda + (K)L^2}{d})^d,$$
$$det(\Lambda_h^{K,\gamma}) \leq (\frac{\text{trace}(\Lambda_h^{K,\gamma})}{d})^d \leq (\frac{d\lambda + 2(K)L^2}{d})^d,$$

Combining all together yields:

$$2\beta_1 \sum_{h=1}^{H} \sqrt{2K(\log(\det(\Lambda_h^{K,Q})) - \log(\lambda^d))} + \frac{2\beta_2 H}{\tau} \sum_{h=1}^{H} \sqrt{2K(\log(\det(\Lambda_h^{K,\gamma})) - \log(\lambda^d))}$$

$$\leq 2\beta_1 \sum_{h=1}^{H} \sqrt{2K(\log((\frac{d\lambda + KL^2}{d})^d) - \log(\lambda^d))} + \frac{2\beta_2 H}{\tau} \sum_{h=1}^{H} \sqrt{2K(\log((\frac{d\lambda + 2KL^2}{d})^d) - \log(\lambda^d))} \quad (120)$$

$$\leq 2(\beta_1 + \frac{\beta_2 H}{\tau}) \sum_{h=1}^{H} \sqrt{2Kd\log((\frac{d\lambda + 2KL^2}{\lambda d}))} = 2(\beta_1 + \frac{\beta_2 H}{\tau})H\sqrt{2Kd\log((\frac{d\lambda + 2KL^2}{\lambda d}))}$$

Given that $\lambda = 1$ and utilizing the Equations (117) through (120), we conclude that:

$$2\beta_1 \sum_{k=1}^{K}\sum_{h=1}^{H}\|\phi(s_h^k, a_h^k)\|_{(\Lambda_h^{k,Q})^{-1}} + (\frac{2\beta_2 H}{\tau})\sum_{k=1}^{K}\sum_{h=1}^{H}\|\phi(s_h^k, a_h^k) - \phi(s_h^k, a_{s_h^k}^0)\|_{(\Lambda_h^{k,\gamma})^{-1}}$$

$$\leq 2(\beta_1 + \frac{\beta_2 H}{\tau})H\sqrt{2Kd\log((\frac{d\lambda + 2KL^2}{\lambda d}))} \quad (121)$$

Now, by integrating the bounds from Eq.(116) and Eq. (121), we establish the desired upper bound on $\mathcal{T}_2$:

$$\text{Regret}(K) \leq 2H\sqrt{(K)H \, log(\frac{d(K)H}{\delta})} + (2\beta_1 + \frac{2\beta_2 H}{\tau})H\sqrt{2Kd\log\left((\frac{d\lambda + 2KL^2}{\lambda d})\right)}$$

**Safety** The safety of our method is guaranteed by Theorem 2 from Abbasi-Yadkori et al. (2011), as stated below:

**Lemma D.6.** *(Theorem 2 in (Abbasi-Yadkori et al., 2011)) Let $\delta \in (0, \frac{1}{H})$. Then, with probability at least $1 - H\delta$, the chosen actions by Algorithm 1 satisfy the safety constraints for all episodes. In other words, for all $(h, k) \in [H] \times [K]$, $\mathcal{A}_h^k(s) \subseteq \mathcal{A}_h^{safe}(s)$.*

$\square$

