# OpenReview forum: "Provably Efficient RL for Linear MDPs under Instantaneous Safety Constraints in Non-Convex Feature Spaces"
_ICML.cc/2025/Conference — ICML 2025 poster_

### Official Review · Reviewer_sJHh · 2025-03-09

**Overall Recommendation:** 3

**Summary:**

This paper establishes a regret bound applicable to both star-convex and non-star-convex cases. Moreover, the violation of safety constraints is zero with high probability throughout the learning process. A key technical challenge in these settings is bounding the covering number of the value-function class, which is essential for achieving value-aware uniform concentration in model-free function approximation. For the star-convex setting, this paper develops a novel technique called Objective–Constraint Decomposition (OCD) to properly bound the covering number. This result also resolves an error in a previous work on constrained RL. In non-star-convex scenarios, where the covering number can become infinitely large, this paper proposes a two-phase algorithm, Non-Convex Safe Least Squares Value Iteration (NCS-LSVI), which first reduces uncertainty about the safe set by playing a known safe policy. After that, it carefully balances exploration and exploitation to achieve the regret bound. Finally, numerical simulations on an autonomous driving scenario demonstrate the effectiveness of NCS-LSVI.

**Claims And Evidence:**

The claims made in this paper are supported by clear and convincing evidence.

**Essential References Not Discussed:**

None

**Experimental Designs Or Analyses:**

The experiments look reasonable.

**Methods And Evaluation Criteria:**

The proposed methods and evaluation criteria make sense.

**Other Comments Or Suggestions:**

Please see the weaknesses above.

**Other Strengths And Weaknesses:**

Strengths:

1. This paper is well executed. It designs an algorithm and establishes regret bounds for safe RL with non-convex feature spaces.
2. Empirical evaluations are provided to demonstrate the effectiveness of the proposed algorithm.
3. This paper is well written overall.

Weaknesses:

1. My main concern is on the unique challenges and technical novelty compared to prior works on safe RL for linear MDPs, especially [Amani et al. 2021]. The proposed algorithm and theoretical analysis in this paper looks similar to [Amani et al. 2021]. More discussions on the challenges brought by non-convex feature spaces are needed.
2. The font in Figure 3 is too small. Is there any more baseline algorithm or adaptation algorithm which can be included for comparison in empirical evaluations?

Reference:
Amani, S., Thrampoulidis, C., and Yang, L. Safe rein forcement learning with linear function approximation. In International Conference on Machine Learning, pp. 243–253. PMLR, 2021.

**Questions For Authors:**

Please see the weaknesses above.

**Relation To Broader Scientific Literature:**

This paper is relevant to the literature.

**Theoretical Claims:**

The theoretical results look reasonable, but I didn’t go through every proof.

---

> ### Author Rebuttal · Authors · 2025-04-01
>
> Thank you for the thoughtful feedback. We respond to each of your comments below.
>
> **Q1** My main concern is on the unique challenges and technical novelty compared to prior works on safe RL, especially [Amani et al. 2021].
>
> **A1**   Below we highlight our unique contributions and technical challenges in two parts.
>
> **1. Identifying and Fixing a Key Proof Gap in Amani et al. [2021]**
>
> We first note that the proof in Amani et al. (2021) is incorrect because they directly apply covering-number arguments from unconstrained RL. In the unconstrained case, $V_h^k$ is computed as a maximum over a fixed decision set $\mathcal{A}$, where the max operator acts as a contraction. Hence, a covering over $Q$-functions directly implies one over $V$-functions. In contrast, in constrained RL, the maximization is performed over the data-dependent estimated safe set $\mathcal{A}_h^k(s)$. Since this set varies with the data, even identical $Q$-functions can induce very different $V$-functions if the estimated safe sets differ significantly. As a result, bounding only the $Q$-function class is not enough to ensure uniform convergence of the value functions.
>
>  To address this issue, we develop a novel Objective–Constraint Decomposition (OCD) technique, explicitly separating the complexities of estimating the $Q$-functions (objective) and estimating the safe set  (please see Section 6).  Our key insight leverages the mild geometric structure provided by star-convexity: small variations in the safety parameters do not drastically alter the feasible set, ensuring two close $Q$-functions yield close value functions despite the evolving safe set. This resolves the proof gap in Amani et al. [2021] and our analysis yields an additional factor of $O(\sqrt{\log(\frac{1}{\tau})})$. This shows how tighter constraints (smaller $\tau$) inflate the covering number. Thus, our contribution significantly advances the theoretical understanding of constrained RL.
>
> **2. Algorithmic and Theoretical Contributions for Non-Star-Convex settings.**
>
> Our next main contribution involves both algorithmic and theoretical advancements. On the algorithmic side, we developed the novel NCS-LSVI algorithm, which differs from the SLUCB-QVI algorithm in Amani et al. [2021], and is designed for non-star-convex safe RL environments, common in applications like autonomous driving and robotics, where obstacles or constraints create disjoint or irregular safe regions. On the theoretical side, we provide a regret analysis of our method. We elaborate below.
>
> **The Challenge of Non-Star-Convex Problems.**
>
> Lemma 5.3 shows that, unlike the star-convex case, small variations in the safety parameter can drastically change  the decision set, leading to an arbitrarily large covering number in non-star-convex scenarios. Thus, in these cases, the algorithm proposed for star-convex settings cannot attain a proper covering bound using our OCD technique, indicating the inadequacy  of previous star-convex-based methods and highlighting the need for novel approaches.
>
>
>  **A Two-Phase Algorithm for Non-Star-Convex Environments**
>
> Motivated by these limitations, we propose NCS-LSVI, a two-phase algorithm designed for non-star-convex feature spaces under our Local Point Assumption. Drawing on the insight that large shifts in the safe set complicate the analysis, NCS-LSVI begins  with a pure-safe exploration phase that samples from a small neighborhood around the initial safe policy.By the end of this phase, the estimated safe set stabilizes in the sense that small changes in the safety parameter lead to bounded changes in the value function with high probability, enabling more tractable covering-number bounds. In Theorem 5.4, we show that by characterizing the number of episodes in the pure exploration phase,  NCS-LSVI achieves sublinear regret, specifically $O(\sqrt{K})$, along with an additional $O(\frac{\log(K)}{\epsilon^2 \iota^2})$ term that stems from the pure exploration phase. This extra term reflects the fundamental complexity of non-star-convexity, making the result nontrivial.
>
>
> **Q2** The font in Figure 3 is too small. Can more baseline or adaptation algorithms be added for comparison?
>
> **A2**  We will revise the font size in Figure 3 for the final submission. To address the baseline comparison, we provide additional simulation plots at this anonymous link https://anonymous.4open.science/r/ICML-Safe-RL-figures-CE2D. Figure [a] shows the regret of NCS-LSVI over more episodes, demonstrating sublinear growth for $K' = 2000$.  Figure [b] shows the regret of a sub-optimal but safe baseline constrained to an \(\epsilon\)-neighborhood of the initial policy. Figure [c] shows the regret of LSVI-UCB (Jin et al., 2020), which achieves lower regret but violates constraints, as shown in Figure [d], where cumulative violations grow linearly. The other two methods have zero violations, so no violation plots are included. We will add these results in the final version.

---

> > ### Comment · Reviewer_sJHh · 2025-04-03
> >
> > Thank you for your explanation on the technical novelty compared to Amani et al. [2021]. My concerns were well addressed. I raised my score from 2 to 3.

---

> > > ### Author Response · Authors · 2025-04-04
> > >
> > > Thank you for your constructive feedback and for reconsidering your score. We are glad our clarifications were helpful and truly appreciate your support.

---

### Official Review · Reviewer_yoBb · 2025-03-13

**Overall Recommendation:** 3

**Summary:**

This paper investigates safe reinforcement learning (RL) with instantaneous safety constraints and linear function approximation, where the objective is to ensure zero violations at each step. The authors first identify a technical error in previous work (Amani et al., 2021) and introduce a novel approach, OCD, to address this issue. They then propose a new assumption on the linear structure, termed the "Local Point Assumption," which is more realistic than the star-convexity assumption in Amani et al. (2021). By incorporating an additional pure exploration phase, they prove that the algorithm achieves sublinear regret while maintaining zero violations. Experimental results further validate their theoretical findings.

**Claims And Evidence:**

The theoretical claims made in the submission are supported by clear proof.

**Essential References Not Discussed:**

I don't find any essential additional references that need to be discussed.

**Experimental Designs Or Analyses:**

The experiment is conducted on an autonomous vehicle path-planning task. However, the results in Figures 3 and 6 do not exhibit a clear sublinear regret pattern. (Even for $K'=2000$, after 2000 episodes, the regret seems still linear and get 6000 regret for 10000 episodes). The authors should run additional episodes to better demonstrate the sublinear regret behavior.

**Methods And Evaluation Criteria:**

The evaluation criteria are the cumulative regret and the violation, which is common and reasonable in the safe RL.

**Other Comments Or Suggestions:**

See the questions part.

**Other Strengths And Weaknesses:**

Strengths:

1. The paper is well-written. The presentation about the previous technical error is clear and easy to understand.

2. The theoretical contribution is essential. [Amani et al. 2021] is an important paper on safe RL with linear function approximation, and identifying and correcting its error represents a big contribution.

Weaknesses:

1. The intuition why the local-point assumption is better than the star-convexity assumption seems unclear. The paper does not contain a convincing example to illustrate that the local-point assumption is reasonable. The example provided just states the unreasonableness about the star-convexity.

2. The experiment result is unsatisfying: First, the author does not clarify why the local-point assumption holds in their experiment setup.  Also, the regret looks linear in the number of episodes. The author should perform more episodes to enhance the quality of the experiment.

**Questions For Authors:**

1. In the given autonomous driving example, I believe the star-convexity assumption still holds. The feasible region remains continuous, while actions corresponding to the inaccessible region are unsafe. In fact, the star-convexity assumption only imposes a continuity structure on the action set, not necessarily on the safe action set. Hence, I think it is a reasonable assumption in real-world scenarios. Could the authors clarify the example further to illustrate why the star-convexity assumption fails in this case? Also, in this case why the local-point assumption hold?

2. Can the covering number be bounded by the union of the covering number of $Q_h^k$ at each step? It will only induce an acceptable $\log K$ factor in the final term. I am happy to increase the score if the author explains why this approach fails.

**Relation To Broader Scientific Literature:**

The key contribution is (a) finding a technical error in an essential paper for safe linear MDP with instantaneous constraint and overcoming it by bounding the covering number using additional tricks, and (b) changing the star-convexity assumption to a more realistic local-point assumption.

**Theoretical Claims:**

I check most of the proof in the paper. It is correct to me. However, the covering number seems to be bounded by the union of the covering number of $V_h^k$ at each step. Hence, it will only induce an extra $\log(K)$ factor in the final regret, making the contribution of correcting the previous theoretical proof somewhat unclear.

---

> ### Author Rebuttal · Authors · 2025-04-01
>
> We appreciate your comments and have addressed your points individually below.
>
> **Q1**  Can the covering number be bounded by the union of the covering number of Q_h^k at each step? It will only induce an acceptable log⁡K factor in the final term.
>
> **A1**  We thank the reviewer for this insightful question. However, this suggested approach does not work in our case due to the unique structure of constrained RL with data-dependent safe sets. We elaborate below.
> While the $Q_h^k$ functions in our setup are linear and their covering number is easy to bound, this is not sufficient for constrained RL. In the unconstrained case, $V_h^k$ is computed as a maximum over a fixed decision set $\mathcal{A}$, where the max operator acts as a contraction. Hence, a covering over $Q$-functions directly implies one over $V$-functions. In contrast, in constrained RL, the maximization is performed over the data-dependent estimated safe set $\mathcal{A}_h^k(s)$. Since this set varies with the data, even identical $Q$-functions can induce very different $V$-functions if the estimated safe sets differ significantly. As a result, bounding only the $Q$-function class is not enough to ensure uniform convergence of the value functions. We address this issue in Section 5 by using OCD to decompose the effects of the estimated safe set and the $Q$-function on the covering number.
>
> **Q2**  In the driving example, the action set seems continuous, so star-convexity may still hold. Could the authors clarify why it fails here, and why the local point assumption applies instead?
>
> **A2**  We agree that the original action space in autonomous driving may appear star-convex. However, in our setup, a pre-trained Collision Avoidance (Collav) module masks out unsafe actions before the RL agent makes decisions. This results in a non-star-convex feasible set. The reason is that collision avoidance itself is inherently non-convex—since it requires the autonomous vehicle to avoid entering a region (typically a ball) around another vehicle. This introduces gaps in the feasible set. So, while the raw action space might be star-convex, the effective decision set seen by the RL agent is not.
>
> To illustrate this concretely, suppose $a_s^0$ is the initial safe action and $ a_s^*$ is the optimal safe action (e.g., moving quickly before the other car arrives). While their convex combination $\alpha \phi(s, a_s^0) + (1 - \alpha)\phi(s, a_s^*)$ might still satisfy the lane-keeping constraint, for some intermediate $ \alpha \in [\epsilon, 1-\iota]$, the resulting action may fall into the collision region, i.e., the car neither waits long enough nor moves fast enough to avoid the other vehicle at the intersection. This violates the collision avoidance constraint and breaks star-convexity.
>
> That said, the Local Point Assumption still holds. Around the initial safe action (e.g., stopping), small perturbations (e.g., low speeds in [0, ε]) remain safe, as they allow the other vehicle to pass. Further, high speeds beyond a threshold $v'$ (up to $v^*$) can also be safe. This assumption only requires local structure and is well-suited for disjoint or irregular safe sets.
>
> Note that, even when $\mathcal{A}$ is star-convex in different applications, the transformed space $\mathcal{F}_s = \phi(s, \mathcal{A})$ may not be star-convex, especially when $\phi$ is nonlinear.
>
> **Q3**   The covering number appears to add only a $\log(K)$ factor, making the correction to the prior proof seem minor.
>
> **A3**  The regret analysis in Amani et al. (2021) is incorrect because it applies unconstrained RL covering-number arguments, ignoring that data-dependent safe sets may significantly alter feasible actions across episodes. A naive union bounding approach would only add a log(K) factor, but it fails once the safe set shifts over time. Our OCD technique explicitly separates Q-function estimation (objective) from safe-set complexity (constraint), yielding an additional term $\mathcal{O}(\sqrt{\log(\frac{1}{\tau})})$ (Lemmas C.2 and C.6). This shows how tighter constraints (smaller $\tau$) inflate the covering number, making our correction nontrivial rather than a minor log(K) overhead.
>
>
> **Q4**  Regret seems linear; more episodes may improve the experiment.
>
>
> **A4**  Our regret is sublinear; please see Figure [a] at the anonymous GitHub link: https://anonymous.4open.science/r/ICML-Safe-RL-figures-CE2D.

---

> > ### Comment · Reviewer_yoBb · 2025-04-07
> >
> > Thanks for the response. I still have questions about Q1. My question is whether it is possible to find a covering number for each timestep $K$? In other words, after the safe set $A_h^k$ is fixed for time $k$, it becomes a normal MDP with restricted actions. Can we bound the covering number for each time step $k$ and get a union of them as the final covering number? (It could be time-dependent since it depends on the shift of the safe action set.) I am unsure whether this data-dependent covering number will lead to some problems in the proof.

---

> > > ### Author Response · Authors · 2025-04-08
> > >
> > > Thank you very much for this insightful question.  Below, we explain  why the approach you have outlined does not resolve this issue, and why we need the uniform concentration bound.
> > >
> > >
> > >
> > > Let us begin with why we need the uniform concentration bound.
> > >
> > >
> > >
> > > **Overview of Value-Aware Uniform Concentration**
> > >
> > > The key step in our analysis is to control the fluctuations arising in the least-squares value iteration. Specifically, we need to establish the bounds of the form at episode $k$:
> > > $$\left\|\sum_{\tau=1}^{k-1}\phi(s_h^\tau,a_h^\tau)\bigl[V_{h+1}^k(s_{h+1}^\tau)-P_h V_{h+1}^k(s_h^\tau,a_h^\tau)\bigr]\right\|_{(\Lambda_h^k)^{-1}}\leq O(d\sqrt{\log K}),.$$
> > >
> > > As discussed by Jin et al. (2020) and Ghosh et al. (2022), handling the dependency between the estimated value function $V_{h+1}^k$ and the collected samples  $[s_{h+1}^{\tau}],\ \tau = 1,\ldots,k-1$ requires value-aware uniform concentration inequalities. This dependency renders the standard self-normalized inequalities insufficient in the model-free setting. The key idea is thus to pre-specify a function class $\mathcal{V_h}$ and subsequently demonstrate that every value function $V_{h}^k$ produced by our algorithm resides within this class, whose log-covering number remains polynomially bounded.
> > >
> > > To apply this value-aware uniform concentration inequality, the general strategy is to fix one function class $\mathcal{V_h}$ in advance, chosen large enough so that it includes every possible value function $V_{h}^k$ our algorithm might produce. This single, fixed function class $\mathcal{V}_{h}$ has a polynomial log-covering number. The value-aware uniform concentration inequality then simultaneously guarantees a high-probability bound uniformly over all value functions within this single function class, across all episodes and timesteps.
> > >
> > >
> > >
> > >
> > >
> > > **The Issue with Bounding Each Episode Separately (why the reviewer’s suggestion will not work)**
> > >
> > > Note that treating each timestep separately and then using a union bound, as noted in He et al. (2023) (see Section 6.3), can inflate the log of the covering number by a factor **proportional to $K$, rather than $\log(K)$**, even in the unconstrained case. Specifically, since the total size of the covering number up to episode K would be the product of individual coverings from episode 1 through K, we have $\mathcal{N} = (\mathcal{N}_q)^K$, resulting in:
> > > $$\log(\mathcal{N}) = \log\left((\mathcal{N}_q)^K\right) = K \log(\mathcal{N}_q),$$
> > >
> > > where $\mathcal{N}$ denotes the total covering number up to episode $K$, and $\mathcal{N_q}$ is the covering number at a single episode.
> > >
> > >
> > >
> > > **The Issue with Uniform Bounding over All Possible Safe Action Sets**
> > >
> > > Moreover, to achieve the uniform concentration bound at each episode, given that we do not know $\mathcal{A_h^k}$ beforehand, how can we achieve the uniform concentration bound? One possible approach, inspired by your suggestion, is to consider all possible safe action sets at each episode and redefine our class of value functions at episode k as follows:
> > >
> > > $$\mathcal{V}^k = \textbraceleft V_h^{k}(\cdot)\mid V_h^{k}(\cdot)=\max_{a \in \mathcal{A}_h^k(\cdot)} Q(\cdot,a), \forall \mathcal{A}_h^k \textbraceright ,$$
> > >
> > > where we explicitly consider all possible estimated safe action sets $\mathcal{A}_h^k$. Since $\mathcal{A}_h^k$ is state-dependent, we must account for all possible action subsets at every state and for every step in the horizon. As a result, the total number of possible safe sets grows exponentially with the size of the state space, the action space, and the horizon, leading to a covering number of order $\exp(|\mathcal{S}| |\mathcal{A}| H)$ even for a single episode $k$. Consequently, this approach quickly becomes intractable as the state space, action space, or horizon $H$ grows.
> > >
> > >
> > >
> > > Thus, despite the intuitive appeal, the fundamental challenge of bounding the covering number under constraints remains unresolved.
> > >
> > >
> > >
> > > **Reference:**
> > >
> > > He, J., Zhao, H., Zhou, D., and Gu, Q. (2023, July). Nearly minimax optimal reinforcement learning for linear Markov decision processes. In International Conference on Machine Learning (pp. 12790–12822). PMLR.

---

### Official Review · Reviewer_2AKc · 2025-03-14

**Overall Recommendation:** 3

**Summary:**

This paper studies the theoretical problem of online reinforcement learning with instantaneous hard constraint in the context of *non-star-convex* decision space, which better characterizes some critic domains requiring safety than the existing star-convex counterpart. The authors propose the Non-Convex Safe Least Square Value Iteration (NCS-LSVI) algorithm which achieves sub-linear online regret for two setups: (i) star-convex decision space, where the results refines the mistake of a previous work; (ii) non-star-convex decision space with the local point assumption. The theoretical results are further demonstrated by some numerical experiments.

**Claims And Evidence:**

Yes, most of the claims made in the submission supported by clear and convincing evidence.

**Essential References Not Discussed:**

To the best knowledge of the reviewer, there is no lacking essential references.

**Experimental Designs Or Analyses:**

Yes, the reviewer checks the validity of the numerical experiments.

**Methods And Evaluation Criteria:**

Yes, the proposed methods make sense for the problem this paper study.

**Other Comments Or Suggestions:**

Please see the question part below.

**Other Strengths And Weaknesses:**

**Strengths:**
1. The paper considers a more realistic decision space model which is non-star-convex.
2. The paper identifies and refines an error in existing results for online RL under safety constraints
3. The theoretical results for online regret bounds in the non-star-convex case is also solid and sound.

**Weaknesses:**
1. Despite interesting, the local point assumption is somehow hard to parse. More intuitive explanations of the assumption could be included.
2. The lower bound of the problem studied is unknown and not discussed.
3. The new results for the non-star-convex setups are not well discussed. There is only the theoretical guarantee but without any explanations and discussions.

**Questions For Authors:**

1. How to understand that the online regret of the problem gets worse when the constraint threshold $\tau$ is decreasing? Especially, the results mean that when $\tau=0$, the algorithm will have no guarantee. If $\tau=0$, we know that the safe action $a_0$ is valid and a baseline policy is to always choose the safe action which in the worst case results in a linear regret. So how to understand the behavior of the upper bound when $\tau$ is approaching zero?
2. What is the reason to consider observed cost values that are also random (there is a sub-Gaussian noise part in the cost value)? What are typical real-world scenarios for such considerations? Technically, does this also contribute to the hardness of the handling of the non-star-convex decision space?

**Relation To Broader Scientific Literature:**

The paper contributes to the line of works on online RL theory under safety constraints. Especially, the paper identifies and refines a theoretical analysis error in a previous work [1] and considers another setup (non-star-convex decision space with local point assumption).

**References:**

[1] Amani, S., Thrampoulidis, C., and Yang, L. Safe reinforcement learning with linear function approximation. In International Conference on Machine Learning, pp. 243–253. PMLR, 2021

**Theoretical Claims:**

Yes, the reviewer is able to briefly check the proofs of the theoretical results.

---

> ### Author Rebuttal · Authors · 2025-04-01
>
> We thank the reviewer for providing the constructive review. Please see our responses to your questions below.
>
> **Q1**  How should we interpret the regret bound as the constraint threshold $\tau$ decreases, especially as $\tau \to 0$, where always choosing the known safe action yields linear regret?
>
>
> **A1**  We implicitly consider the upper bound $\tilde{\mathcal{O}}(\min(KH, (1 + \frac{1}{\tau}) \sqrt{\log(\frac{1}{\tau}) d^3 H^4 K} + \frac{1}{\epsilon^2 \iota^2}))$, ensuring linear regret in the worst case (we will clarify this in the final version). Intuitively, smaller $\tau$ corresponds to tighter budgets, limiting safe exploration opportunities. Theorem 20 in Pacchiano et al. (2024) formalizes this by proving a lower bound of $\Omega(1/\tau^2)$, showing the $\tau$-dependence is information-theoretically necessary. As $\tau \to 0$, the regret becomes linear, and the policy remains sub-optimal, as the only policy we may be able to play without violating the constraint is the prior safe policy.
>
>
> **Q2** Why model cost values with sub-Gaussian noise?
>
> **A2** Measurements are often inaccurate due to sensor errors, environmental uncertainties, or modeling errors. For example, sensor readings in robotics or medical applications are noisy. Assuming observed cost values include sub-Gaussian noise effectively captures these uncertainties. The sub-Gaussian assumption is particularly useful as it models noise with light tails, which is common in practice (e.g., Abbasi-Yadkori et al., 2011).
>
> Regarding the non-star-convex challenge, while randomness in cost and reward observations can add complexity, the primary source of hardness lies in the **environment dynamics** (state transition kernel in the MDP). If the dynamics were independent of the chosen actions, the problem would reduce to a contextual bandit setting, and we could remove the pure-exploration phase (even for non-star-convex decision sets), merely adjusting the exploration-exploitation bonus for near-optimal performance. Thus, the noise in costs and rewards is **not** the main contributor to the problem’s hardness; however, we have included the randomness to make our analysis complete.
>
> **Q3** The non-star-convex results lack explanation and are only stated theoretically.
>
> **A3** Due to rebuttal character limits, we provide a brief version below and will elaborate fully in the final paper:
> The regret dependence on $K$ in Theorem 5.4 is $O(\sqrt{K})$. The key difference between Theorems 5.4 and 5.1 lies in the need for a pure exploration phase: while Theorem 5.1 requires none (i.e., $K' = 0$), Theorem 5.4 uses $K' = O(\log(K)/(\epsilon^2 \iota^2))$, which stems from the Local Point Assumption and reflects the added complexity of non-star-convex settings compared to star-convex ones. This term also appears explicitly in the regret bound, capturing the cost of ensuring safe exploration in such environments.
>
>
> **Q4** The lower bound is unknown and not discussed.
>
> **A4** Since linear bandits are a special case of our setting, Theorem 20 of Pacchiano et al. (2024) and Theorem 3 of Shi et al. (2023) imply a lower bound of $\tilde{\Omega}(\max(\frac{1}{\tau^2}, d H \sqrt{K}))$ in the star-convex case, indicating that the $\tau$-dependence in constrained RL is unavoidable. Our regret bound is $O((1 + 1/\tau)\sqrt{\log(1/\tau)}\sqrt{K})$, which matches the unconstrained regret in Jin et al. (2020) up to additional terms accounting for instantaneous constraints and non-star-convexity. Our goal was to design an algorithm with sublinear regret that remains safe in both star-convex and non-star-convex environments. Deriving a tight lower bound is beyond the scope of this paper.
>
>
> **Q5** The Local Point Assumption is hard to parse and would benefit from a more intuitive explanation.
>
> **A5** Thank you for the suggestion. In the final version, we will add the following intuition to clarify Assumption 3.2: This assumption requires that we can perturb the initial safe action $a^0_s$ slightly, i.e., we can safely sample from a small neighborhood around it, with radius determined by $\epsilon$. For example, in the autonomous driving setup in Section 3, if speed $v_0$ is safe, then speeds in $[v_0 - \epsilon, v_0 + \epsilon]$ should also be safe. The second part of the assumption requires local connectivity property in the small neighborhood around the constraint boundary,  e.g., if $ v^*$ is optimal and exactly at the constraint boundary, then speeds in $[v^* - \iota, v^*]$ must be safe.

---

> > ### Comment · Reviewer_2AKc · 2025-04-04
> >
> > Most of the concerns of mine have been addressed, and I am willing to raise my score to 3 accordingly. Please add the corresponding explanations of the theorem and the intuitions behind assumptions to the revised version.

---

> > > ### Author Response · Authors · 2025-04-04
> > >
> > > We appreciate your support and your decision to raise the score. We will incorporate explanations of the theorem and the intuitions behind the assumptions in the revised version, as you suggested.

---

### Official Review · Reviewer_eWaW · 2025-03-14

**Overall Recommendation:** 4

**Summary:**

The paper addresses the challenge of safe reinforcement learning (RL) under instantaneous hard constraints in an episodic Linear MDP setting. The focus is on scenarios where the set of safe actions can be non-convex (or non-star-convex) – for example, safe actions might form disjoint or irregular regions due to obstacles, as in autonomous driving​. In such tasks, the agent must never violate safety requirements at any time step (as opposed to constraints in expectation or over an episode)​. This strict requirement is critical in domains like robotics and autonomous vehicles (e.g., always avoiding collisions) and makes exploration challenging.

Contributions: The paper’s primary contributions are: (1) Provable performance guarantees for safe RL in both star-convex and more general non-convex safe spaces. It derives a regret bound on the agent’s performance: roughly $\tilde O!\Big((1+\frac{1}{\tau})\sqrt{\ln(\frac{1}{\tau})},d^3H^4K\Big)$ (ignoring constant/log factors), where $d$ is the feature dimension, $H$ the episode length, $K$ the number of episodes, and $\tau$ is the safety threshold​. This bound indicates that the cumulative reward regret grows sublinearly in $K$ (i.e. slower than $O(K)$) under certain conditions, meaning the algorithm learns an optimal safe policy over time. Crucially, safety is never violated with high probability throughout training​ – the algorithm maintains zero constraint violations at all times with overwhelming probability. (2) The paper introduces a novel analytical technique called Objective–Constraint Decomposition (OCD) to handle the complication of a changing safe action set during learning​. OCD provides a way to bound the covering number of the value-function class even when the feasible action set varies over time, which is essential for proving uniform convergence and regret bounds. This technique leverages the geometry of star-convexity, essentially showing that if the safe set is nicely shaped (star-convex), small changes in constraint parameters won’t drastically change the optimal value function​. Using OCD, the authors correct a flaw in a prior analysis from Amani et al. (2021) – an earlier work on safe RL in linear MDPs – by properly accounting for the dynamic safe set in the theoretical guarantees​. (3) For the more difficult non-star-convex case, the paper proposes a new two-phase learning algorithm called Non-Convex Safe LSVI (NCS-LSVI)​. In the first phase, NCS-LSVI performs safe exploration using a known initial safe policy: the agent stays in a local neighborhood of this baseline policy, exploring only actions believed to be safe to gradually expand its knowledge of the safe region​. Once the safe set is sufficiently well-estimated (stabilized with high confidence), the algorithm enters the second phase of normal exploration–exploitation (optimistic value iteration) but restricted to the now-established safe set. This carefully structured approach allows the authors to handle even infinitely large covering numbers by ensuring the effective policy class remains well-behaved. They prove that NCS-LSVI achieves the same order of regret as in the star-convex case (up to constant terms)​, marking the first provably efficient safe RL result for non-convex safe action spaces.

**Claims And Evidence:**

Provable Regret Bound: The authors claim a specific $\tilde{O}(d^3 H^4 K)$ regret bound (with additional dependence on the safety threshold $\tau$) for their algorithms​. This claim is backed by rigorous proofs outlined in the main text and detailed in the appendices. They present formal theorems for both the star-convex case (Theorem 5.1) and the non-convex case (Theorem 5.4), each establishing the stated regret upper bound. The structure of the proof is clearly sketched: by developing high-probability confidence bounds and using the OCD technique, they derive uniform concentration results that lead to the regret guarantee.

Zero Constraint Violations: A standout claim is that the algorithm incurs no safety violations with high probability during learning​. This is a strong claim (ensuring safety at all times, not just asymptotically or on average) and is supported by clear reasoning. The authors ensure this by design: the agent always acts within an estimated safe set that is initialized with a known safe action and expanded cautiously. They provide a theoretical guarantee (with high probability) that the estimated safe set is a subset of the true safe set at every step, which directly implies no constraint violation occurs.

Effectiveness of OCD (Objective–Constraint Decomposition): The authors claim that OCD is a novel technique that properly bounds the covering number in the star-convex scenario and “resolves an error in a previous work”. This refers to correcting the analysis of Amani et al. (2021). They give a detailed explanation of the previous error: in prior work, the value function’s covering number was treated like in standard RL, but in safe RL the feasible action set depends on past data, invalidating a direct application of covering arguments​.

**Essential References Not Discussed:**

There are possibly a few key missing references. Berkenkamp et al. (2017) – Safe Model-Based Reinforcement Learning with Stability Guarantees. This work deals with safe exploration in continuous state spaces using control-theoretic (Lyapunov) methods. The idea of slowly exploring to expand the region of safe actions is explored in this paper. It would be nice if the author's include a discssuion re this method.

**Experimental Designs Or Analyses:**

Soundness of Design: The experiment is designed to mirror the paper’s problem setting: an autonomous driving merging scenario is chosen to exemplify an MDP with an instantaneous hard constraint (stay in lane) and a non-convex action feasibility due to an external collision-avoidance system.

Clarity of Goals and Metrics: The goal in the experiment is for the agent to learn an optimal policy (presumably merging efficiently without leaving its lane) and the metrics used are regret and safety violations which are suitable in this context.

Results and Analysis: The results (as described) show that with a sufficiently long pure exploration phase ($K'=2000$ episodes of safe random exploration), the algorithm achieves sublinear regret growth, indicating it successfully learns the optimal policy.

This analysis is correct but minimal – it states the key observation (successful learning with sublinear regret) and doesn’t delve into much more detail. For instance, they didn’t explicitly say what the optimal policy’s performance was or how quickly the algorithm approached it in absolute terms.

Missing baselines: No baseline algorithm is compared. It would be informative to compare NCS-LSVI with, say, the initial safe policy (which would have zero violation but likely higher regret, essentially a flat line of regret increasing linearly if it never learns). Or compare with a naive RL algorithm that ignores safety (which would likely achieve lower regret initially but incur violations). Such comparisons could illustrate the trade-off between safety and performance. The absence of baselines means we only see that the algorithm works, but not how much better it is than “do nothing” or how much worse it might be than an unsafe method. Including at least the “Initial Safe Policy” as a baseline regret (which would accumulate regret if that policy is suboptimal) would contextualize the results.

Lack of multiple scenarios: It would be nice to see another environment – perhaps a different type of constraint or a different shape of safe region – to confirm the algorithm’s versatility.

**Methods And Evaluation Criteria:**

Proposed Methods: The paper proposes two main methods: the analytical OCD technique and the algorithmic framework NCS-LSVI. Both are well-motivated and appropriate for the problems they target. The OCD technique is essentially a novel analytical lens that decomposes the problem of bounding the value function class into objective and constraint parts​. This method is highly appropriate for the star-convex safe RL problem – it directly addresses the core difficulty of a changing feasible action set. The NCS-LSVI algorithm is also well-designed for the given problem. Safe exploration is notoriously hard when the safe region is not a single nice set. The two-phase approach – first conservative exploration using a safe policy, then optimistic exploitation/exploration within the enlarged safe set – is a sensible strategy.

Benchmark Evaluation (Datasets/Environments): The paper’s primary experimental evaluation is on a custom autonomous driving simulation (vehicle merging scenario). In this scenario, the agent must learn to merge lanes while obeying a lane-keeping constraint; a separate module (called “Collav”) filters out obviously unsafe (collision-causing) actions, making the remaining safe set for lane-keeping non-convex (because the safe actions at each moment can be split by the removed unsafe ones)​. This setup is a reasonable and relevant benchmark for the problem. It’s not a standard benchmark from prior literature, but safe RL scenarios often require custom environments to test specific constraints.

While the chosen scenario is appropriate, one could assess whether the range of experiments is sufficient. The paper appears to present only this one primary scenario in the main text (though they mention additional details and experiments in Appendix A​). The merging scenario provides one data point of success. It would have been beneficial to see the algorithm tested on a variety of environments or constraints – for example, a different robotics task or a synthetic grid-world with a complicated safe region – to further validate generality.

**Other Comments Or Suggestions:**

Clarity – Definition of the Local Point Assumption: The paper introduces a “Local Point Assumption” for the non-convex case (mentioned around Section 3). While the general idea can be inferred (it ensures some local connectivity of the safe set around the initial policy), it would benefit the reader to have a very clear definition in the main text

In the description of OCD’s results, the text says “(See Rmark 6.1)”

“scaler $\alpha_3$” in Appendix (it should be “scalar” I think)

**Other Strengths And Weaknesses:**

Strengths:
The paper tackles a crucial problem in reinforcement learning – ensuring safety at every step of learning.
The work makes original contributions in theory. The introduction of the Objective–Constraint Decomposition (OCD) is a novel analytical technique that advances understanding of how to handle dynamic feasible sets in RL​.
A major strength is that the paper provides strong theoretical guarantees – provably sublinear regret (meaning the agent approaches optimal performance efficiently) while incurring zero safety violations (with high probability)​.
The paper demonstrates a deep understanding of prior work by identifying and correcting a mistake in earlier research (Amani et al. 2021).

Weaknesses:

Restrictive Assumptions: One of the paper’s weaknesses is that the results rely on strong assumptions that may limit direct real-world applicability. The linear MDP assumption (linear function approximation with known feature map) is a simplification that might not hold in complex environments where one would truly need safe RL (e.g., driving with raw images, robotics with complex dynamics). Similarly, the requirement of a known initial safe action/policy and the Local Point Assumption mean the method requires prior knowledge to get started and that the safe region is nicely connected. If the environment’s safe space is disconnected or if no baseline policy is known, the current approach wouldn’t apply. These assumptions, while necessary for theoretical guarantees, weaken the generality of the results.

The derived regret bound, $\tilde{O}(d^3 H^4 K(1+1/\tau)\sqrt{\ln(1/\tau)})$, has quite large polynomial factors ($d^3 H^4$) and a linear dependence on $K$ (for the leading term)​. This suggests that the worst-case sample complexity could be high. In simpler terms, the algorithm might require a large number of episodes to reach near-optimal performance, especially if the feature dimension or horizon is long.

Limited Empirical Evaluation: The experimental evaluation, although positive, is quite limited. Only one primary scenario (autonomous merging) is presented in the main paper. There is no comparison to any baseline methods or ablation of algorithm components. This makes it hard to gauge how robust or effective the approach is relative to alternatives or in different conditions.

Scope of Safety Consideration: The paper deals with a single constraint (like one cost with threshold $\tau$). In many real applications, there could be multiple constraints (e.g., joint torque limits, collision avoidance, battery usage all at once). The current framework is not explicitly extended to multiple constraints.

Algorithm Complexity and Practicality: The algorithm NCS-LSVI is conceptually sound, but it may be complex to implement in practice. It likely requires maintaining large confidence sets, computing optimistic value functions via some form of extended value iteration at each episode, and carefully managing two phases. For high-dimensional problems or ones requiring function approximation beyond linear, this could be challenging.

Another minor weakness in presentation is that the related work discussion is largely deferred to the appendix. This can make it a bit difficult for a reader of the main paper to immediately see how this fits with prior efforts.

**Questions For Authors:**

1. The regret bound derived is $\tilde O(d^3 H^4 K + \text{const})$. Do you believe this dependence on $K$ (essentially linear) and on $d, H$ is order-optimal for safe RL, or could it be improved? In particular, is it possible to achieve a $\tilde O(\sqrt{K})$ regret (as in unconstrained linear RL) while still guaranteeing zero violations, or is the linear-in-$K$ growth a fundamental cost of imposing instantaneous safety?

2. The bound has factors $(1+1/\tau)\sqrt{\ln(1/\tau)}$ indicating slower learning for smaller $\tau$. How critical is knowing the exact threshold value $\tau$ for the algorithm?

3. How would you extend your approach to multiple simultaneous safety constraints (e.g., constraints $c_1(s,a)\le \tau_1$ and $c_2(s,a)\le \tau_2$ that must both hold)?

4. The Local Point Assumption essentially assumes a connected safe region around the initial safe policy. Suppose in an environment the safe state-action space actually consists of two disjoint regions, A and B. The initial safe policy lives in region A, but the globally optimal safe policy is in region B (which is not reachable following safe actions from A). In that case, your algorithm would never discover region B (since it can’t safely get there). Is my understanding correct that the method is limited to the connectivity of the initial safe region? And if so, do you have thoughts on strategies to deal with such situations?

5. In NCS-LSVI, the length of the pure exploration phase $K'$ is a crucial parameter. How should one choose $K'$ in practice? The experiments tried a few fixed values (like 2000) – was this based on a theoretical formula (e.g., a function of $\epsilon,\iota,d,H$) or more from empirical intuition? If $K'$ is chosen too small, what failure mode do you observe – is it that the safe set hasn’t stabilized and thus the algorithm remains overly cautious or incurs regret due to a small feasible action space? Conversely, if $K'$ is very large, aside from more exploration cost, does it ever hurt the stability (maybe by over-exploring)?

6. You mention interest in exploring “Deep Safe RL” with nonlinear function approximation. Do you foresee the OCD technique extending to certain classes of nonlinear function approximators (for example, kernel methods or neural networks with particular architectures)?

7. In your algorithm, you maintain an estimated safe set of actions at each state. How is this represented and updated in implementation? For example, in the driving scenario, is the safe set represented implicitly via upper confidence bounds on the constraint function $c(s,a)$? Do you discretize actions or use some parametric form to check which actions are safe?

**Relation To Broader Scientific Literature:**

The paper positions its contributions in the context of existing literature on safe reinforcement learning and constrained MDPs, particularly those with theoretical guarantees.

Safe RL with Instantaneous Constraints: Prior to this work, Amani et al. (2021) tackled a very similar problem – safe RL in linear MDPs with per-step (instantaneous) constraints under the assumption of a star-convex safe action set​. Amani et al. introduced the idea of maintaining an estimated safe set and guaranteed safety by always acting within that set. However, as noted in the paper, their theoretical analysis claiming sublinear regret was flawed. The current work directly builds on and corrects Amani et al.’s approach. The OCD technique in this paper fixes the analytical gap, enabling a correct proof of sublinear regret in the star-convex case.

**Theoretical Claims:**

Core Analytical Approach: The authors adapt the optimism in the face of uncertainty approach (common in RL theory) to the constrained setting. They maintain confidence intervals for the estimated model or value function and define an optimistic value function that the agent uses to choose actions. This standard approach is complicated here by the safety constraint, which limits the action set. The authors identify that the usual uniform convergence arguments (which rely on fixed function classes) need refinement because the feasible action set is history-dependent​. Their analysis via the OCD technique splits the problem: one part deals with the usual uncertainty in the $Q$-function approximation, and another part deals with how errors in constraint estimation can affect the value function. This decomposition is conceptually sound – it acknowledges that even if two $Q$-functions are similar, their induced policies could differ if their feasible actions differ, and it bounds that difference by leveraging star-convexity​.

Soundness of Theoretical Results: The regret bounds derived are high-level plausible given the problem difficulty. For star-convex safe sets, the authors essentially reclaim a sublinear regret guarantee similar to unconstrained RL (with some extra factors for safety).

Assumptions and Their Restrictiveness: The theoretical results do rely on several assumptions, which are explicitly stated. It’s important to assess if any of these are too restrictive or unrealistic:

Linear MDP assumption: The paper assumes the MDP’s dynamics and rewards can be embedded in a linear feature space of dimension $d$. This means essentially that $Q^*$ functions lie in a known $d$-dimensional linear span (or the transition kernel is linear in features).

Known initial safe action/policy: The authors assume that at each state (for star-convex case, as in Amani et al. 2021) or at least at the start of learning (for non-convex case via the Local Point Assumption), the agent has some action it knows is safe​. In the non-convex scenario, the Local Point Assumption (detailed in Section 3.2) likely means that from any state reachable by the initial safe policy, there exists a safe action “not too far” in feature space from the initial safe action – ensuring a locally connected safe region. This is a critical assumption for safety: it essentially provides a foothold to begin exploring without any risk. If no initial safe action is known, then ensuring zero violations from scratch is generally impossible (unless the agent is extremely conservative or gets external guidance).

Safety threshold $\tau$ and cost function: The problem formulation involves a cost function $c(s,a)$ and a threshold $\tau$ such that actions are safe if $c(s,a)\le \tau$ (presumably)​. They likely assume $c(s,a)$ is known or at least can be observed (the agent can tell if it violated the constraint or measure the cost feedback). Possibly, $c(s,a)$ might also be linear in the features or share the linear structure (the paper doesn’t explicitly say, but many constrained RL works assume a linear structure for cost if they assume it for reward).

In terms of how these assumptions affect applicability: the most limiting one is perhaps the need for an initial safe action/policy and local safe connectivity.

---

> ### Author Rebuttal · Authors · 2025-04-01
>
> Thank you for the thoughtful feedback. We respond to each of your comments below.
>
> **Q1** Is the regret bound $\tilde{O}(d^3 H^4 K + \text{const})$ optimal, or can a bound of $\tilde{O}(\sqrt{K})$ be achieved?
>
> **A1** Please note that the regret is already $\tilde{O}(\sqrt{K})$. The actual  the regret bound is given by  $\tilde{\mathcal{O}}((1 + \frac{1}{\tau}) \sqrt{\log(\frac{1}{\tau}) d^3 H^4 K} + \frac{1}{\epsilon^2 \iota^2}),$ as stated in Theorems 5.1 and 5.4. This is already sublinear in $K$, and  the dependence on $d$ and $H$  is under the square root, which matches the unconstrained regret in Jin et al. (2020) up to additional terms accounting for instantaneous constraints and non-star-convexity. We're happy to address any further questions.
>
>
>
>
>
> **Q2** How important is it to know the exact value of $\tau$?
>
> **A2** In many applications, $\tau$ is known. For example, in lane keeping, the car can measure its distance from lane boundaries, and in financial or power systems, $\tau$ corresponds to budget or power limits. When $\tau$ is unknown, a conservative estimate can be used, and our algorithm and regret bound continue to hold. While this may lead to a slightly higher regret, the dependence on $K$ remains $\tilde{\mathcal{O}}(\sqrt{K})$. The regret’s dependence on $\tau$ is inherent, as established by the $\Omega(1/\tau^2)$ lower bound in Theorem 20 of Pacchiano et al. (2024).
>
>
> **Q3** How would you extend your approach to multiple constraints?
>
> **A3** Our framework extends to multiple constraints by defining $\mathcal{A}^k_h(s) = \cap_{j=1}^M \mathcal{A}^{k,j}_h(s)$ in Line 11, where each $\mathcal{A}^{k,j}_h(s)$ is the estimated safe set for constraint $j$.  Hence, our approach is readily applicable to multi-constrained settings. We will include the above discussion in the final version.
>
>
> **Q4**  Is the method limited by the connectivity of the initial safe region?
>
> **A4** Unlike star-convexity, which requires global connectivity, the Local Point Assumption only imposes local conditions near the initial safe action and constraint boundary, making it suitable for disconnected decision sets. Our method, NCS-LSVI, is specifically designed to handle such environments.
>
>
> **Q5** Limited Empirical Evaluation
>
> **A5**  Please see our response to Reviewer sJHh, Comment A2.
>
>
> **Q6** How should K′ be chosen in NCS-LSVI?
>
> **A6**  Based on Theorem 5.4, in non-star-convex settings, $K'$ should be $O(\log(K)/(\epsilon^2 \iota^2))$. In star-convex cases (Theorem 5.1), $K' = 0$, so no pure exploration is needed. The choice of \(K'\) only affects regret and does not impact the safety guarantee or cause stability issues.
>
>
> **Q7** Can the OCD technique be extended to nonlinear settings?
>
> **A7** ​​ The key insight of OCD, decoupling the effect of Q-function estimation from that of the estimated safe set, naturally extends to nonlinear settings. In these settings, the policy is still recovered over an estimated safe set, so controlling both sources of error remains essential. With appropriate smoothness assumptions, our covering number analysis can be adapted.
>
> **Q8** The method relies on strong assumptions, such as linear MDPs and a prior safe policy.
>
> **A8** Linear MDPs are widely studied and effective in practice. Zhang et al. (2022) demonstrate state-of-the-art performance on MuJoCo and DeepMind Control benchmarks, and Jin et al. (2020) show that linear MDP solutions offer regret guarantees even when the true MDP is nonlinear. Also, a safe sub-optimal policy can often be identified offline using domain knowledge (Amani et al., 2019; Khezeli & Bitar, 2020; Shi et al., 2023).
>
>
>
>
> **Q9** NCS-LSVI may be computationally expensive. How is the estimated safe set computed?
>
> **A9**  As we explained in Appendix A, to improve computational efficiency, our implementation updates the $Q$-function estimates every $2^n$ episodes rather than at each episode.
> Regarding the complexity of maintaining the estimated safe set and computing UCB-based values, in some cases we can skip the explicit computation of lines 11 and 12 in Algorithm 1 and directly solve the constrained optimization in line 13. Since the $Q$-function is linear in $\phi$, the maximum lies on the boundary of $\phi(s, \mathcal{A}_h^k)$. When the safe set has a favorable structure (e.g., unions of convex sets), this is tractable. In complex or nonlinear cases, we use discretization, heuristics, or nonconvex solvers. We will add this to Appendix A.
>
>
> **Q10** The related works, omits Berkenkamp et al. (2017).
>
> **A10** We will move the related work to the main text and include Berkenkamp et al. (2017) in the final submission.
>
>
>
>
>
>
>
> **References**
>
> Zhang et al. (2022), Making Linear MDPs Practical via Contrastive Representation Learning, International Conference on Machine Learning (ICML), pp. 26447–26466. PMLR.

---

### Decision · Program_Chairs · 2025-05-01

**Decision:**

Accept (poster)

**Comment:**

Provably Efficient RL for Linear MDPs under Instantaneous Safety Constraints in Non-Convex Feature Spaces


Paper-Summary: The paper presents a new algorithm and theoretical analysis for safe reinforcement learning (RL) in settings with instantaneous hard safety constraints, particularly when the feasible action space is non-convex. The authors propose a novel regret-minimizing algorithm called Non-Convex Safe Least Squares Value Iteration (NCS-LSVI), designed for linear MDPs where the geometry of the safe decision space may be star-convex or non-star-convex. For the star-convex case, they introduce a new Objective–Constraint Decomposition (OCD) technique to bound the covering number of value functions, correcting a key mistake in prior work.  Their regret analysis explicitly characterizes the dependence on the safety threshold and problem dimensions. Additionally, simulations validate the effectiveness of their algorithm and theoretical claims.


Review and feedback: We received 4 expert reviews, with scores, 3, 3, 3, 4, and the average score is 3.25.

The reviewers are generally positive about this paper. Reviewers noted that this paper fills a key gap in the literature by providing a correct and general regret analysis for safe RL under linear MDPs. The OCD technique proposed is novel and it elegantly resolves the analytical challenge of history-dependent feasible sets, and helps to characterize the regret bound. Reviewers have also appreciated the innovation and presentation style of the NCS-LSVI algorithm that leverages a two-phase learning protocol (safe exploration followed by optimistic planning). They particularly noted that this is the first paper to guarantee zero safety violations in a non-star-convex setting with high probability. Although limited in scope, the experiments are well-aligned with the theoretical setting.

Reviewers have also made suggestions for improvement. Most reviewers have asked for more detailed experimental evaluation and comparison with other baseline algorithms. More than one reviewer has asked for more explanation/justification about the assumptions used and intuitive proof sketches. Please address these comments/suggestions while preparing the final submission.